# Modeling short visual events through the BOLD moments video fMRI dataset and metadata

Benjamin Lahner [1] ✉, Kshitij Dwivedi [2,3], Polina Iamshchinina [2], Monika Graumann [2], Alex Lascelles [1], Gemma Roig [3,4], Alessandro Thomas Gifford [2], Bowen Pan[1], SouYoung Jin[1], N. Apurva Ratan Murty [5,6], Kendrick Kay[7], Aude Oliva[1,8] & Radoslaw Cichy[2,8]

Studying the neural basis of human dynamic visual perception requires extensive experimental data to evaluate the large swathes of functionally diverse brain neural networks driven by perceiving visual events. Here, we introduce the BOLD Moments Dataset (BMD), a repository of whole-brain fMRI responses to over 1000 short (3 s) naturalistic video clips of visual events across ten human subjects. We use the videos' extensive metadata to show how the brain represents word- and sentence-level descriptions of visual events and identify correlates of video memorability scores extending into the parietal cortex. Furthermore, we reveal a match in hierarchical processing between cortical regions of interest and video-computable deep neural networks, and we showcase that BMD successfully captures temporal dynamics of visual events at second resolution. With its rich metadata, BMD offers new perspectives and accelerates research on the human brain basis of visual event perception.

Understanding visual events is a hallmark of human intelligence that engages a distributed and functionally diverse network of cortical regions. For example, extracting the meaning of even a simple event, such as a person opening a door, requires at a minimum parsing the scene into relevant components (a person, door, indoor room)[1–6] and integrating these components over time for accurate action recognition (the door is being opened, not closed)[7–11], including socially relevant cues (is the person opening the door angry or friendly?)[12–14]. Part of the event is also encoded into brain memory regions for later recall[15–19]. Due to the richness of this cortical network, ecologically-valid visual event understanding is challenging to study with experimental rigor.

To advance empirical evaluation of this network, we introduce the BOLD Moments Dataset (BMD), a dataset of fMRI responses to 1102 3-second naturalistic videos from ten human adult participants. The videos span a range of events that humans may witness in daily life and evoke a representative range of neural responses. Each video clip in BMD is also human-annotated with object, scene, action, sentence, and memorability labels to relate brain activity to meaningful aspects of visual events.

Short naturalistic videos (3 s long) strike a good balance between ecological validity and experimental control, making them well suited for investigating human perception of dynamic visual content. Short video stimuli supplement work using still images by driving a greater extent[20–24] and different pattern[9,11,25,26] of neural responses. Short visual events also complement work using longform movies by isolating meaningful actions from long term contextual effects[18,27,28].

[1]Computer Science and Artificial Intelligence Laboratory, MIT, Cambridge, MA, USA. [2]Department of Education and Psychology, Freie Universität Berlin, Berlin, Germany. [3]Department of Computer Science, Goethe University Frankfurt, Frankfurt am Main, Germany. [4]The Hessian Center for AI (hessian.AI), Darmstadt, Germany. [5]Department of Brain and Cognitive Science, MIT, Cambridge, MA, USA. [6]School of Psychology, Georgia Institute of Technology, Atlanta, GA, USA. [7]Center for Magnetic Resonance Research (CMRR), Department of Radiology, University of Minnesota, Minneapolis, MN, USA. [8]These authors jointly supervised this work: Aude Oliva, Radoslaw Cichy. ✉e-mail: blahner@mit.edu

Below, we detail BMD's experimental design and investigate human dynamic perception using a diverse set of modeling approaches. Aided by BMD's large number of stimuli and widespread reliable activity, we use a video-computable deep neural network (DNN) trained on an action recognition task to predict brain responses through all cortex, including the dorsal visual and parietal cortices. We additionally input frame-shuffled videos and the first/last second of video frames into DNNs to highlight the temporal content captured by the brain responses: the shuffled video inputs significantly reduce the DNN's prediction accuracy in most regions of interest, and the first (last) second of video frames best explains the early (late) BOLD signal estimates in early and ventral visual cortex. A representational similarity analysis[29] comparing language model embeddings of BMD's rich stimuli metadata with multivariate brain activity reveals that the sentence descriptions correlate with ventral and dorsal visual cortex at nearly twice the amount of the object, scene, and action labels. Finally, we show the videos' memorability scores correlate with brain responses not only in the visual cortex, as seen with images[30–32], but also in the parietal cortex to further reveal how dynamic visual perception and memory interface.

Together, BMD's 1102 thoroughly sampled short video stimuli isolate brain responses to ecologically-valid but experimentally controlled dynamic content. This dataset is well-suited to address a range of scientific questions: developing methodologies to model rapid event BOLD signals[33–35], characterizing interactions between visual processing pathways[9,11,14,36], and bridging the gap between still image and longform movie perception[37–40] (for further discussion, see Supplementary section The added value of a short video versus a static image neuroimaging dataset). Crucially, BMD's shared experimental design across subjects enables robust and generalizable conclusions, and its preparation in BIDS format facilitates easy adoption among researchers. The range of stimuli and subjects, reliable brain activity, and rich metadata will enable a wide range of interdisciplinary analyses to reveal the neural mechanisms underlying visual event understanding.

## Results

### Sampling brain activity for 1102 distinct visual events

We sampled 1102 naturalistic 3-second videos depicting diverse dynamic visual events from the Memento10k video memorability dataset[41], which is a subset of the 1 million video large Moments in Time (MiT) and Multi-Moments in Time (MMiT) datasets[42,43] (Fig. 1a). As used here, the term visual event is a 3-second video depicting someone or something performing an action in the real world. Understanding a visual event reflects a viewer's ability to parse and integrate the video's spatiotemporal information to report its relevant happenings, such as in a sentence description. The 3-second events from Moments in Time datasets were typically cropped from a longer in-the-wild video (e.g., a 3-second segment of a dance recital) and selected to contain no emotionally jarring or inappropriate content. The naturalistic content engages rich neural processes[44–47]. The 3-second video, approximately the duration of human working memory[19,48,49], is an ideal duration to capture brain responses to meaningful events in context (e.g., opening a door, cooking BBQ food, playing tennis). Videos shorter than 3 seconds risk capturing incomplete actions (e.g., extending the arm) while videos longer than 3 seconds risk undesired complexity by capturing a sequence of actions (e.g., opening a door, then walking into the kitchen, and finally removing a jacket) that introduce highly correlated events that are difficult to disentangle from brain responses.

The experimental design has the benefits of both a large stimulus set and several stimulus repetitions to enable downstream analyses that depend more strongly on one or the other. We thus divided the 1102 stimuli into two non-overlapping sets, one consisting of 1000 stimuli with three repetitions per subject (the training set) and the other 102 stimuli with ten repetitions per subject (the testing set). Together, this amounts to 4020 unique fMRI trials per participant, and

40,200 unique fMRI trials across the entire dataset. The large number of trials and participants enable direct comparisons of results across participants and invite potential integration of data across participants[18,50–52].

Functional brain responses were recorded with whole-brain 3 T fMRI at $2.5 \times 2.5 \times 2.5$ mm voxel resolution to densely sample the widely-distributed cortical responses to video across the whole cortex[24–26,53–56]. The fMRI experiment was split into 5 sessions. Session 1 (Fig. 1b) contained crucial auxiliary brain measurements, interleaving high-resolution T1- and T2-weighted structural, video-based functional localizer[57], and resting state scans[58,59]. Sessions 2 to 5 (Fig. 1c) contained the main experiment and had identical structure. Each video trial was 4 s long, consisting of a 3 s silent video presentation followed by a 1 s intertrial interval.

### Semantic and behavioral metadata on visual events

Revealing how the brain mediates visual event understanding benefits from detailed, human-labeled descriptions of a visual event. Thus, we used human crowd-sourced experiments to annotate each clip with five word-level scene, object, and action labels, five sentence-level text descriptions, one spoken transcription, one behavioral memorability score, and one memorability decay rate (Fig. 1a).

Each metadata category was collected in a separate experiment. The possible scene, object, and action labels were sampled from the Places365[60], THINGS[40,61], and Multi-Moments in Time[43] datasets, respectively, for their carefully crafted label coverage and extensive overlap with other resources available in computer science. Because most videos clearly contain more than one object, the annotators in the object label experiment were instructed to label three different objects for a total of 15 object labels per video. The 1102 BMD stimuli cover 305 of 365 possible scene labels, 1002 of 1854 possible object labels, and 261 of 292 possible action labels. The sentence text descriptions (13.06 mean ±2.800 std words per sentence) were typed free-form by the annotator to detail salient interactions in the video and summarize the pertinent content. The spoken transcriptions, collected through free-form audio recordings and transcribed (audio not available due to privacy considerations), tend to be more verbose with additional emotional and linguistic subtleties present in speech but often not in text (26.72 mean ±17.55 std words per transcription)[62]. Finally, to determine how visual event understanding interfaces with memory, we use a crowd-sourced memory game to behaviorally measure if participants recognize a visual event at a later time period (memorability: 0.8422 mean ±0.0888 std) and how this recognition performance fades over time (memorability decay rate: −0.0014 mean ±0.0011 std)[41].

These metadata allow the use of grouping, subdivision, or other transforms of these measures to investigate additional perceptual and cognitive processes related to visual event understanding (see Supplementary Fig. 8 for metadata distributions between train and test splits). We provide first example analyses on this basis in the following sections.

### (f)MRI data processing, response modeling, and ROI definition

We provide raw as well as preprocessed versions of the MRI data in the community-backed and standardized BIDS format[63] to ensure transparent quality assessment, reproducible preprocessing, and easy-to-share results. The raw data gives researchers control over preprocessing to pursue research questions at any stage of the analysis pipeline. The preprocessed data allows for analyses into the spatial and temporal neural dynamics underlying visual event understanding, at both the group and single-participant level.

We processed the data using fMRIPrep[64] to achieve reproducible and transparent results. During acquisition, we sampled fMRI data at a TR of 1.75 s to achieve a densely sampled time course with respect to the onset of the stimulus[65]; since 1.75 does not evenly divide into the

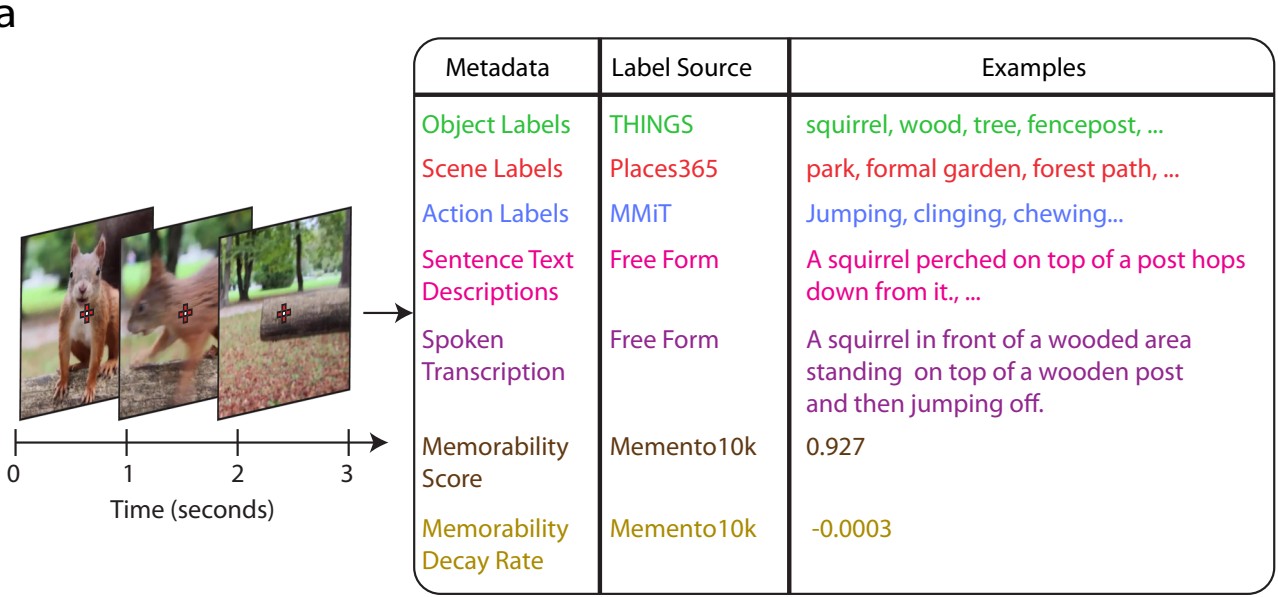

4 second trial length, the BOLD response to a single video across three repetitions might be sampled at onset delays of [0.5, 2.25, 4, 5.75,…], [0, 1.75, 3.5, 5.25, …], and [1, 2.75, 4.5, 6.25,…] seconds for the three (or ten) repetitions. In the preprocessing we then temporally resampled the data from a TR of 1.75 s to a TR of 1 s in order to acquire BOLD responses exactly time-locked to the onset of each trial (stimulus presentation happens every 4 s, i.e., a multiple of 1 s, but not 1.75 s). We used Finite Impulse Response (FIR) basis functions to accommodate variability in the hemodynamic responses at different locations in cortex, to resolve the temporal structure of the stimuli, and to address response overlap from the rapid event-related design. We modeled the hemodynamic response to visual events using data from 1-9 s after

**Fig. 1 | Experimental design and data acquisition. a** Stimuli and metadata: The 1,102 3-second video stimuli are sampled from the Moments in Time dataset[42,43] and annotated with: 15 object labels, 5 scene labels, 5 action labels, 5 sentence descriptions, 1 spoken transcription, 1 memorability score, and 1 memorability decay rate. Due to licensing restrictions, the video frames shown here are sourced from a representative video by NaIzletuSi ™ and cropped (YouTube: https://www.youtube.com/watch?v = QT0o0fTfZKE, CC BY 3.0 license: https://creativecommons.org/licenses/by/3.0/legalcode). **b** fMRI session 1: Subjects underwent T1- and T2-weighted structural runs interspersed with resting state and functional localizer runs to define various category selective regions. **c** fMRI sessions 2-5: Sessions were identical in design and consisted of a fieldmap run followed by thirteen runs (ten training runs and three testing runs) in random order. The 3-second videos were presented on a gray background at five degrees of visual angle overlaid with a red fixation cross. Stimuli presentation was followed by a 1-second intertrial interval composed of a red fixation cross on a gray background. Participants reported

luminance changes of the fixation cross occurring irregularly between videos with a button press and did so reliably (hit rate 0.964 ± 0.014 (mean ± SD)). Video frames are cropped from a representative video by NaIzletuSi ™ (YouTube: https://www.youtube.com/watch?v = QT0o0fTfZKE, CC BY 3.0 license: https://creativecommons.org/licenses/by/3.0/legalcode). **d** Region of Interest Definitions: Regions of interest for a representative subject (subject 1), functionally (from session 1) and anatomically defined. The ROI abbreviations are as follows: V1v−first visual area ventral; V1d−first visual area dorsal; V2v−second visual area ventral; V2d−second visual area dorsal; V3v−third visual area ventral; V3d−third visual area dorsal; hV4−human visual area 4; EBA−extrastriate body area; STS−superior temporal sulcus; RSC−retrosplenial cortex; FFA−fusiform face area; OFA−occipital face area; LOC−lateral occipital complex; PPA−parahippocampal place area; TOS−transverse occipital sulcus; V3ab−visual area 3a and visual area 3b; IPS0−intraparietal sulcus 0; IPS1-2-3−intraparietal sulcus 1, 2, and 3; 7AL−lateral area 7; BA2−Brodmann area 2; PFt−parietal area F, part t; PFop−parietal area F, part operculum.

stimulus onset (to account for the hemodynamic lag) in 1 s steps (i.e., 9 bins of 1 s length each) for each trial separately. The FIR model's flexible BOLD estimates, randomization of stimulus presentations across runs and sessions, and ability to average over multiple repetitions reduce potential unwanted memory effects.

To guide analysis in a region-specific manner, we used the auxiliary functional localizer data from session 1 to define a set of 22 regions of interest (ROIs) previously reported to be involved in visual perception of natural images, natural video, or motion (Fig. 1d)[5,55,56,66–68]. The set includes early visual, category-selective ventral visual, dorsal visual, and parietal regions.

This preprocessing suite ensures a low threshold for researchers to interact with the brain data at their desired processing level. All results presented here in the main text use version A of the dataset. We provide another preprocessed version of BMD (version B) available in more output spaces (version B is detailed in Supplementary section Version B preprocessing pipeline).

### (f)MRI image scans of high quality across subjects and task

To assess the quality of the raw and preprocessed MRI data, we used the open-source and community-based MRIQC analysis package[69]. This yielded a comprehensive set of 44 functional and 68 structural image quality metrics (IQMs) (full report available), of which we present a representative set of six structural (Fig. 2a) and six functional MRI metrics (Fig. 2b) (for more details on the IQMs, see Supplementary section Structural and functional scan quality assessment). Because no single metric can completely describe data quality, the selection of the IQMs to display here considered metrics especially relevant to functional or structural scans (e.g., Temporal SNR for functional scans and Contrast to Noise ratio for structural scans), metrics shared between functional and structural scans for a more cohesive set (e.g., SNR and Full-Width Half Maximum Smoothness), and metrics common across other literature and analysis packages to increase familiarity with readers (e.g., SNR, Framewise Displacement, AFNI Outlier Ratio, AFNI Quality Index). Since most quality metrics do not have a ground-truth reference value to compare against, we contextualize our values against the values of hundreds of anonymized studies with similar scanner parameters (1 < Tesla < 3, 1 <= TR < 3; aggregated with the MRIQCeption API).

These measures show that both the structural and functional BMD scans are of excellent quality. The distribution of all but one IQM for each subject falls within or is noticeably better than the distribution seen in similar fMRI studies (Fig. 2, green). The strong results of the IQMs tSNR (a measure of SNR over time), aor (an indicator of the number of outliers per fMRI volume), and aqi (a correlational measure of quality per volume) assure satisfactory functional SNR quality in light of the below-typical per volume SNR IQM. We highlight that within each participant, the range of quality metric values was especially consistent between the training and

testing sets (Fig. 2, blue and orange boxplots). This shows that the training and testing sets are of comparable data quality, facilitating analyses that depend on this split. Further, none of the 10 participants were outliers as indicated by consistently lower within- than between-participant variability, encouraging group result inference (see Supplementary Fig. 1 for results on the resting state and functional localizer scans).

### Reliable univariate and multivariate fMRI response profiles

We provide both univariate and multivariate metrics to evaluate the reliability of BMD relevant to the two main fMRI analysis traditions today: the univariate framework that focuses on local information at a single voxel scale[70–74], and the multivariate analysis framework that emphasizes the distributed nature of information in population codes[29,75,76]. Neural encoding and decoding analyses often use single-voxel responses to measure information content in an ROI[72], and hypothesis models of neural representations derived from computational or behavioral models can be easily compared to the brain in multivariate analysis frameworks[77]. These reliability measures provide intuition on data quality and can used at the discretion of the researcher to normalize results with respect to the noise in BMD's data.

To assess univariate reliability, we identify voxels whose beta value estimates satisfy a Spearman-Brown (SB) split-half reliability criterion (mean reliability of all combinations of 5-trial averaged splits from the 10-trial testing set) of $p < 0.05$ (assessed by stimulus label permutation):

$$SB = (2\rho/(1+\rho)) \tag{1}$$

Where $\rho$ is the Pearson correlation between the 5-trialed averaged splits.

To assess multivariate reliability, we perform representational similarity analysis (RSA) in a searchlight approach[78] to determine the upper (subject-to-group RDM correlation per voxel) and lower (leave-one-out RDM correlation per voxel) estimate of the noise ceilings. We report the univariate and multivariate reliability results across the whole brain (Fig. 3a, b for a representative subject, see Supplementary Figs. 2 and 3 for all subjects) and in ROIs (Fig. 3c, d).

In both the whole-brain univariate and multivariate reliability analyses, we observe statistically significant reliability values across the occipital, temporal, and parietal cortices, even extending into the frontal lobe. These ROI analyses show high explainable variance in a functionally diverse set of ROIs responsible for visual event understanding. This demonstrates that BMD is well suited for comprehensive and advanced analysis at both the single- and multi-voxel spatial scale.

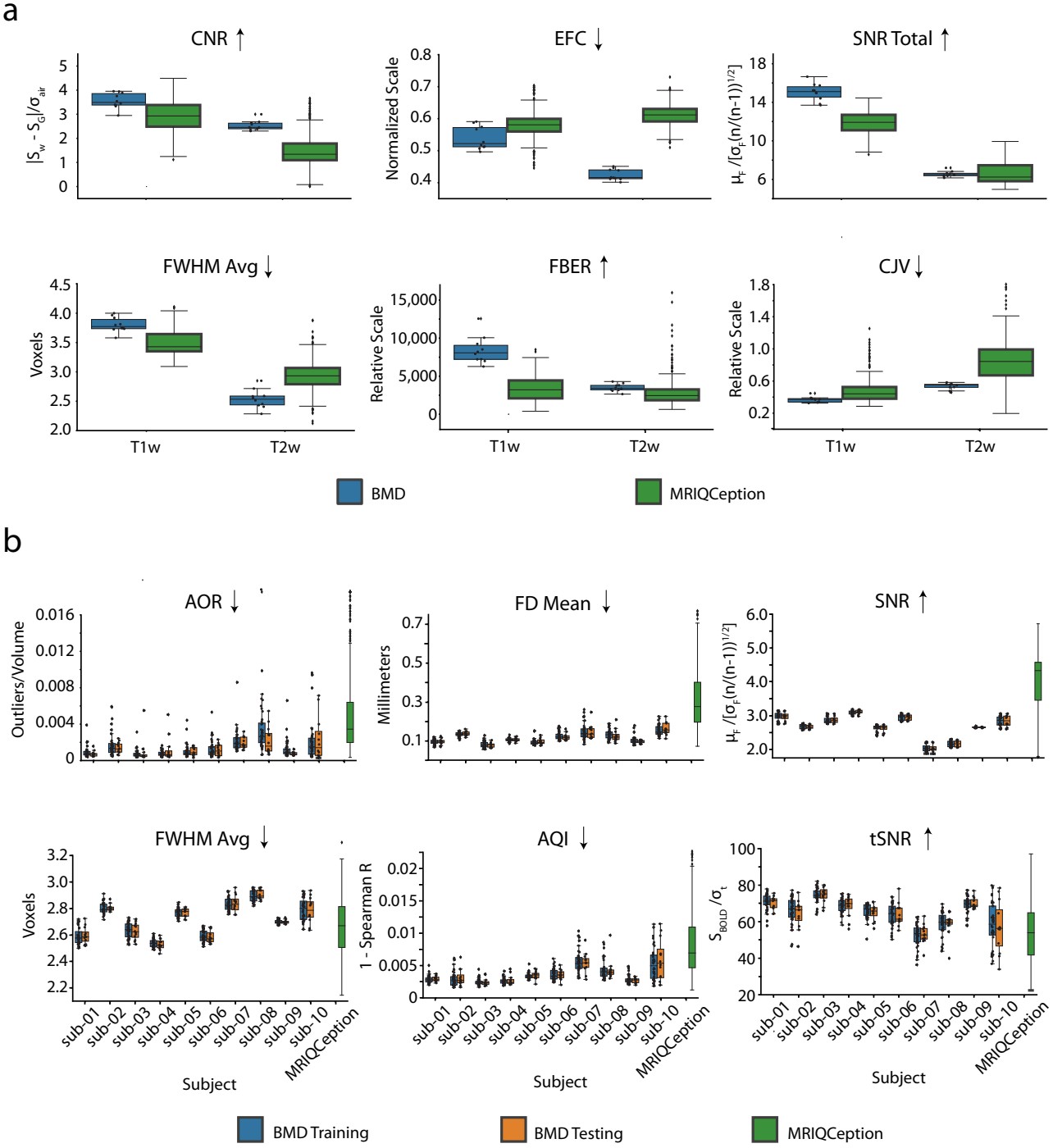

**Fig. 2 | Preprocessed scan quality measures. a** Data quality of the T1- and T2-weighted structural scans: Boxplots compare Image Quality Metrics (IQMs) of the T1- and T2-weighted structural scans of the BMD dataset (blue) (T1-weighted, $n = 10$; T2-weighted, $n = 10$) with a collection of anonymous data from MRIQCeption API (green) (T1-weighted, $n = 219$; T2-weighted, $n = 695$). We report the Signal to Noise Ratio (SNR Total), Contrast to Noise Ratio (CNR), Coefficient of Joint Variation (CJV), Entropy Focus Criterion (EFC), Average Full-Width Half Maximum Smoothness (FWHM Avg), and Foreground-Background Energy Ratio (FBER) as a summary of the structural scan quality. The overlaid points correspond to the data value from an individual subject in BMD. An up arrow after the IQM title means a higher value corresponds to better data quality, while a down arrow after the IQM title means a lower value corresponds to better data quality. Source data are provided as a Source Data file. **b** Training and testing functional scans data quality: Panels compare boxplots of IQM values for each subject for the training (blue) and testing (orange) functional runs in the BMD dataset (per subject: training, $n = 40$; testing, $n = 12$) with other anonymous BOLD data from the MRIQCeption API (green) (BOLD, $n = 624$).

## Modeling visual event understanding with a video-computable deep neural network for action recognition

A major goal in explaining brain responses is providing explicit quantitative models that predict the underlying computations and their cortical organizations[79]. Deep Neural Network (DNN)-based modeling has emerged as a dominant form of scientific modeling in visual neuroscience, due to their image-computable design, biologically-inspired architecture, and high neural prediction performance[38,74,80–83].

However, modeling visual events has been limited by the lack of a suitable dataset that accounts for complex distributed

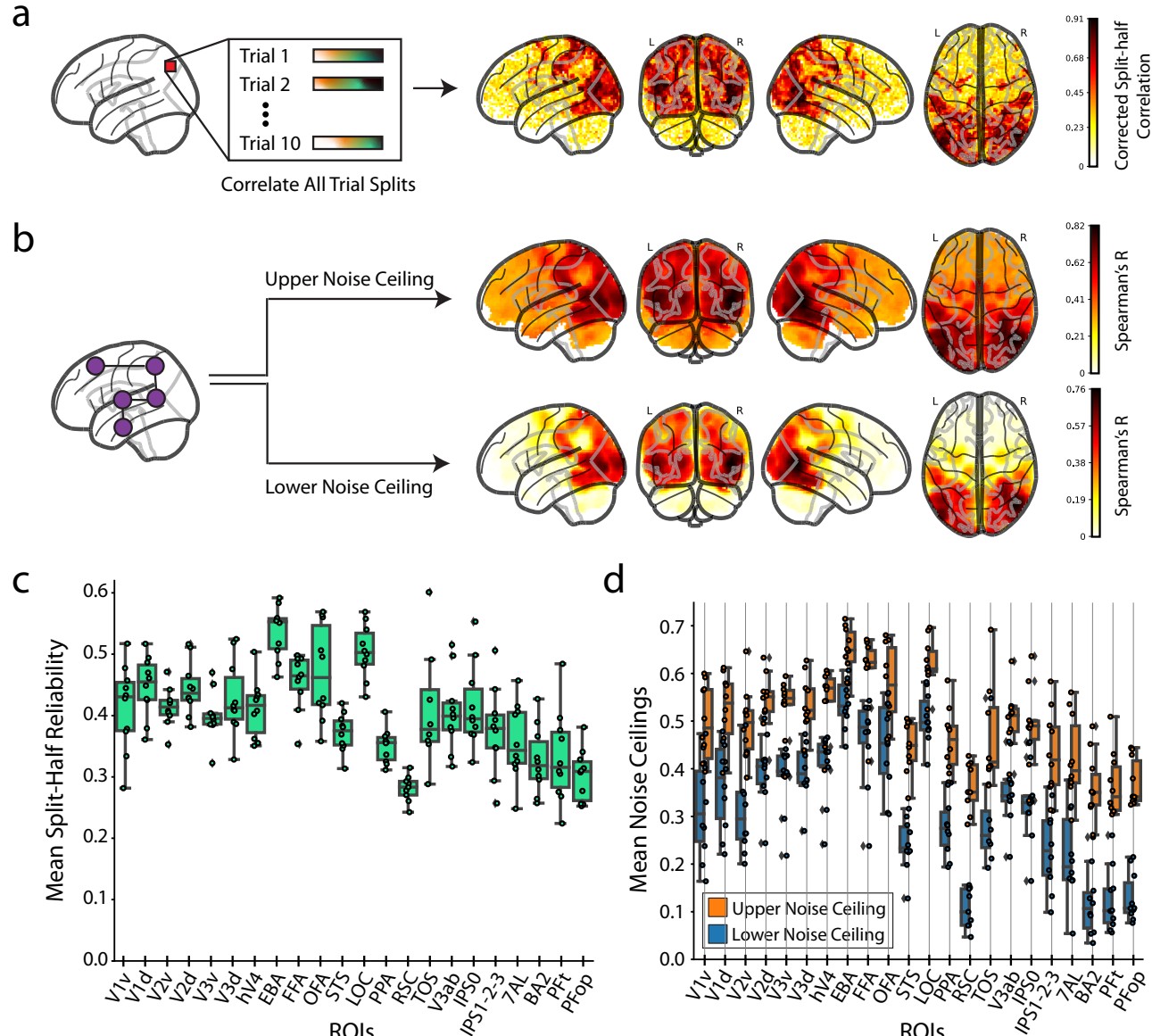

**Fig. 3 | Whole-brain and ROI reliability after response modeling. a** Whole-brain single-subject split-half reliability analysis: We perform a voxelwise split-half reliability analysis and present the voxels with Spearman-Brown corrected values that pass the reliability criteria (p < 0.05, permutation-based, one-sided) for subject 1. **b** Whole brain searchlight noise-ceiling analysis: We estimate the upper and lower noise-ceilings across the whole brain for a representative subject (subject 1) from a searchlight representational dissimilarity analysis (RSA). **c** ROI-based group split-half reliability: For each of the 22 ROIs, we present the mean split-half reliability across participants from the voxels passing the reliability criteria. Source data are provided as a Source Data file. **d** ROI-based group searchlight noise ceilings: For

each of the 22 ROIs, we calculate the upper (orange) and lower (blue) noise ceilings across participants. Source data are provided as a Source Data file. The box plots encompass the first and third data quartiles and the median (horizontal line). The whiskers extend to the minimum and maximum values within 1.5 times the interquartile range, and values falling outside that range are considered outliers (denoted by a diamond). The overlaid points show the value at each observation (n = 10 for all ROIs except transverse occipital sulcus (TOS, n = 8) and retrosplenial cortex (RSC, n = 9)). The brain responses used for the reliability analyses are the beta values averaged over TRs 5-9 (the peak of the BOLD signal) from the testing set.

processes across the whole brain[24], drastic differences to image understanding[9,11,23,25,26,84–86], and the temporal boundaries of a visual event[8,37,87,88]. BMD helps with its large number of short video stimuli and whole-brain responses.

Towards the goal of modeling visual events, we used a video-computable DNN following biological constraints. The DNN uses a recurrent ResNet50 backbone[89] that mimics the biological recurrent computations essential for human motion perception and categorization[90–94] and builds on the ResNet family's strong neural predictivity performance seen for still images[82,95]. The ResNet50's four recurrent blocks are connected using a Temporal Shift Module[96] (TSM) designed to process video input in the natural, uni-directional

temporal order. We train the model on an action recognition task using the same dataset from which the BMD stimuli were sampled, the Moments in Time dataset[42,43] (BMD stimuli were excluded from model training; for training details, see Methods section Action recognition TSM ResNet50 model training). We release this model to aid investigations of visual event understanding (model available at: https://github.com/pbw-Berwin/M4-pretrained).

We queried the relationship between each of the model's blocks and BMD. Using a voxelwise encoding model approach[72] (Fig. 4a), we observe a correspondence between DNN block depth and predictivity performance along the visual processing hierarchy and beyond (for encoding model details, see Methods section DNN block to cortex

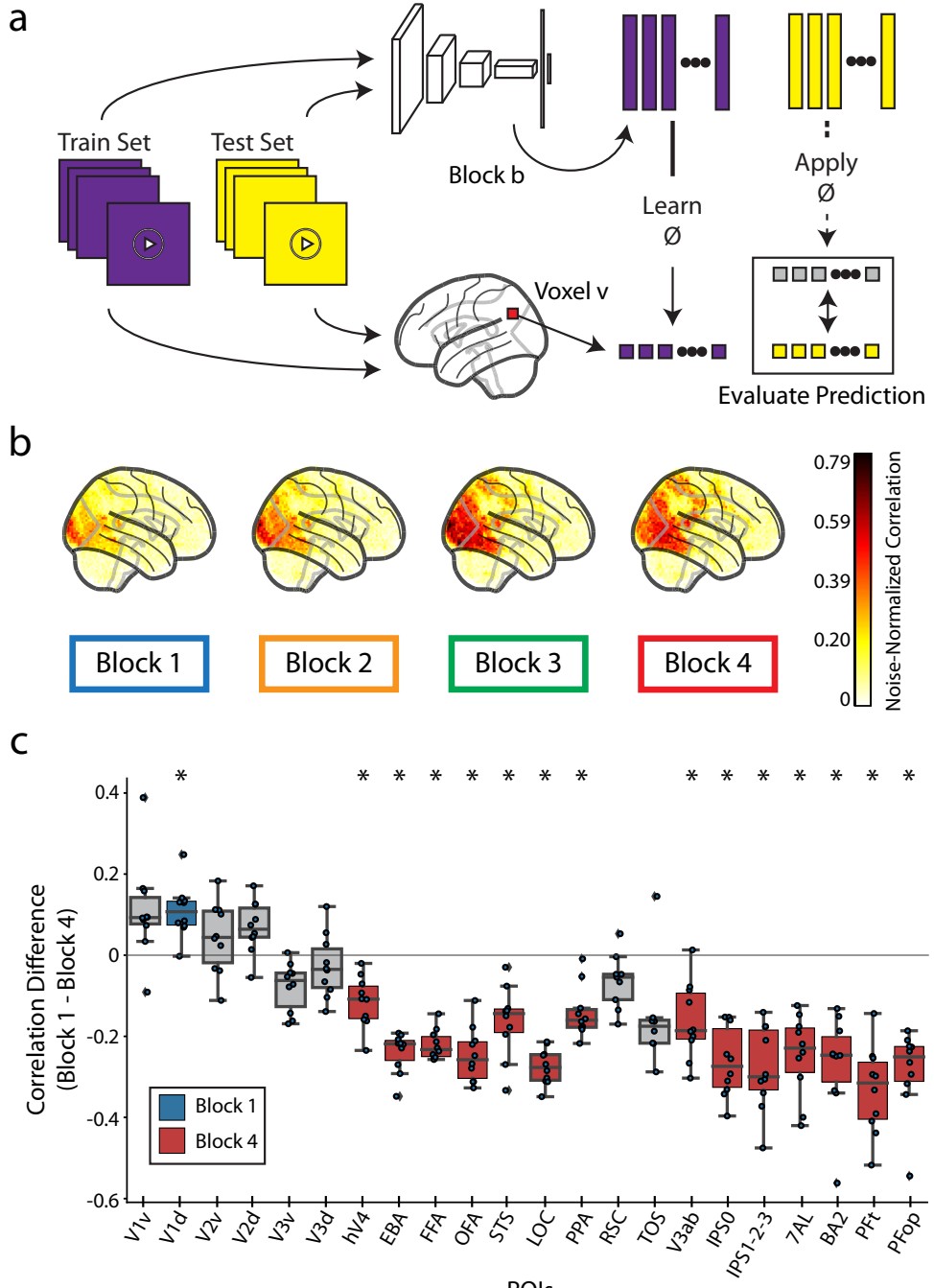

**Fig. 4 | Evaluation of biologically-similar video-based encoding model.**
**a** Voxelwise encoding model procedure: All videos are shown to both a TSM ResNet50 DNN and a human. Training set video embeddings are extracted from a block *b* of the DNN and used to learn a voxelwise mapping function to the human responses. This mapping is then applied to the testing set video embeddings to predict the brain response at each voxel. **b** Whole-brain encoding accuracy across blocks: We use the encoding model procedure with each of the four blocks of a TSM ResNet50 model trained to recognize actions in videos to predict the neural response at each voxel in the whole brain. The brain figures show the subject-average noise-normalized predictive correlation (divided by the voxel's upper noise ceiling) at each voxel. **c** ROI-based encoding accuracy difference: Difference in predictive performance between block 1 and block 4 at each of the 22 ROIs.

Predictive performance at each voxel is measured as the noise-normalized correlation between the brain responses and the predicted responses, averaged over all reliable voxels in each ROI. Significant ROIs are denoted with an asterisk and a color (blue for Block 1, red for Block 4, gray is not significant) corresponding to the significant layer ($p < 0.05$, one sample two-sided t-test against a population mean of 0, Bonferroni corrected across $n = 22$ ROIs). Source data are provided as a Source Data file. The box plot encompasses the first and third data quartiles and the median (horizontal line). The whiskers extend to the minimum and maximum values within 1.5 times the interquartile range, and values falling outside that range are considered outliers (denoted by a diamond). The overlaid points show the value at each observation ($n = 10$ for all ROIs except transverse occipital sulcus (TOS, $n = 8$) and retrosplenial cortex (RSC, $n = 9$)).

correspondence procedure). The predictivity of each of the DNN's four blocks progressively spreads across the brain from posterior to anterior (Fig. 4b). As a proxy for low-level and high-level features, we extract the video features at early (Block 1) and late (Block 4) blocks of

the DNN, respectively[83,95,97,98]. We then estimate the dominance of low-level and high-level feature processing across ROIs by computing the difference in predictivity between DNN Block 1 and DNN Block 4. We observe that predictivity of DNN Block 4 becomes significantly greater

than DNN Block 1 beginning in the ventral visual cortex and extending into dorsal visual cortex and parietal cortex (Fig. 4c), reflecting the increase of feature complexity in the representations across the visual processing hierarchy. We repeat this encoding analysis with different architectures and training diets to see how this result generalizes across models. Specifically, we extract features from a TSM MobileNetV2[96,99] and a TimeSformer[100] architectures trained on Kinetics-400[101] and HowTo100M[102] datasets, respectively. We find that both are good predictors of dorsal visual and parietal regions, but the TSM MobileNetV2 does not show increasing dominance of high-level model features along the cortical hierarchy (see Supplementary Fig. 7 for results on all architectures and blocks).

This result extends previous research demonstrating a hierarchical correspondence between DNNs and brains from still image stimuli[95,98,103–105] to dynamic video stimuli, a non-trivial outcome given that many cortical regions in the ventral visual and temporal cortex and beyond respond to stimulus features uniquely present in videos and not images (e.g., movement kinematics, temporal interactions)[9,11,13,106]. The finding that a transformer-based TimeSformer model follows a similar pattern to the TSM ResNet50 while the TSM MobileNetV2 does not invites further inquiry into the effect of training diet, parameter count, and architecture on visual event understanding. Finally, these results also help clarify previously conflicting results about whether or not DNNs trained on action recognition tasks accurately predict dorsal stream regions[36,107,108], showing that all three architectures accurately predict responses not only in the dorsal visual stream but also in the parietal cortex.

## fMRI responses capture temporal event structure

A real world event unfolds in a systematic, spatiotemporal sequence. Is the temporal structure of such an event important for its cortical representation? Shuffling the frames of a video effectively destroys its meaningful temporal structure. We reasoned that if a voxel captured an event's spatiotemporal relationships, unshuffled (meaningfully ordered) video input would correspond better to the brain responses than shuffled video input. The shuffled and unshuffled video inputs are both temporally dynamic and contain identical frame-averaged spatial content, thereby isolating the effects of the encoding of ordered temporal content.

To investigate, we measured the neural prediction performance of the DNN model introduced above when given unshuffled video input and shuffled video input (Fig. 5a). Since this DNN is engineered to process video input in a unidirectional temporal order, we know that shuffling the video frames will affect the DNN activation in at least one of the four blocks. By using the shuffled and unshuffled activations from these blocks to predict the brain responses, we can assess the effect of correct temporal ordering on our BMD brain responses (for details, see Methods section Shuffling analysis to determine importance of temporal order).

Our results show the frame-shuffled input decreased DNN prediction accuracy across most of the visual system. The whole-brain analysis (Fig. 5b) shows that this decrease in DNN prediction accuracy is most pronounced in both magnitude and coverage with increasing DNN blocks. The ROI analysis (Fig. 5c) reveals that early visual ROIs were sensitive to differences in all DNN blocks, while ventral and dorsal ROIs were most affected by differences in DNN blocks 3 and 4. These results suggest cortical regions have functionally-specific sensitivities to shuffled input in line with known feature preferences in the visual system and DNN computations[5,95,109].

In order to further elucidate the type of temporal dynamics in the brain that this frame shuffling analysis is affecting, we perform an additional small-scale experiment comparing the difference in prediction accuracy between a Temporal Shift Module (TSM) ResNet50[96] and a Temporal Segment Network (TSN)[110] ResNet50. Since the only difference between the two models is how spatial information is shared between frames, this analysis effectively isolates temporal integration. We find that implementation of TSM significantly improves encoding accuracy in early visual ROIs primarily from DNN blocks 3 and 4 (see Supplementary Fig. 9 for results). This pattern is different from the one observed from the frame shuffling analysis (Fig. 5c), suggesting that the brain activity is capturing various forms of temporal dynamics.

Lastly, we repeat this frame shuffling analysis on a TSM MobileNetV2[96,99] and TimeSformer[100] architecture trained on Kinetics-400[101] and HowTo100M[102] datasets, respectively, to see if the frame-shuffling results generalize to models of varying architectures, training diets, parameter counts, and task performance. We find that the TSM ResNet50 is the only model of the three that sees robust effects in most ROIs and DNN blocks (see Supplementary Fig. 10 for all results), implying that model architecture and the level of temporal information in the model training datasets may be closely tied to the effects of frame shuffling.

Together this battery of analyses demonstrates that most visual regions captured meaningful temporal structure of the video stimuli, providing the necessary background for investigations into the processing of high-level dynamic concepts, such as event categories and actions. The use of video-computable DNN models allows the extraction of feature spaces at specific stages of video processing, inviting precise inquiry into an ROI's function in visual event understanding.

## BMD tracks the temporal dynamics unfolding within events

Does the temporally sluggish BOLD signal track temporal information within events? One possibility is that the BOLD signal only captures a global representation of the event that has no temporal structure itself, akin to a time-less semantic label (e.g., a person opens a door). Another possibility is that the BOLD signal captures delayed but temporally-resolved information, where different time points of the BOLD signal capture local snapshots of the changing event (Fig. 6a).

To test the latter hypothesis, we extracted two sets of activations from a DNN, one set using only the first second (first epoch) of the videos and the other set using only the last second (third epoch) of the videos (Fig. 6b). We utilized an encoding model procedure to measure the two sets' predictive performance at each of the first 9 seconds (corresponding to TRs in acquisition) (Fig. 6a). This was done with a feed-forward DNN trained on object categorization from still images to avoid any confounds from temporal integration. We expected that if a voxel captures snapshots of the changing event, the best prediction accuracy of the first video epoch encoding model is at least one TR (one TR = 1 s) earlier than the best prediction accuracy of the third video epoch encoding model. We used variance partitioning to identify the unique contribution of the first and third video epochs' predictions to the real fMRI responses (Fig. 6c) (for details, see Methods section Encoding and variance partitioning analysis procedure).

A whole-brain voxel-wise analysis (Fig. 6d) revealed the percentage of subjects at each voxel that show a significant 1–3 s delay between the best predicted time point (TR) using a video's first epoch and using a video's third epoch (only significant voxels are plotted). Results highlight significant temporal delays most pronounced in the early visual cortex. Equivalent ROI-based analysis (Fig. 6e upper ROI panels and main panel, see Supplementary Fig. 6 for all ROIs) yielded a similar result pattern. ROIs in the early and ventral visual brain (14 of the 22 total) showed a significant timing difference (black asterisks) between the time points at which fMRI responses are most related to the contents of the first and the third epoch of video with tighter confidence intervals in early visual regions (See Source Data).

Together, these results support the hypothesis that early and late TRs of the BOLD signal better encode temporally distinct early and late

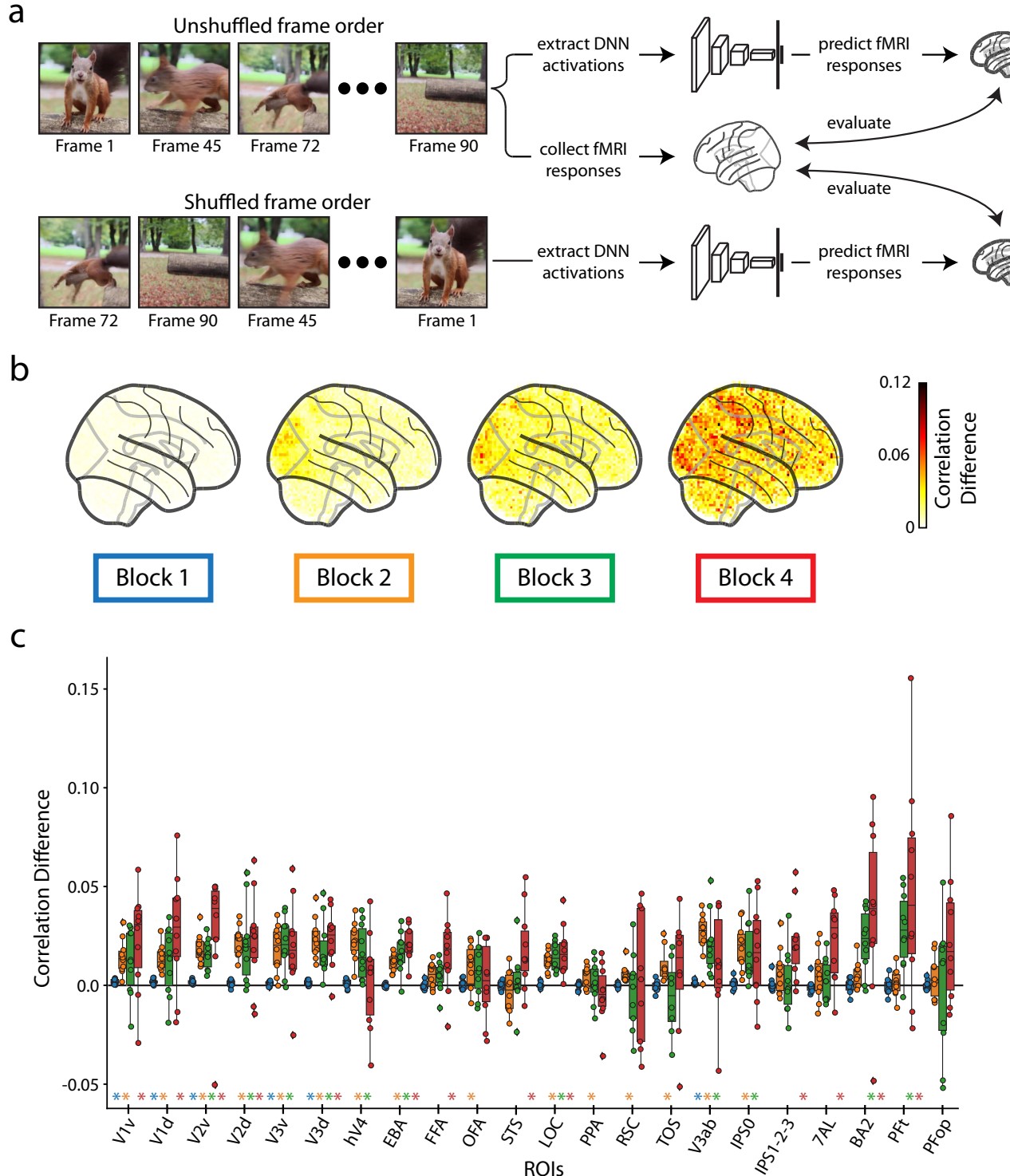

**Fig. 5 | Importance of temporal order in predicting fMRI responses. a** Frame shuffling procedure: We predict the real fMRI responses using both the DNN activations of the original (unshuffled) frame order and the DNN activations of the randomly shuffled frame order. A difference in prediction accuracy between the DNN activations of the unshuffled and shuffled frames indicates the preservation of correct temporal order in the fMRI response. Video frames are cropped from a representative video by NaIzletuSi ™ (YouTube: https://www.youtube.com/watch?v=QT0o0fTfZKE, CC BY 3.0 license: https://creativecommons.org/licenses/by/3.0/legalcode). **b** Whole-brain prediction difference: Difference in the correlation averaged over participants between the shuffled framed prediction accuracy and unshuffled frame prediction accuracy across the whole brain at different DNN layers (TSM model). **c** ROI-based prediction difference: Difference in the correlation between the shuffled frame prediction accuracy and unshuffled frame

prediction accuracy at different ROIs and DNN layers (TSM model). A colored asterisk along the x-axis indicates significant difference between the unshuffled and shuffled prediction accuracy at that DNN block ($p < 0.05$, one sample two-sided t-test against a population mean of 0, FDR correction across 22 ROIs x 4 blocks=88 comparisons). Source data are provided as a Source Data file. The box plot and asterisks are colored blue, orange, green, and red for blocks 1 – 4, respectively. The box plot encompasses the first and third data quartiles and the median (horizontal line). The whiskers extend to the minimum and maximum values within 1.5 times the interquartile range, and values falling outside that range are considered outliers (denoted by a diamond). The overlaid points show the value at each observation ($n = 10$ for all ROIs except transverse occipital sulcus (TOS, $n = 8$) and retrosplenial cortex (RSC, $n = 9$)).

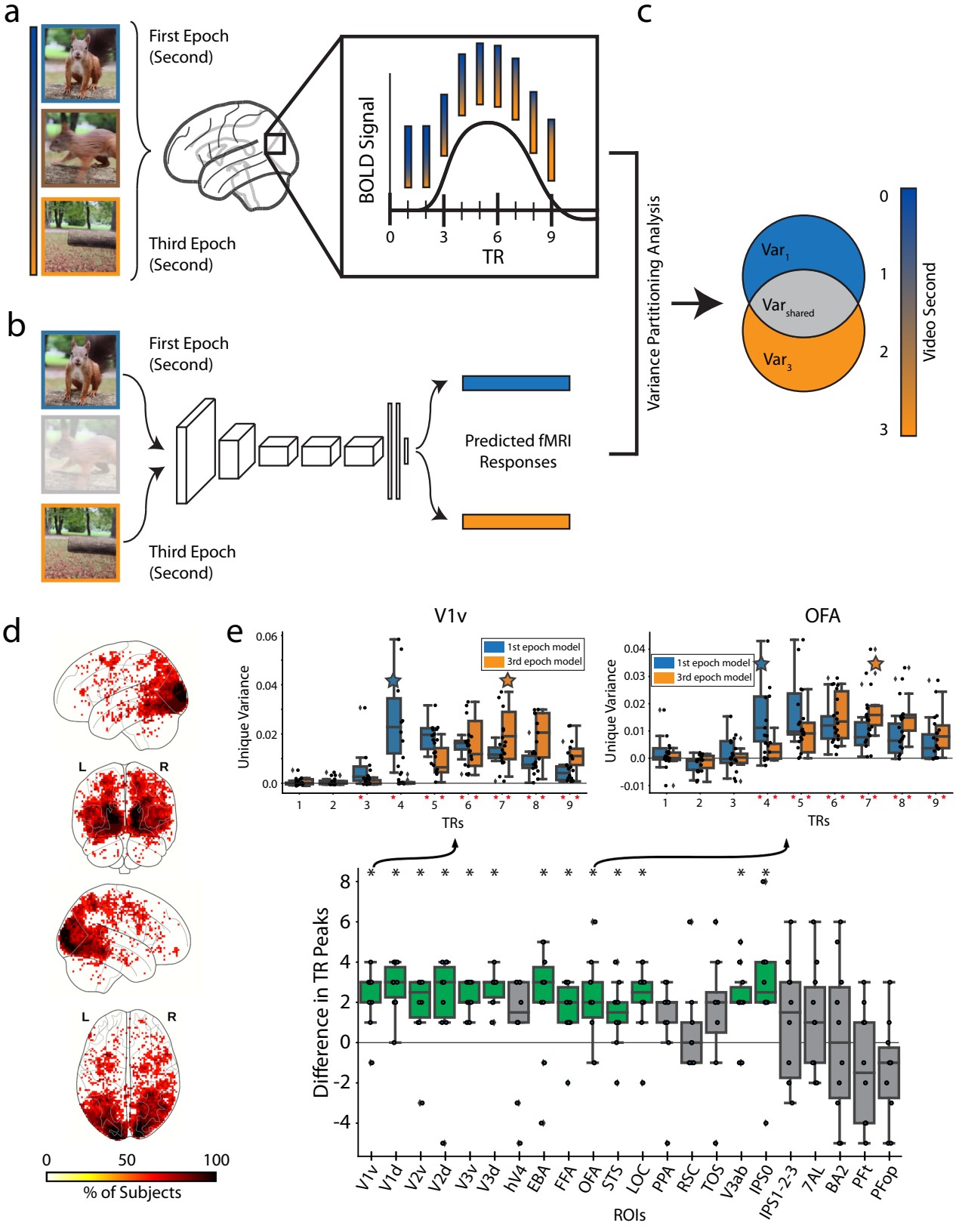

video snapshots respectively. Previous work showed that the content and sequence order of distinct images presented in under a second can be reliably decoded in the BOLD signal[35]. Here we additionally demonstrate that BMD can differentiate between two visually and conceptually similar snapshots of one second duration (i.e., first and third video second) separated by another highly similar snapshot of

one second duration (the second video second) with the most pronounced effects in the early visual cortex[7,111]. This invites future research to use BMD's temporally well-defined stimuli to explore how visual event information is integrated over shorter time periods, bridging an important gap to temporal integration studies of longform movies and BOLD encoding of rapidly presented stimuli.

**Fig. 6 | Encoding the temporal dynamics of the BOLD signal. a** TR estimated fMRI responses: We estimate the video-evoked brain response (beta values) of the first 9 TRs of the BOLD signal. Video frames are cropped from a representative video by NaIzletuSi ™ (YouTube: https://www.youtube.com/watch?v = QT0o0fTfZKE, CC BY 3.0 license: https://creativecommons.org/licenses/by/3.0/legalcode). **b** DNN predicted fMRI responses: We use an encoding model to extract DNN activations from the first video second (first epoch, blue) and third video second (third epoch, orange), then predict fMRI responses. Video frames are cropped from a representative video by NaIzletuSi ™ (YouTube: https://www.youtube.com/watch?v = QT0o0fTfZKE, CC BY 3.0 license: https://creativecommons.org/licenses/by/3.0/legalcode). **c** Variance partitioning analysis: We calculate the unique variance explained by the first and third video epochs' predicted fMRI responses at each TR. **d** Whole brain analysis: Each voxel shows the percentage of subjects with a TR peak difference of 1 to 3 TRs. Only significant voxels are plotted ($p < 0.05$, one-sided binomial test, FDR corrected). **e** ROI analysis: Upper ROI panels: We show the unique variance explained by the predicted fMRI responses of the first (blue) and third (orange) video epoch at TRs 1-9 in representative ROIs first visual area ventral (V1v) and occipital face area (OFA). Red asterisks indicate significance greater than 0 ($p < 0.05$, one-sample one-side t-test, FDR corrected across 9 TRs x 2 video epochs=18 comparisons). Blue/orange stars indicate the TR with the maximum subject averaged unique variance for the first/third video epochs, respectively. Source data are provided as a Source Data file. Main ROI Panel: We compute each subject's TR peak difference by subtracting the first video epoch's maximum TR from the third video epoch's maximum TR at each ROI. Green bars with asterisks indicate significant differences greater than 0 ($p < 0.05$, one-sample two-sided t-test). Source data are provided as a Source Data file. The box plot shows the first and third quartiles, median (horizontal line), and whiskers extending to 1.5 times the interquartile range. Outliers (diamonds) and individual observations are also shown ($n = 10$ for all ROIs except transverse occipital sulcus (TOS, $n = 8$) and retrosplenial cortex (RSC, $n = 9$)).

## Semantic metadata reveal strong similarity between sentence-level descriptions and visual brain activity

Varying levels of semantic information content, from static objects and scenes (e.g., duck, water) to temporal actions (e.g., swimming) to complex relations between parts (e.g., the duck is swimming on the water), can describe a visual event. It is unclear how these varying levels of complexity and content are reflected in ROIs while viewing visual events, especially given that regions throughout the ventral visual, dorsal visual, and parietal cortices have all been implicated in processing temporal aspects of videos[9,11,21,25,67] but also diverse feature preferences[21,25,57,73,112,113]. We leverage our metadata labels to construct five semantic video descriptions—objects, scenes, actions, objects+scenes+actions, and sentence text descriptions—to explore how semantics of varying information content and granularity are represented across brain regions during visual event perception. The word-level object, scene, and action labels provide low information descriptions of the video's static (object and scene) and temporal (action) content. The sentence text descriptions not only describe the video's object, scene, and temporal content but also how they relate. The combined objects+scenes+actions description concatenates the individual object, scene, and action label information to serve as an intermediate between the word-level and sentence-level descriptions (for details on the metadata collection, see Methods section Metadata).

We compute a neural representational dissimilarity matrix (RDM) at each voxel by computing the pairwise distances between vector embeddings obtained from a searchlight procedure (four voxel radius)[78] (Fig. 7a). We compute one RDM for each metadata category by calculating the pairwise distances between vector embeddings obtained by feeding the text-based labels through natural language processing models (FastText[114] for the single-word object, scene, and action labels, and Sentence-BERT[115] for sentence-level text descriptions) (Fig. 7b). We use representational similarity analysis (RSA)[29] to correlate the metadata representations (Fig. 7c) with neural representations to measure how similarly the different metadata descriptions are reflected in brain activity of dynamic videos (for details, see Methods section Metadata RSA analysis procedure).

Whole-brain results show the metadata labels differ in strength and pattern of similarity with brain activity (Fig. 7c). ROI results (Fig. 7d) compare the correlational strength between the five metadata descriptions at each ROI and depict a clear dominance of sentence text descriptions throughout cortex (purple bar). Taken individually, the object (green) and scene (red) description correlations are highest in their respective category-selective regions (LOC, EBA, FFA, OFA for object labels and PPA, RSC, and TOS for scene labels) as expected[112,113,116–118] (note that the object metadata can and often does describe people and animals present in the videos). Action labels (blue) correlate with ventral and dorsal visual regions[9,106] more strongly than parietal regions[119]. In all ROIs, the combined object+scene+action description (yellow) correlated with each ROI at a level between or equal to individual object, scene, and action descriptions and the text description.

Overall, the sentence text description results in stronger (or equally strong) correlation values than the other four semantic descriptions in all ROIs. Additionally, the three concatenated single-word labels (object+scene+action) results in stronger (or equally strong) correlation values than the individual single-word labels across all regions, even in category-selective regions. Both of these results are consistent with the idea that complex scene analysis, rather than simpler tasks such as object recognition, is the objective of the visual brain[120]. One might suspect that the category-selective ventral regions would best correlate with their respective metadata (e.g., PPA, RSC, and TOS for scene metadata), reasoning that the text description and object+scene+action labels, while including the pertinent category information, contain mostly irrelevant and distracting extra-category content[73,121] (see Ref. [122]).

Lastly, we assess if a representation of single frame text descriptions (generated by GIT[123]) would correlate just as strongly as a representation of our full video text descriptions (Supplementary Fig. 11). Although both sets of captions use sentences to describe the core elements of the video, the representation of the full video text descriptions correlates with the neural representations significantly better primarily throughout ventral visual cortex (V3v, hV4, EBA, FFA, OFA, STS, LOC, PPA, and V3ab). These results strongly suggest that BMD's brain responses are not only capturing dynamic information content, but also this information content is reflected in the full video text descriptions.

These analyses demonstrate that BMD can reveal visuo-semantic representations throughout cortex to better inform theories of the visual system's objective and action encoding. This methodology can also be extended to externally collected metadata to interrogate other facets of visual event understanding.

## Video memorability is reflected in high-level visual and parietal cortices

Some images and videos are inherently more memorable than others[17,41,124–126]. However, fMRI studies have mostly focused on the memorability of still images (but see[17]). Next, we evaluate the likely locus of video memorability in the brain to give insights into the relation between perception and memory[15,16].

Under the hypothesis that stimuli with higher memorability scores elicit a greater magnitude of brain response[30,31,127], we correlate a vector of video memorability scores with a vector of each voxel's corresponding brain responses (beta values) (Fig. 8a) (for details, see Methods section Memorability analysis procedure). Whole-brain (Fig. 8b) and ROI-based (Fig. 8c) analyses converged in revealing significant correlations in ventral visual, dorsal visual, and parietal cortex, all regions involved in video perception (green colored bars in Fig. 8c).

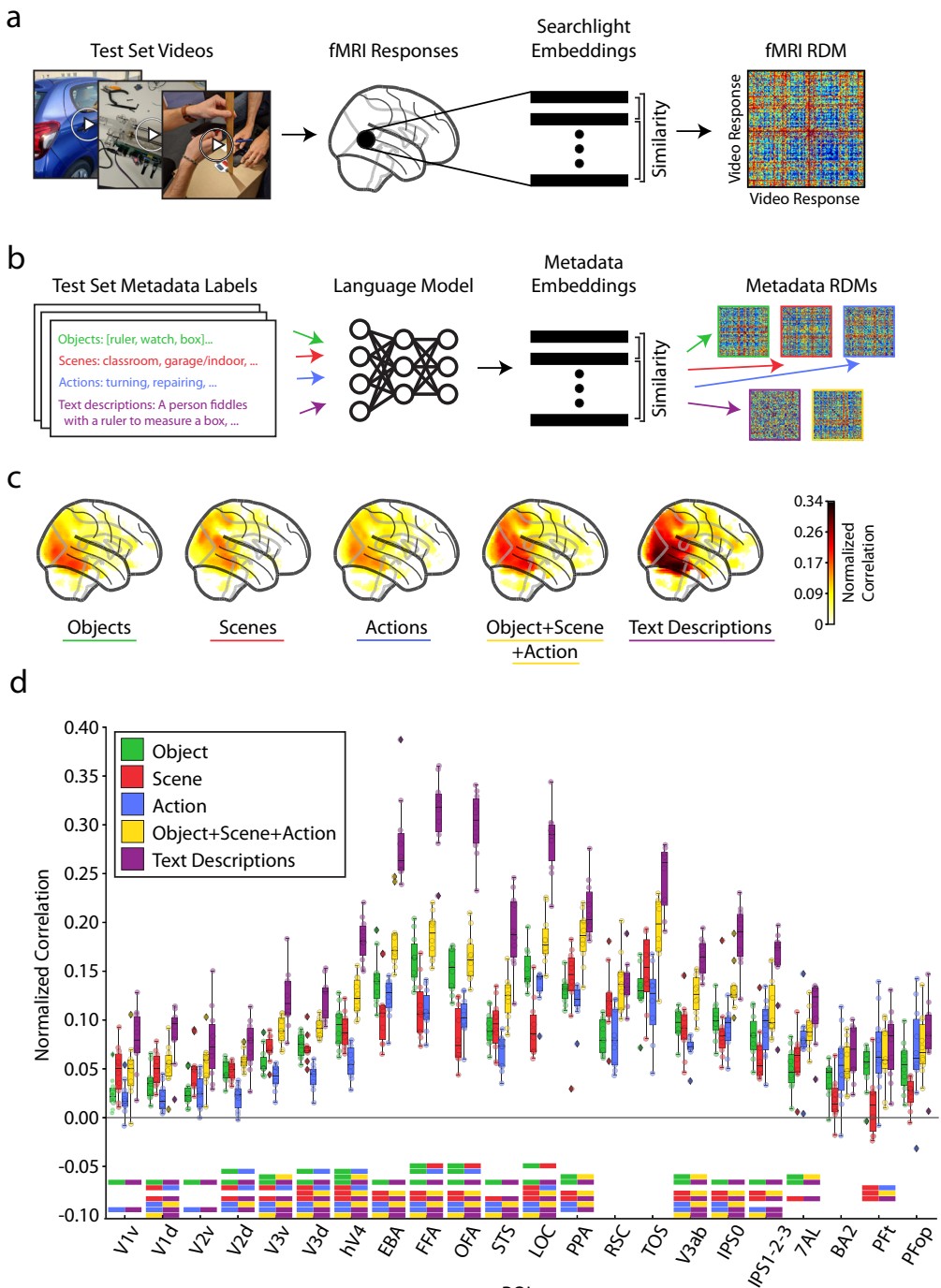

**Fig. 7 | Metadata-driven analysis of the semantics of visual event encoding.**
**a** Searchlight RDM methodology: The video-evoked brain responses to the 102 test set videos were extracted within a spherical searchlight to produce a vector embedding at each voxel. The correlation distance (1-Pearson's R) between the searchlight brain embeddings were computed to create a representational dissimilarity matrix (RDM) at each voxel in the brain. **b** RSA with metadata methodology: Video metadata were fed into a language model to produce a vector embedding. Similar to the searchlight RDM computation, the pairwise distance (cosine distance) between the metadata embeddings were calculated to create an RDM for the object (green), scene (red), action (blue), object+scene+action (yellow), and text description metadata (purple). **c** Whole-brain correlation of metadata RDMs with searchlight-based RDMs: The metadata RDM was correlated (Spearman's R) with each searchlight-based RDM at each voxel in each subject. After statistical analysis (one-sample, two-sided t-test against 0 correlation, FDR correction with q = 0.05), dividing each voxel by the subject's upper noise ceiling, and averaging across

subjects, the results were plotted in a glass brain to show each metadata's different pattern and magnitude of whole-brain responses. Only significant voxels are shown. **d** ROI-based correlation: The mean noise-normalized correlations are shown within each subject's ROI. At each ROI, a one-way ANOVA test compared the mean noise normalized correlation between the 5 conditions: objects (green), scenes (red), actions (blue), objects+scenes+actions (yellow), text descriptions (purple) (p < 0.05, Bonferroni corrected with n = 22 ROIs). If the ANOVA test was significant at an ROI, a Tukey's Honestly Significant Difference test determined pairwise significance (FWER = 0.05; significance of pairs denoted by the dual-colored bars under each ROI). Source data are provided as a Source Data file. The box plot encompasses the first and third data quartiles and the median (horizontal line). The whiskers extend to the minimum and maximum values within 1.5 times the interquartile range, and values falling outside that range are considered outliers (denoted by a diamond). The overlaid points show the value at each observation (n = 10 for all ROIs except transverse occipital sulcus (TOS, n = 8) and retrosplenial cortex (RSC, n = 9)).

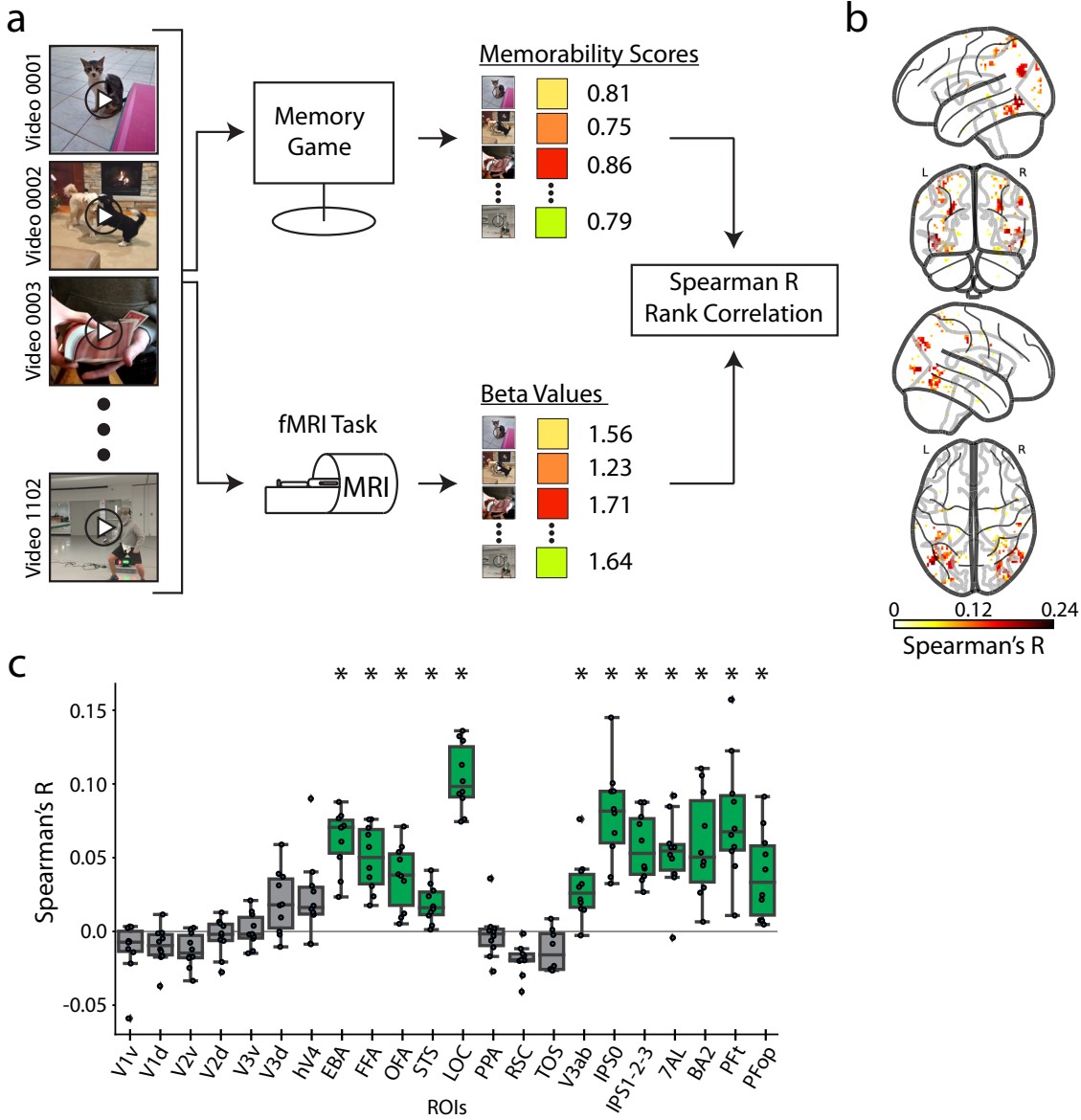

**Fig. 8 | Correlates of video memorability in cortex. a** Memorability analysis procedure: For every video, we collect a memorability score (values between 0-1, collected via a behavioral memory game in an independent study by Newman et al., 2020) and brain response (beta value averaged across repetitions). We then correlate (Spearman's R) the stimulus memorability scores with the stimulus-evoked beta values at each voxel to obtain a correlation coefficient. **b** Whole-brain memorability correlation results: We show significant whole brain voxel-wise correlations (Spearman's R) between brain activity and memorability scores. Significant voxels are determined by a t-test across subjects (one-sample, one-sided) and FDR correction (q = 0.05, positive correlation assumption). **c** ROI memorability correlation results: We show the Spearman correlation value between brain activity and memorability scores for each ROI. Significance is determined by a t-test against a null population mean correlation of 0 (p < 0.05, one-sample one-sided, Bonferroni corrected across n = 22 ROIs). Significant ROIs are denoted with a green box and black asterisk. Source data are provided as a Source Data file. The box plot encompasses the first and third data quartiles and the median (horizontal line). The whiskers extend to the minimum and maximum values within 1.5 times the inter-quartile range, and values falling outside that range are considered outliers (denoted by a diamond). The overlaid points show the value at each observation (n = 10 for all ROIs except transverse occipital sulcus (TOS, n = 8) and retrosplenial cortex (RSC, n = 9)).

This result contrasts with previous work using images, where the effects were largely relegated to the ventral visual cortex associated with image perception[30–32,127,128].

BMD allows the study of the neural correlates of memorability into the video domain, inviting further work into how visual perception and memory formation share computational resources[129–134] and bridging the study of static image memory with longform movie memory[18,135,136].

## Discussion
Disentangling the intricacies of human visual perception requires datasets varying in stimuli type[80,137,138], experimental design[37,88,139], and

neuroimaging acquisition efforts[38,39,61] (see Supplementary Table 1 for a detailed comparison between currently available large-scale visual fMRI datasets). Here, we contribute the BOLD Moments Dataset (BMD), a resource of human fMRI brain responses to 1102 annotated short videos to model human dynamic visual perception. The 3 second, naturalistic video stimuli balance the ecological validity of longform movies with the experimental control of still images to better model discrete events. BMD's ten subjects, high-repetition subset, and rich stimuli metadata accommodate uni- or multi-variate and single-subject or group-level modeling approaches, and its auxiliary resting state and dynamic functional localizer scans supplement insights into

cortical activity. Together, BMD invites a uniquely comprehensive set of analyses into human dynamic visual perception. BMD can help bridge the research communities of computer science and computational neuroscience and reveal how perceptual and cognitive brain systems extract information from complex visual inputs.

For example, we perturb various temporal properties of the videos and use DNNs to predict the corresponding brain activity. We observe how frame shuffling, by interrupting (at least) the video's temporal continuity and motion direction, affected a mix of early visual, ventral visual, and parietal cortex ROIs (Fig. 5). Comparing the brain prediction performance between the Temporal Shift Module (TSM) architecture and the Temporal Segment Network (TSN) architecture (Supplementary Fig. 9) isolates the effect of sharing information across frames and shows significant differences predominantly in early visual ROIs. Our encoding model optimized with a temporally-based objective function (action classification) demonstrates a correspondence in processing stages between video-computable DNN model blocks and cortical depth (Fig. 4), thereby both extending previous studies that analyzed responses to still images[82,83,95] and showing we can accurately predict brain activity in the dorsal visual and parietal regions that are largely driven by a stimulus's dynamic properties[21,55,140–142]. Critically, we demonstrate that the sluggish BOLD response tracks visual information over seconds (Fig. 6)[8,10,50,111,143,144]. These analyses consistently demonstrate BMD's temporal content and thus present a unique opportunity to leverage existing computational methods of social expressions[145–147], action recognition[43,148,149], integration of temporal features[100,150,151], and object detection[152,153] to study brain function.

We have shown how our provided metadata can link brain activity to meaningful and interpretable features of a visual event to yield theoretically relevant insights. For example, the higher correlations between brain activity and complex sentence descriptions over word-level labels (see Fig. 7 and Supplementary Fig. 11) suggest that the function of these brain regions extends beyond object recognition[120]. Additionally, the significant correlations between the crowd-sourced behavioral data on memorability and brain activity distributed in ventral visual, dorsal visual, and parietal regions (see Fig. 8) imply the memorability effect occurs in regions implicated in perception.

BMD's unique combination of stimuli type (short dynamic videos), experimental design (single-trial event related), and rich metadata make it well-suited for a diverse set of analyses extending beyond those demonstrated here, such as video reconstruction[87,154,155], video synthesis for cortical discovery[156,157], encoding and decoding models[72,158], spatio-temporal integration[7,145], and action observation networks[54,159–161]. BMD is thus well positioned to probe the functionally diverse and widespread cortical network recruited during dynamic visual perception.

## Methods
### Participants
Ten healthy volunteers (6 female, mean age ± SD = 27.01 ± 3.96 years, sex self-reported) with normal or corrected-to-normal vision participated in the experiment. All participants gave informed consent and were screened for MRI safety. They were compensated for their time at a rate of $90 USD per session (5 sessions total, each session lasted approximately 2.5 h). The experiment was conducted in accordance with the Declaration of Helsinki and approved by the local ethics committee (Institutional Review Board of Massachusetts Institute of Technology, approval code: 1510287948).

### Stimuli
The stimulus set consisted of 1102 videos in total. The videos were sampled from the Memento10k dataset[41], which is a subset of the Moments in Time dataset[42] and Multi-Moments in Time dataset[43]. Each video was square-cropped and resized to 268×268 pixels.

Videos had a duration of 3 s and frame rates ranging from 15 to 30 frames per second (mean = 28.3). The 1102 videos were manually selected from the Memento10k dataset by two human observers to encompass videos that contained movement (i.e., no static content), were filmed in a natural context, and represented a wide selection of possible events a human might witness. Additional criteria were to be free of post-processing effects, textual overlays, excessive camera movements, blur, and objectionable or inappropriate content.

The 1102 videos selected for the main experiment were split into a training and a testing set; 102 videos were chosen for the testing set, and the remaining 1000 videos formed the training set. Specifically, the testing set videos were chosen randomly from the 1102 videos, and then checked manually to ensure no semantic overlap, in terms of objects plus actions, occurred between any pair of testing set videos. If semantic overlap was found between a pair of videos as determined by an author, one of these videos was swapped with a video randomly selected from the pool of remaining videos and incorporated into the testing set. This was repeated until a semantically-diverse testing set was formed. The training and testing sets are only intended to differ in the number of repetitions shown to the participant. In this way, the BMD dataset contains a low repetition (3 repetitions) training set of 1000 videos and a high repetition (10 repetitions) testing set of 102 videos to facilitate analyses dependent on either large number of stimuli or large number of repetitions. The training and testing sets are additionally intended to mirror training and testing sets common in machine learning applications, reflecting its potential use for model building and evaluation.

### (f)MRI experimental design
The fMRI data collection procedure was as follows: subjects completed a total of 5 separate fMRI sessions on separate days. Session 1 consisted of structural scans, functional localizer runs, and functional resting state scans all interspersed (Fig. 1b). Sessions 2–5 consisted of the main functional experimental runs where the subjects viewed the training and testing set videos (Fig. 1c). Throughout the whole experiment for a given subject, each training set video was shown a total of 3 times and each testing set video was shown a total of 10 times, resulting in 3000 and 1020 trials in training and testing sets, respectively. We organized trials into experimental runs such that they either only contained testing set videos (test runs) or only contained training set videos (training runs). Stimuli were presented using Psychtoolbox-3 (http://psychtoolbox.org/) with MATLAB version 2017.

### Session 1
**Functional localizers.** Subjects completed five functional localizer runs (Fig. 1b). Subjects freely viewed colored, naturalistic videos (18 s length, composed of 6 3-second videos) corresponding to one of five categories (faces, bodies, scenes, objects, and scrambled objects) in order to functionally localize each subject's category selective regions[13,57,73,162]. Subjects viewed the videos freely and performed a one-back vigilance task to ensure attention to the task (videos are available with the dataset release). Each stimulus category included 48 individual stimuli. Each run consisted of 5 blocks of fixation baseline (null) and 20 blocks of stimulus presentation for a total of 25 blocks. Baseline blocks occurred every sixth block, with 5 stimulus blocks of each category presented in between baseline blocks in a randomized order. The duration of each video was 3 seconds, with each block lasting 18 seconds. Each category stimulus block included 5 unique videos chosen randomly from the 48 stimuli plus one one-back stimulus repetition. Subject accuracy on the one-back task was 0.941 ± 0.011 (mean ± SD). Each run began and ended with an 18 second fixation (null) block. The end of the run contained an additional 19 seconds of gray screen (not modeled in the GLM) for a total duration of 268 volumes.

**Resting state.** Resting state data was obtained across 5 runs (Fig. 1b). Within each run, participants were instructed to keep their eyes closed, to not think of anything specific, but to remain awake. The duration of resting state runs is 212 volumes.

**Session 2–5.** Each of the 4 sessions after session 1 had identical structure. Since fixation against a dynamic video background is difficult, each session began with a 2.5 minute fixation training outside the scanner to provide the subjects with real-time feedback of any eye movements, voluntary or involuntary[163]. In this fixation training, subjects viewed a black-and-white random dot display flickering in counter phase. With fixation, this display evoked the illusion of a uniform gray display. Breaking fixation with eye movements disrupts this illusion, and a vivid black-and-white random dot display is perceived. Subjects were instructed to keep their eyes fixated as to minimize the disruption of the illusion.

**Main experiment:.** Inside the scanner, videos were presented at the center of the screen subtending 5 degrees of visual angle and overlaid with a central red fixation cross (0.52 degrees of visual angle). Subjects were instructed to focus on the fixation cross for the duration of the main experiment. The experiment consisted of both video presentation trials (occurring 75% of the time) and null trials (occurring 25% of the time). Both trial types were 4 s long. The video presentation trial consisted of a 3 s video presentation followed by a 1 s intertrial interval. The videos were presented in random order with the constraint of no consecutive repetition. The null trial consisted of the presentation of a gray screen. During the null trial the fixation cross turned darker for 1000 milliseconds and participants reported the change with a button press. Participant accuracy in this task was $0.964 \pm 0.014$ (mean ± SD).

The testing and training set videos were presented within testing and training runs, respectively, where each test run consisted of 113 trials and each train run consisted of 100 trials. The order of these runs was randomized except for the constraint that two testing runs cannot be shown back-to-back in the same session in order to reduce potential memory effects caused by the same stimulus being presented within a short period of time. Each session contained 3 test runs and 10 training runs and lasted approximately 100 minutes. Naturally, some testing set videos were repeated within a session (255 testing video presentations for only 102 possible testing videos), but no training set videos were repeated within a session. Each run began with 4 seconds of fixation and ended with 13 seconds (testing runs) or 12.5 seconds (training runs) of fixation. The 0.5 second difference of the duration of the ending fixation period between the testing and training runs is due to their different number of trials (113 and 100) and the acquisition TR of 1.75 seconds not being an even factor the 4 second trial length. In total, testing and training runs consisted of 268 and 238 volumes, respectively.

## Metadata

Visual events consist of complex combinations of objects, locations, actions, and more. To capture the many dimensions of visual events, we characterized each video with a set of seven metadata categories: object labels, scene labels, action labels, text descriptions, a spoken transcription, a memorability score, and an index of memorability decay rate (Fig. 1a). Five object, scene, action, and text description labels were collected for each stimulus to ensure comprehensive coverage (the five annotators' labels reflect their unique interpretations) and form a group consensus (the five annotators' labels can be used to converge on a single label). Labels for each metadata category were collected in different human crowd-sourced experiments. An annotator was allowed to label up to 20 different videos but no more than one label per video to encourage a diverse sampling of human annotators within videos and throughout the stimulus set. While crowd-source workers were not restricted from participating in multiple metadata experiments, this was unlikely due to the experiments being collected at different times and the large population pool from which the crowd-source workers were drawn.

**Object labels.** For each video, we obtained at least 5 sets of up to three different object labels in a human crowd-sourced experiment on Prolific. Each annotator was instructed to select up to three different object labels visible in the video. They selected at least one object label, and if they believed no more objects were present in the video, they were allowed to select, up to two times, an option labeled No more objects in the video. This option thus encouraged accurate labels and carried information on the density of objects in the video. Each object label was one of 1854 possible labels from the THINGS dataset[61] to encourage overlap with computational neuroscience work and leverage the additional THINGS metadata on each label (e.g., animacy, size, indoor/outdoor). The object label selections can be different or the same across annotators. One author manually reviewed the labels to ensure the labels assigned to the video were sensical (i.e., participants were not choosing labels at random).

**Scene labels.** For each video, we obtained at least 5 scene labels using a human crowd-sourced experiment on Prolific. Each of the five different annotators were instructed to select a scene label that best describes the scene of the video. All scene labels came from the Places365 dataset[60] for its broad scene coverage and overlap with computer vision resources. The scene label selections can be different or the same across videos. One author manually reviewed the labels to ensure the labels assigned to the video were sensical (i.e., participants were not choosing labels at random).

**Action labels.** The 5 action labels were selected by workers on the crowd-sourcing platform Prolific. We restricted the possible action labels to be from one of 292 possible action labels that broadly encompass meaningful human actions[43]. The participant viewed these 292 possible action labels, watched the video, and selected one action label that best described the video. Each of the 5 action labels per video were produced by different participants. Given that an action, by definition, unfolds across time and the BMD stimuli have a short 3 second duration, the majority of the videos contain one primary action. Additionally, the limited number of stimulus repetitions and the central fixation further limits the fMRI scanner participant's ability to perceive a video's small, inconsequential actions if they are present. In the edge case that a complex video captures multiple salient actions, the five annotations can capture these multiple actions. Note that annotators labeled up to 3 objects in a video (described above) because objects, unlike actions, have clear visual boundaries, often occur in multiple instances in a video, and have no temporal dimension. Two authors manually reviewed the labels to ensure the labels assigned to the video were sensical (i.e., participants were not choosing labels at random).

**Text descriptions.** We obtained five high-level descriptions of the content of each video in the form of written descriptions. The five text descriptions were human-generated from participants on the crowd-sourcing platform Amazon Mechanical Turk (AMT). Their task was to watch the video and type a 10–15 word caption in complete English sentences. This instruction was not used as acceptance criteria. Each of the 5 text descriptions were produced by a different human participant, and each participant was allowed to annotate multiple videos. The authors manually checked the text descriptions to ensure the text descriptions pertained to the video (i.e., participants followed instructions) and to correct obvious typos.

**Spoken transcriptions.** We obtained one spoken description per video to capture emotional and descriptive nuances typically conveyed in speech but not present in typing. The spoken description was

collected via human participants on AMT, as in Monfort and colleagues[62]. Participants were instructed to watch the video and verbally describe it. They were given no instructions pertaining to the length of the description. We record the audio file and use Google's speech-to-text transcription to generate a text transcription. The transcription was manually checked to ensure it pertained to the video (i.e., participants followed instructions) and to correct obvious typos. We release the text transcription but not the original audio file for privacy purposes.

**Memorability score and decay rate.** The memorability score and decay rate were measured by Newman and colleagues[41], where AMT human participants played a video memory game. The game consisted of a continuous video stream where the participant pressed the spacebar upon seeing a repeated video. Repeated videos were presented at various delays, from 30 seconds to ten minutes. The participant's responses were then used to calculate a video's memorability score from 0 (no recall) to 1 (perfect recall) and memorability decay rate from 0 (no decay) to -inf (instantaneous decay). Specifically, a video's memorability score is the fraction of correct identifications in the memory game normalized to a lag of 80 videos in between consecutive presentations. A video's memorability decay rate describes how a video's memorability score changes over different lags. The memorability decay rate was regressed from the output of SemanticMemNet[41], an Inflated 3D (I3D) Convolutional Neural Network trained to predict a video's memorability score and text caption from the video frames and optical flow. A video's memorability score (m) and decay rate ($\alpha$) can be used to predict its memorability at any time lag t (the number of videos in between the first and second presentation) according to the following equation[41]:

$$m_t = m_T + \alpha(t - T) \qquad (2)$$

where T is a set lag of 80. The memorability scores given in the stimuli metadata were computed at a lag of t = 80.

Note that Newman and colleagues[41] experimentally observed memorability decay rates above 0. Since a positive memorability decay rate may reflect a currently unknown stimulus or cognitive feature, we preserve them in BMD's stimuli metadata. One may wish to clip them to a maximum value of 0 depending on the analysis.

**Motion energy features.** We provide video-computable motion features of each video using a motion energy model[87,164,165]. The motion energy model uses a set of spatial and temporal Gabor filters to extract a video's motion and direction. We use these features to predict brain activity in motion-selective regions of MT[166,167], hV4[168,169], V3AB[21,170], and IPSO[21]. This analysis uses version B of the dataset and is detailed in Supplementary section Motion energy features computation and encoding model.

### fMRI data preprocessing and analysis
**fMRI data acquisition.** The MRI data were acquired with a 3 T Trio Siemens scanner using a 32-channel head coil. During the experimental runs, T2*-weighted gradient-echo echo-planar images (EPI) were collected (TR = 1750 ms, TE = 30 ms, flip angle = 71°, FOV read = 190 mm, FOV phase = 100%, bandwidth = 2268 Hz/Px, resolution = 2.5 × 2.5 × 2.5 mm, slice gap = 10%, slices = 54, multi-band acceleration factor = 2, ascending interleaved acquisition). Additionally, a T1-weighted image (TR = 1900 ms, TE = 2.52 ms, flip angle = 9°, FOV read = 256 mm, FOV phase = 100%, bandwidth = 170 Hz/px, resolution = 1.0 × 1.0 × 1.0 mm, slices = 176 sagittal slices, multi-slice mode = single shot, ascending) and T2-weighted image (TR = 7970 ms, TE = 120 ms, flip angle = 90°, FOV read = 256 mm, FOV phase = 100%, bandwidth = 362 Hz/Px, resolution = 1.0 × 1.0 × 1.1 mm, slice gap = 10%, slices = 128, multi-slice mode = interleaved,

ascending) were obtained as high-resolution anatomical references. We acquired resting state and functional localizer data using acquisition parameters identical to the main experimental runs. Dual echo fieldmaps (TR = 636 ms, TE1 = 5.72 ms, TE2 = 8.18 ms, flip angle = 60°, FOV read = 190 mm, FOV phase = 100%, bandwidth = 260 Hz/Px, resolution = 2.5 × 2.5 × 2.5 mm, slice gap = 10%, slices = 54, ascending interleaved acquisition) were acquired at the beginning of every session to post-hoc correct for spatial distortion of functional scans induced by magnetic field inhomogeneities.

**Preprocessing.** All MRI data was first converted to BIDS format[63], and the T1-weighted structural scans were anonymized using PyDeface (https://github.com/poldracklab/pydeface) before public release. All data from all sessions was then preprocessed using the standardized fMRIPrep preprocessing pipeline (version 20.2.1). Preprocessing and analysis used both the SPM12 toolkit (https://www.fil.ion.ucl.ac.uk/spm/software/spm12/) in MATLAB (version 2017) and Python3 (https://www.python.org/). For all results unless stated otherwise, we use the standard MNI152NLin2009cAsym volumetric space for its frequent use in other work and preservation of natural left-right asymmetries in the brain. As recommended by fMRIPrep to increase transparency and reproducibility in MRI preprocessing, we copy their generated preprocessing text in its entirety below:

Results included in this manuscript come from preprocessing performed using *fMRIPrep* 20.2.1 ([64,171]; RRID:SCR_016216), which is based on *Nipype* 1.5.1 ([172,173] RRID:SCR_002502).

**Anatomical data preprocessing.** A total of 1 T1-weighted (T1w) images were found within the input BIDS dataset. The T1-weighted (T1w) image was corrected for intensity non-uniformity (INU) with N4BiasFieldCorrection[174], distributed with ANTs 2.3.3 ([175]; RRID:SCR_004757), and used as T1w-reference throughout the workflow. The T1w-reference was then skull-stripped with a *Nipype* implementation of the antsBrainExtraction.sh workflow (from ANTs), using OASIS30ANTs as target template. Brain tissue segmentation of cerebrospinal fluid (CSF), white-matter (WM) and gray-matter (GM) was performed on the brain-extracted T1w using fast (FSL 5.0.9, RRID:SCR_002823[176],). Brain surfaces were reconstructed using recon-all (FreeSurfer 6.0.1, RRID:SCR_001847[177],), and the brain mask estimated previously was refined with a custom variation of the method to reconcile ANTs-derived and FreeSurfer-derived segmentations of the cortical gray-matter of Mindboggle (RRID:SCR_002438[178],). Volume-based spatial normalization to one standard space (MNI152NLin2009cAsym) was performed through nonlinear registration with antsRegistration (ANTs 2.3.3), using brain-extracted versions of both T1w reference and the T1w template. The following template was selected for spatial normalization: *ICBM 152 Nonlinear Asymmetrical template version 2009c* [[179], RRID:SCR_008796; TemplateFlow ID: MNI152NLin2009cAsym].

**Functional data preprocessing.** For each of the 62 BOLD runs found per subject (across all tasks and sessions), the following preprocessing was performed. First, a reference volume and its skull-stripped version were generated using a custom methodology of *fMRIPrep*. A B0-nonuniformity map (or *fieldmap*) was estimated based on a phase-difference map calculated with a dual-echo GRE (gradient-recall echo) sequence, processed with a custom workflow of *SDCFlows* inspired by the epidewarp.fsl script and further improvements in HCP Pipelines[180]. The *fieldmap* was then co-registered to the target EPI (echo-planar imaging) reference run and converted to a displacements field map (amenable to registration tools such as ANTs) with FSL's fugue and other *SDCflows* tools. Based on the estimated susceptibility distortion, a corrected EPI (echo-planar imaging) reference was calculated for a more accurate co-registration with the anatomical reference. The BOLD reference was then co-registered to the T1w reference using

bbregister (FreeSurfer) which implements boundary-based registration[181]. Co-registration was configured with six degrees of freedom. Head-motion parameters with respect to the BOLD reference (transformation matrices, and six corresponding rotation and translation parameters) are estimated before any spatiotemporal filtering using mcflirt (FSL 5.0.9[182],). BOLD runs were slice-time corrected using 3dTshift from AFNI 20160207 ([183], RRID:SCR_005927). The BOLD time-series (including slice-timing correction when applied) were resampled onto their original, native space by applying a single, composite transform to correct for head-motion and susceptibility distortions. These resampled BOLD time-series will be referred to as *preprocessed BOLD in original space*, or just *preprocessed BOLD*. The BOLD time-series were resampled into standard space, generating a *preprocessed BOLD run in MNI152NLin2009cAsym space*. First, a reference volume and its skull-stripped version were generated using a custom methodology of *fMRIPrep*. Several confounding time-series were calculated based on the *preprocessed BOLD*: framewise displacement (FD), DVARS and three region-wise global signals. FD was computed using two formulations following Power (absolute sum of relative motions[184],) and Jenkinson (relative root mean square displacement between affines[182],). FD and DVARS are calculated for each functional run, both using their implementations in *Nipype* (following the definitions by[184]). The three global signals are extracted within the CSF, the WM, and the whole-brain masks. Additionally, a set of physiological regressors were extracted to allow for component-based noise correction (*CompCor*[185],). Principal components are estimated after high-pass filtering the *preprocessed BOLD* time-series (using a discrete cosine filter with 128 s cut-off) for the two *CompCor* variants: temporal (tCompCor) and anatomical (aCompCor). tCompCor components are then calculated from the top 2% variable voxels within the brain mask. For aCompCor, three probabilistic masks (CSF, WM and combined CSF + WM) are generated in anatomical space. The implementation differs from that of Behzadi et al. in that instead of eroding the masks by 2 pixels on BOLD space, the aCompCor masks are subtracted a mask of pixels that likely contain a volume fraction of GM. This mask is obtained by dilating a GM mask extracted from the FreeSurfer's *aseg* segmentation, and it ensures components are not extracted from voxels containing a minimal fraction of GM. Finally, these masks are resampled into BOLD space and binarized by thresholding at 0.99 (as in the original implementation). Components are also calculated separately within the WM and CSF masks. For each CompCor decomposition, the *k* components with the largest singular values are retained, such that the retained components' time series are sufficient to explain 50 percent of variance across the nuisance mask (CSF, WM, combined, or temporal). The remaining components are dropped from consideration. The head-motion estimates calculated in the correction step were also placed within the corresponding confounds file. The confound time series derived from head motion estimates and global signals were expanded with the inclusion of temporal derivatives and quadratic terms for each[186]. Frames that exceeded a threshold of 0.5 mm FD or 1.5 standardized DVARS were annotated as motion outliers. All resamplings can be performed with *a single interpolation step* by composing all the pertinent transformations (i.e., head-motion transform matrices, susceptibility distortion correction when available, and co-registrations to anatomical and output spaces). Gridded (volumetric) resamplings were performed using antsApplyTransforms (ANTs), configured with Lanczos interpolation to minimize the smoothing effects of other kernels[187]. Non-gridded (surface) resamplings were performed using mri_vol2surf (FreeSurfer).

Many internal operations of *fMRIPrep* use *Nilearn* 0.6.2 ([188], RRID:SCR_001362), mostly within the functional processing workflow. For more details of the pipeline, see the section corresponding to workflows in fMRIPrep's documentation.

Next, the fMRI data from the main experimental runs (sessions 2-5) underwent temporal resampling. In detail, we resampled each voxel's time series using cubic interpolation to change the acquisition TR of 1.75 second to a new TR of 1 s. The 1.75 s acquisition TR combined with a 4 second trial length allowed for a dense sampling of the BOLD response relative to stimulus onset and thereby a good estimate of the BOLD response shape. However, as on every trial the timing of MR acquisition with respect to the trial was different, analysis of the BOLD response time-locked to the onset of each trial was cumbersome. The interpolation to a 1 s TR achieved a time series with regular sampling of the BOLD response relative to stimulus onset, enabling analysis time-locked to the onset of the videos.

Functional localizer scans were subsequently smoothed with a 9 mm full width half maximum of the Gaussian kernel. The main experimental functional runs use unsmoothed data.

### General linear model

**Functional localizer**. To model the hemodynamic response to the localizer videos, the preprocessed fMRI data, video, and fixation baseline onsets and durations were included in a general linear model (GLM). The fixation and five category (faces, objects, scenes, bodies, and scrambled objects) blocks were included as regressors of interest. Motion and run regressors were included as regressors of no interest. All regressors were convolved with a hemodynamic response function (canonical HRF) to calculate beta estimates.

**Main experiment**. We modeled the BOLD-signal of each voxel in the preprocessed fMRI data of each participant as a weighted combination of simple Finite Impulse Response (FIR) basis functions. We modeled the BOLD response with respect to each video onset from 1 to 9 s in 1 s steps (corresponding to the resolution of the resampled time series). The stimulus presentation trials, but not the null response trials, were included as regressors of interest. Within this time interval the voxel-wise time course of activation was high-pass filtered (removing signal with f < 1/128 Hz) and serial correlations due to aliased biorhythms or unmodelled neuronal activity were accounted for using an auto-regressive AR(1) model.

Using FIR in the way described above, we modeled every trial in the experimental run of each session, simultaneously capturing the spatial variability and the temporal evolution of brain responses underlying visual event understanding. For every session we generated separate FIR models for training and testing sets. Overall, for each video condition in the testing set we extracted 10 (repetitions) x 9 (seconds) beta value estimates, and for each video condition in the training set we extracted 3 (repetitions) x 9 (seconds) beta value estimates.

### ROI definitions of early visual cortex and ventral visual stream

We computed five t-contrasts (FWE corrected at p = 0.05) per subject based on the beta values from the localizer experiment to quantify category-specific voxel activations. The t-contrasts were used to localize voxels in early visual regions (objects & scrambled objects > baseline; V1v, V1d, V2v, V2d, V3v, V3d, hV4), a body-selective region (bodies > objects; EBA), an object-selective region (objects > scrambled objects; LOC), face-selective regions (faces > objects; FFA, OFA, STS), and scene-selective regions (scenes > objects; PPA, RSC, TOS). Together, these 15 ROIs cover brain regions along the ventral visual pathway thought to transform low-level visual features into complex, semantic representations useful for object recognition.

The ROI masks for each subject were defined using the subject's t-contrast maps. Voxels outside the anatomical region of interest were manually set to 0, and ROIs were limited to 1000 voxels in size. In the event that two or more ROIs overlapped, we used a probability map computed from all ten subjects' t-contrasts to assign each overlapping voxel to the ROI it most likely belonged to. In detail:

We first created a probabilistic map for each of the 8 category-selective and 7 early visual ROIs, using data from all 10 subjects. Each

subject's five t-contrast maps from the localizer experiment were family-wise error (FWE) corrected at p = 0.05. We then added each subject's binarized version of the FWE corrected t-contrast maps (1 if the voxel passed FWE correction, 0 if not) and divided by the total number of subjects. This resulted in 5 probabilistic maps corresponding to each of the 5 t-contrasts, where each voxel had a value from 0 to 1 in steps of 0.1 (0, 0.1, 0.2, …, 1.0) representing the decimal percentage of the number of subjects where that voxel was significant. We defined a probabilistic map for each of the 7 early visual ROIs by masking the t-contrast probability map (objects & scrambled objects > baseline) with the early visual masks defined by Wang and colleagues[189]. We used the remaining 4 category-selective t-contrast probabilistic maps to define a probabilistic map for each of the 8 category-selective ROIs by visually inspecting the appropriate t-contrast probability map (e.g., inspecting the scenes > objects t-contrast probability map to define the scene-selective ROIs) and manually setting the voxels clearly outside the region of interest to 0. This way we defined a total of 15 separate probabilistic maps.

Next, we defined 15 ROI masks for each subject. For each subject, we masked their own appropriate FWE corrected t-contrast map with the corresponding binarized t-contrast probability map. The ROIs were limited to the top 1000 voxels. This process resulted in fifteen subject-specific ROI masks.

Lastly, we assigned any overlapping voxels to a single ROI. In the event two or more voxels overlapped, we assigned the overlapping voxels to a single ROI based on the ROI probability map in step 1. Exemplified for two ROIs A and B, if ROI A and ROI B overlapped on voxel x, we indexed into ROI A's probability map at voxel x and ROI B's probability map at voxel x. We assigned voxel x to the ROI with the higher probability. If ROI A and B have equal probabilities at voxel x, we grew a patch by one voxel along each dimension and compared the ROIs' mean probabilities within that patch. This process was repeated until there was no longer a tie between ROIs and each ROI was non-overlapping.

Subject 6 did not show any responses in TOS and RSC, and subject 7 did not show any responses in RSC. The ROI masks were used in subsequent analyses to extract active voxels within a subject's ROI during the main experiment.

### ROI definitions of dorsal visual stream
We additionally defined dorsal regions of interest using anatomical landmarks, since our functional localizer was designed to define early visual and ventral category-selective regions only. All dorsal regions were defined the same way for all subjects. Specifically, we use the maximum probability map in Wang and colleagues[189] to anatomically define area V3ab (grouping areas V3a and V3b), IPS0, and IPS1-2-3 (grouping areas IPS1, IPS2, and IPS3). We additionally define 7AL, BA2, PFt, and PFop in more superior regions of the dorsal stream using the atlas described in Glasser and colleagues[190]. Atlas versions registered in MNI volume space were downloaded and resampled to BMD's functional voxel resolution with nearest neighbor interpolation using SPM 12's reslice function. The resampled atlases were then used to index ROIs in the BMD volume data.

### Univariate split-half reliability analysis
To select voxels with high signal-to-noise ratio, we defined a selection criteria based on split-half trial reliability[74,191]. Our assumption behind the criteria was that the voxel responses on different trials corresponding to the same video should be more correlated with each other than voxel responses to different videos. As the voxel response to a video, we used the beta estimates from the FIR model averaged over time points TR 5-9 (representing the peak of a typical BOLD signal). We then z-scored the voxel responses across the videos.

We divided the voxel responses of the testing set stimuli trials (n = 10 trials) into two equal splits and calculated the Pearson

correlation (ρ) between the splits. The split-half reliability was calculated using the Spearman-Brown formula (Eq. 1), where the maximum reliability is 1. We calculated the split-half reliability for all possible combinations of splits and used the mean reliability as the reliability for that voxel.

To assess if reliability is better than chance, we first estimated chance-level reliability. For each voxel, we calculated the split-half reliability for all possible combinations of splits while randomly permuting the video indices for one of the two splits. This process was repeated 100 times with a different video index permutation each time. This procedure resulted in 100 random reliability values for each voxel, which was used to calculate a p-value. The voxels that satisfy our reliability criteria (p < 0.05) are referred to as reliable voxels. Due to the testing set's high number of repetitions, the reliable voxels were defined using data from the testing set runs. See Supplementary Fig. 4 for each subject's number of reliable voxels at each ROI, and see Supplementary Fig. 5 for each subject's mean reliability at each ROI.

### Multivariate searchlight-based reliability analysis
We computed the upper and lower noise ceilings at each voxel in the whole brain using the testing set and present subject-specific and subject-averaged results (Supplementary Fig. 3). For each subject separately, the raw FIR beta estimates to the video stimuli at each voxel were z-scored across stimuli, averaged across TRs 5-9, and averaged across stimuli repetitions to result in a (n_stimuli x 1) vector of beta values. A spherical searchlight with radius 4 voxels was defined and centered on a voxel $v$. The (n_stimuli x 1) vector of beta values over all voxels contained within the searchlight sphere compose a (n_stimuli x n_voxels) matrix. The 1-Pearson correlation of the matrix resulted in a single Representational Dissimilarity Matrix (RDM) of size (n_stimuli x n_stimuli) for the voxel $v$. The searchlight then centered on the next voxel, and the procedure was repeated until an RDM was calculated for each voxel. This procedure was repeated for each of the 10 subjects.

The upper and lower noise ceilings of the multivariate searchlight results were computed[192,193]. To compute the upper noise ceiling at voxel $v$, the voxel's searchlight-computed RDM from one subject was correlated (Spearman's R) with the 10-subject group averaged RDM. The average of each subject's correlation to the 10-subject group averaged RDM is the upper noise ceiling for voxel $v$. This process was repeated over all voxels to result in the upper noise ceiling values throughout the whole brain displayed in Fig. 3b. The upper noise ceiling estimates the highest correlation that a model can be expected to obtain given the noise in the data.

To compute the lower noise ceiling at voxel $v$, the voxel's searchlight-computed RDM from one left-out subject was correlated (Spearman's R) with the remaining 9-subject group averaged RDM. The average of each left-out subject's correlation to the corresponding 9-subject group averaged RDM is the lower noise ceiling for voxel $v$. This process was repeated over all voxels to estimate the lower noise ceiling values throughout the whole brain.

### Action recognition TSM ResNet50 model training
The model adopts the architecture of a Temporal Shift Module (TSM)[96], with ResNet50 as the backbone network. We trained our model on the M4 (Multi-Moments minus Memento) training dataset for 120 epochs by using LSEP (log-sum-exp pairwise) loss[43]. LSEP loss was first proposed in[194] and modified in[43] as an appropriate loss function to train on multi-label and class imbalanced datasets, such as actions. The M4 training dataset consists of 1012,169 videos which are in the Multi-Moments in Time dataset but not in the Memento dataset to ensure no overlap with the 1102 BMD stimuli. Our model was initialized with the weights of the ResNet50 trained on ImageNet-1k dataset. We chose the model hyperparameters to closely follow those used in Lin and colleagues[96]. Specifically, during the training phase, our model split the input video into 8 segments and sampled 1 frame from

each segment. We used SGD optimizer to optimize our model. The learning rate followed the cosine learning rate schedule and was initialized as 0.02. The weight decay was set to be 0.0001 and the batch size 128. The model achieved a precision-at-one score 0.593, a precision-at-five score of 0.829, and a mAP score of 0.636 (loss of 2.75054). Model training took 3 months on 16 V100 GPUs and is available here: https://github.com/pbw-Berwin/M4-pretrained.

**DNN block to cortex correspondence procedure**

We used an encoding model procedure to quantify the correspondence between DNN Blocks and regions of cortex[72,74,195]. For the DNN, we train a Temporal Shift Module (TSM) network[96] with a ResNet50 backbone on the M4 dataset (Multi-Moments in Time Minus Memento10k). In this way, we achieved a model that consecutively processed video frames (as opposed to frame averages), incorporated biologically necessary recurrent computations, and learned to perform a video-based task (i.e., action recognition) from the same set of short, natural videos of which we sample the BMD stimuli.

We ran inference on the TSM ResNet50 model using the 1102 videos used in the fMRI experiment and extracted the activations for each video. The activations for a given block were extracted after the nonlinearity. We then used an encoding model procedure. In detail, we standardized the DNN activations (using the mean and standard deviation of the training videos) and performed principal component analysis (PCA) using the top 100 components of the DNN activations to ensure fair comparison of activations of different embedding sizes. The PCA procedure was fit on the 1000 training set video activations and applied to the 102 testing set video activations. For each voxel $v$, we fit a linear model from the training set DNN activations (size (n_training_videos x n_PCA_components)) to the training set fMRI responses (beta values z-scored across stimuli and averaged over TRs 5-9) averaged over the 3 trial repetitions (size (n_training_videos x n_voxels)). We then predicted testing set voxel responses (size (n_testing_videos x n_voxels)) by applying the linear fit on the testing set DNN activations (size (n_testing_videos x n_PCA_components)). We evaluated the performance of the prediction by correlating (Pearson) the predicted testing set voxel responses with the true testing set fMRI responses of each of the 10 testing set repetitions (size (n_testing x n_voxels)). The final performance of the prediction is the average correlation of the 10 repetitions. The noise-normalized correlation is this 10-repetition average Pearson correlation divided by the voxel's split-half correlation value (the Pearson correlation value before Spearman-Brown). We only predicted the values of the voxels that met the split-half reliability criteria, as described in the Univariate Split-Half Reliability Analysis Methods section above, in order to model meaningful signal.

In this way, we obtained an encoding model accuracy (correlation) at each voxel in the whole brain for each of the four ResNet50 Blocks and for each subject. We averaged the noise normalized correlation at each voxel across subjects for each of the four Blocks and displayed the results in a whole-brain volume (Fig. 4b).

We then computed the difference in encoding accuracy between Block 1 and Block 4 at each of the 22 ROIs to determine if a region's brain responses were predicted significantly better by activations of early or late DNN Blocks. For each subject, we computed both Block 1 and Block 4's average noise-normalized correlation (encoding accuracy) within each ROI and took the difference (Block 1−Block 4). We then performed a t-test (one-sample, two-sided) against a null hypothesis of zero correlation and corrected for multiple comparisons across ROIs (Bonferroni, p < 0.05 with n = 22). We plotted the subject-averaged Block 1−Block 4 noise-normalized correlation differences and denoted the significant ROIs with an asterisk and a color corresponding to significance with Block 1 (blue) and Block 4 (red) (Fig. 4c). See Supplementary Fig. 7 to see each Block's encoding accuracy at each ROI.

We note that while the temporal dimension in videos invites exciting modeling opportunities, it also adds complexities in the fMRI data that may make modeling difficult. For example, regions may reconfigure their roles over the duration of the video or integrate features over time in a manner that cannot be resolved with fMRI. Thus, the extent that models can predict fMRI brain responses to videos may be inherently limited by the temporal resolution of fMRI and best be modeled alongside millisecond-level temporal resolution neuroimaging data (M/EEG).

**Shuffling analysis to determine importance of temporal order**

To determine whether the temporal order of visual information is preserved in fMRI responses of the human visual system, we compared the encoding performance of the TSM model trained on the M4 dataset (Multi-Moments in Time Minus Memento10k) with preserved order of visual information with the model with a randomly shuffled order of visual information. TSM requires eight frames as the input to the model. These frames were sampled uniformly. In the original (unshuffled) order, the order of frames was preserved for all the videos. In the shuffled case, the indices of the frames were shuffled randomly, and then for all the videos, the same order of shuffled indices was used to create the input to the TSM model. We used ten such random shuffles of indices to introduce more randomness.

We first extracted the activations from the four blocks of the TSM model for the unshuffled case and each shuffled case. Then we performed PCA to extract the top 100 components of each block's activations to reduce the number of features and equate their dimensionality while preserving the variance in the activations. Then, we performed an encoding model procedure to predict the fMRI responses of the testing videos. We repeated the shuffling ten times and then took the mean encoding correlation across ten shuffles to compare with the encoding results using unshuffled order of frames. As brain responses, we used beta values z-scored across stimuli and averaged over TRs 5-9 (TRs 5-9 reflect the peak of a typical BOLD response). We only predicted the reliable voxels, as defined in the Univariate Split-Half Reliability Analysis Methods section above.

At each of the four blocks, we computed the difference in encoding accuracy (correlation) between activations that used the unshuffled and shuffled video input (unshuffled minus shuffled). We visualized the difference in correlation at each block in a whole-brain volume (Fig. 5b). Within each Block, we then computed the difference in encoding accuracy (unshuffled minus shuffled) within each ROI. We performed a t-test (one-sample, two-sided) between the subject-averaged difference in correlation and a null hypothesis correlation of zero. We corrected for multiple comparisons across ROIs (FDR, assuming positive correlation, p < 0.05 with n = 88 comparisons). The bar plot in Fig. 5c displays the subject-averaged unshuffled minus shuffled correlation difference at each TSM ResNet50 Block and ROI. Significant blocks were marked with an appropriately colored asterisk.

We additionally correlate (Pearson) the unshuffled activations with each of the ten shuffled activations at each of the four blocks to examine the effect of frame shuffling on the model itself without the brain data. We average the correlations across activation units and the ten random seeds and find the following block correlations:

Block 1: 0.998
Block 2: 0.984
Block 3: 0.928
Block 4: 0.808

These results show decreasing activation similarity between the unshuffled and shuffled activations through the four blocks, with a notably steep drop in activation similarity by block 4. This result suggests that block 4 is most impacted by frame shuffling. Subsequently, block 4 also has the largest impact in brain prediction performance across cortex (Fig. 5b) and ROIs (Fig. 5c).

## Encoding and variance partitioning analysis procedure

We use an encoding model and variance partitioning analysis to identify any unique variance explained by the first and third video seconds in the brain activity. In this way, we measure if the brain activity captures temporal content. The encoding algorithm involved two steps. In the first step, we fed the first and third video second frames of the 1000 training and 102 testing videos to an AlexNet architecture[196] pre-trained on the ILSVRC-2012 image classification challenge[197], and we extracted the corresponding activations at each layer. We then applied the following operations to the activations of both video seconds (first and third), independently: we appended the feature maps of all layers, averaged them across frames, standardized them (using the mean and standard deviation of the training videos feature maps) and downsampled them to 100 components through principal components analysis (PCA) (computed on the training videos feature maps). This resulted in the training activations of size 1000 ×100 x 2 and testing activations of size 102 ×100 x 2 (number of videos x features x video seconds). In the second step, we linearly mapped the stimuli videos feature space onto voxel space, thus predicting the fMRI responses to videos. For each combination of subjects (10), fMRI voxels (N), and fMRI TRs (9), we trained the weights of a linear regression model to predict the fMRI training data (averaged over the three repetitions) using the training activations of both video seconds independently as predictors, and then multiplied the learned weights with the test activations. This resulted in two synthetic fMRI test data instances (size 10 subjects × 102 test videos × 9 fMRI TRs × N fMRI voxels), one for each video second.

To test our hypothesis that the brain activity captures information unique to the first and third video seconds, we ran a variance partitioning analysis between the biological fMRI test data and the two instances of synthetic fMRI test data. At each subject, TR, and voxel we ran a searchlight[78] to calculate the portion of the biological fMRI test data (averaged over the ten test set repeats) uniquely explained by, respectively, the synthetic fMRI test data of the first or third video seconds. The unique biological fMRI variance explained by the first/third video second fMRI synthetic data consisted in the adjusted $R^2$ score of a linear regression trained to predict the biological fMRI data using both synthetic fMRI data instances as predictors, minus the adjusted $R^2$ score of a linear regression trained to predict the biological fMRI data using only the third/first synthetic fMRI data instance as predictor. We then observed at which TRs the unique variance explained by the two versions of synthetic test data peaked, and subtracted the peak TR of the first video second synthetic data from the peak TR of the third video second synthetic data. Next, we created subject wise binary whole brain masks with ones in voxels that show TR peak differences in the range 1 to 3 and zeros elsewhere, summed the binary masks across subjects, and performed a binomial test with FDR correction to remove the non-significant voxels.

The variance partitioning analysis for the ROIs was similar but performed on the reliable (split-half reliability p < 0.05) voxels within each ROI. Again, this results in time courses that reveal how well the synthetic fMRI test data from either the first or third video second explains the real fMRI data at each of the nine TRs. To quantify this difference, we again subtracted the peak TRs of the first and third video second synthetic data.

## Metadata RSA analysis procedure

We performed Representational Similarity Analysis (RSA)[29] between metadata RDMs and neural RDMs to examine the extent that representations defined by the semantic metadata of varying information content are reflected in neural activity. The analysis broadly consisted of correlating a Representational Dissimilarity Matrix (RDM) defined by the metadata (Fig. 7b) with a RDM at each voxel in the brain defined by the brain responses (Fig. 7a).

We defined five metadata RDMs—object, scene, action, object+scene+action, and text description RDMs—from each of the 102 testing set videos. The object, scene, action, and text description RDM was defined by first indexing the annotations from the first five annotators (if the video contained annotations from more than five annotators). We then feed the 5 object, 5 scene, 5 action, and 5 text caption metadata from each of the 102 testing set videos into a language model to generate vector embeddings for each label. In the case of the object labels, since each annotator labeled up to three different objects, we computed the word embedding of each object label individually then averaged them (object labels corresponding to No more objects in video were skipped) to obtain one object embedding per annotator. The object, scene, and action labels were fed into the FastText model[114] to compute single-word embeddings (length 300) and the text descriptions were fed into the Sentence-BERT[115] model to compute sentence-level embeddings (length 384). To minimize the effect of noise in the metadata labels, we averaged the 3 most similar vector embeddings together to result in a single vector embedding that represents the object, scene, action, or text caption for that video. At this step, the object+scene+action embedding was created by concatenating the individual object, scene, and action vector embeddings (length 900). We then computed the pairwise cosine distance between each video's vector embedding to produce a single 102 ×102 Representational Dissimilarity Matrix (RDM) for the object, scene, action, object+scene+action, and text caption metadata. Figure 7b shows the rank-normalized (rank each distance value and divide by the maximum rank) RDM for the object, scene, action, object+scene +action, and text description RDMs.

We compute the correlations (Spearman) between the metadata RDMs to measure the similarity of their information content. As expected, the correlations between the individual single-word object, scene, and action labels were generally lowest, since these labels highlight explicitly different components of the video. The correlations between the individual single-word object, scene, and action labels with the text descriptions were generally next highest, also expected because the text description likely contains information about each of the single-word labels plus extra information. The combined object+scene+action RDM with the text description RDM was higher still given even more overlapping information. The highest similarities were between the single-word object, scene, and action RDMs and the combined object+scene+action RDM because of explicitly overlapping information content (computed using the same embedding) but less extra information than the text description.

[scene, object]: 0.1808
[scene, action] 0.1133
[scene, scene+object+action]: 0.6641
[scene, text description]: 0.1969
[object, action]: 0.0884
[object, scene+object+action]: 0.6137
[object, text description]: 0.2618
[action, scene+object+action]: 0.5238
[action, text description]: 0.1560
[scene+object+action, text description]: 0.3145

To define the RDMs at each voxel in the brain for each subject, we perform a searchlight analysis in the way described in the Methods section Multivariate searchlight-based reliability analysis. To summarize, we center a sphere (radius of 4 voxels) around voxel $v$ and extract the beta values (TRs 5-9 averaged over repetitions and z-scored across conditions) for all testing set conditions at all voxels encompassed in the sphere. Each stimulus thus has a corresponding vector of beta values, one from each voxel within the searchlight sphere. We compute the 1-Pearson's R correlation between all pairs of stimulus vectors to obtain an RDM at the centered voxel $v$. We repeat this process for all voxels in the whole brain for each subject (Fig. 7a).

We then correlate (Spearman's R) the metadata RDM (cosine-distance, not rank-normalized) with the searchlight-based RDMs at each voxel for each of the 10 subjects separately. For the whole-brain analysis (Fig. 7c), we compute a t-test (one-sample, two-sided) against a null hypothesis of a correlation of 0 at each voxel then perform FDR correction (q = 0.05, assuming positive correlation) on all p-values in the whole brain to obtain a set of significant voxels. We compute the noise-normalized correlation by dividing the correlation with the voxel's upper noise ceiling and plot the 10-subject average noise-normalized correlation at each significant voxel (Fig. 7c). For the ROI-based analysis (Fig. 7d), after we correlate (Spearman's R) the metadata RDM with the searchlight-based RDM at each voxel, we compute the average noise-normalized correlation within each ROI. For each ROI, we compute a one-way ANOVA test between the average noise-normalized correlations corresponding to the five semantic metadata models. If the p-value of the ANOVA test is significant (p < 0.05, Bonferroni corrected with n = 22 ROIs), we perform a pairwise Tukeys Honestly Significant Difference test (alpha=0.05). Significant differences between a pair of metadata models are reported with the dual-colored bars under the ROI name in Fig. 7d.

The RSA analysis comparing the sentence text descriptions of the full video with the sentence text descriptions of a single frame, presented in Supplementary Fig. 11, follow a similar pipeline as above. First, we generate five different captions of the middle frame of each video using the captioning model GIT, version git-large-coco[123] (generation parameters max_length=100, num_beams=5, temperature=1, top_k = 250, top_p = 1). These captions are available alongside the human-annotated metadata but in a separate file.

Following the same pipeline as used for the full video sentence text descriptions above, we compute vector embeddings using Sentence-BERT[115] for each of the five frame captions. We average the top 3 most similar captions and compute the pairwise cosine distance (1-cosine similarity) between each test set video's averaged embedding to obtain a $102 \times 102$ RDM.

The frame text description RDM has the following Spearman correlations with the other metadata RDMs:

[frame text description: object]: 0.2451
[frame text description: action]: 0.1335
[frame text description: scene]: 0.2025
[frame text description: scene+object+action]: 0.2878
[frame text description: video text description]: 0.6653

We correlate (Spearman's R) the frame text description RDM with the searchlight-based neural RDMs at each voxel for each subject separately. We then normalize the correlation by each voxel's upper noise-ceiling and average the correlations within the ROI (Supplementary Fig. 11a). We compute the difference in correlation (Supplementary Fig. 11b) between the full video text description and the frame text description at the subject-level and compute statistical significance for each ROI against 0 correlation (p < 0.05, one sample two-sided t-test, Bonferroni corrected with n = 22 ROIs).

### Memorability analysis procedure

For each subject, we averaged the beta values (z-scored across conditions) over TRs 5-9 and over repetitions (3 repetitions per training set stimuli and 10 repetitions per testing set stimuli) to obtain one beta value per video. From the memory game implemented by Newman and colleagues[41], we had one memorability score per video. Under the hypothesis that the magnitude of brain response positively correlates with stimuli memorability[30,31,127], we performed a ranked correlation (Spearman's R) between the vector of memorability scores (size (1102 ×1)) and the vector of beta values (size (1102 ×1)) at each voxel for each subject. In this way, we obtained a correlation value at each voxel in the brain for each subject.

For the whole-brain analysis, we first performed a t-test (one-sample, one-sided) at each voxel against a null hypothesis of zero correlation. We then performed FDR correction on the p-values (q = 0.05, assuming positive correlation). We visualized the subject-averaged correlations of the significant voxels that passed FDR correction in the whole-brain volume (Fig. 8b).

For the ROI analysis, we computed the average correlation within each ROI for each subject. We then computed a t-test (one-sample, one-sided) for each ROI against a null hypothesis of zero average correlation and corrected for multiple comparisons (p < 0.05, Bonferroni corrected with n = 22). We plotted the subject-average correlation at each ROI (Fig. 8c) and denoted significance with an asterisk and a green colored bar. A one-sided, as opposed to two-sided, test against a correlation of 0 was computed because there exists clear a-priori hypotheses that relate memorability effects to larger (not smaller) magnitudes of response[16,30,31,127].

### Version A and version B data pipelines

We provide two preprocessed versions of the (f)MRI data that are identical up until the fMRIPrep preprocessing stage (see Supplementary Fig. 12). The version A pipeline, detailed here, is used in the main manuscript and Supplementary Figs. 1-11. The version B pipeline is used for Supplementary Figs. 13-14. Version B provides the data in five output spaces (native volume, native surface, MNI152NLin2009cAsym volume, fsLR32k surface, and fsaverage surface) compared to version A's one output space (MNI152NLin2009cAsym), uses a newer version of fMRIPrep (version 23.2.0), differentiates between left and right hemisphere for the ROIs, and estimates single trial beta values with GLMsingle[34]. Beta value estimates are provided for Version B data in MNI152NLin2009cAsym volume space (Supplementary Fig. 13), fsLR32k surface space (Supplementary Fig. 14), and fsaverage surface space. For more details on Version B preprocessing and analysis, see Supplementary section Version B preprocessing pipeline.

### Reporting summary

Further information on research design is available in the Nature Portfolio Reporting Summary linked to this article.

## Data availability

The (f)MRI data, stimulus metadata, and TSM ResNet50 model weights generated in this study have been deposited in the OpenNeuro database under accession code ds005165. The original video stimuli can be accessed from the Moments in Time, Multi-Moments in Time, and Memento10k datasets, available at the following links [http://moments.csail.mit.edu/] and [http://memento.csail.mit.edu/]. Source data are provided with this paper.

## Code availability

Code used in this manuscript have been provided alongside the data in the OpenNeuro database under accession code ds005165. The TSM ResNet50 model training code is available here [https://github.com/pbw-Berwin/M4-pretrained]. Starter code demonstrating basic usage of the dataset is available here [https://github.com/blahner/BOLDMomentsDataset].

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

## Acknowledgements

This research was funded by the Multidisciplinary University Research Initiative (MURI) award by the Army Research Office (grant No. W911NF-23-1-0277) to A.O; the Vannevar Bush Faculty Fellowship program funded by the Office of Naval Research (N00014-16-1-3116) to A.O.; the DFG (CI-241/1-1, CI241/1-3, CI-241/1-7) and ERC grant (ERC-2018-StG- 803370) to R.M.C.; the DFG (DFG Research Unit FOR 5368) and the Alfons and Gertrud Kassel foundation to G.R.; the MIT EECS MathWorks Fellowship to B.L.; Berlin School of Mind and Brain PhD scholarship to P.I.; the NIH grant R01EY034118 to K.K; K99/R00 Pathway to Independence Award from the National Eye Institute of the NIH (grant 1K99EY032603) to N.A.R.M. We also thank the MIT-IBM Watson AI Lab for support. The experiments were conducted at the Athinoula A. Martinos Imaging Center at the McGovern Institute for Brain Research, Massachusetts Institute of Technology, on a Siemens PrismaFit 3 T scanner (Erlangen, Germany) supported with funding from a NIH Shared Instrumentation Grant (1S10OD021569). We would like to thank Santani Teng and Emilie Josephs for their valuable writing feedback.

## Author contributions

R.M.C. and A.O. conceptualized the study. R.M.C., A.O., and G.R. supervised the work. R.M.C., A.O., K.K., P.I., and M.G. designed the MRI protocols. A.L. and A.O. selected the stimuli. B.L., A.L., P.I., M.G., and N.A.R.M. collected the data. B.L., P.I., M.G., and K.D. preprocessed the data. B.L., K.D., and A.G. analyzed and visualized the data. B.P. trained the ResNet50 TSM model. S.J. and B.L. ran crowd-sourced stimulus labeling experiments. B.L., S.J., and A.L. quality checked the stimuli metadata. B.L. and R.M.C. wrote the original draft of the manuscript. All authors reviewed and edited the manuscript.

## Competing interests

The authors declare no competing interests.
