## [Peer Review File · Nature Communications]

Modeling short visual events through the BOLD Moments video fMRI dataset and metadata

Editorial Note: Parts of this peer review file have been redacted as indicated to avoid any copy right infringement.Reviewer #1 (Remarks to the Author):

This manuscript presents a rich fMRI dataset collected from ten human participants while watching 1,102 short naturalistic video clips collected from the Moments in Time database depicting visual events. The authors provide a range of different quality measures of the data and a number of regions of interests in early visual cortex and the ventral visual stream. In addition, the authors explore to which degree the dataset can be used to explore different stages involved in the processing of visual events. To this aim, they present a number of modelling approaches in combination with representational similarity analysis and encoding and variance partitioning to capture higher-level visual aspects of the stimuli, and word- and sentence-level descriptions. These models are derived from different layers of a deep neural network and human ratings. In addition, the authors examined to which degree the data capture information regarding memorability scores and temporal order.

There is a lot to like about this manuscript. The authors use a number of state-of-the-art approaches carried out in a rigorous way demonstrating the wide range of ways in which this dataset can be used. In light of the growing interest in the use of naturalistic stimuli in Cognitive Neuroscience and reproducibility, I am convinced that this dataset is going to be of interest to a wide audience struggling to find a good balance between ecological validity and experimental control. That said, I think this manuscript could profit from a somewhat deeper embedding into the existing literature – please find below suggestions for threads the authors may want to pick up. Moreover, given the large number of different analyses reported in this manuscript, I believe that the readability of the manuscript could be improved substantially if the authors offered some guidance from time to time regarding the questions they aimed to address, and if occasionally some more details are added, and/ or explicit links to the corresponding methods sections are established. I hope that my comments and suggestions below can help the authors to further improve this otherwise really interesting and rich manuscript.

signed by Angelika Lingnau

Major comments

(01) In the introduction (first paragraph), the authors state that the processing of visual events engages regions performing visual object recognition, temporal integration and memory. While this is certainly true, the authors may want to acknowledge that many more processes are known to contribute to the processing of visual events, including information about scenes, action semantics, body posture and movement kinematics (in line with some of the analyses using metadata labels reported in the manuscript).

(02) In the introduction (4th paragraph), the authors state that they aimed to predict brain responses in the dorsal visual and parietal cortices. The authors may want to motivate this choice, also in light of a growing literature that points towards the importance of ventral stream regions for the processing of visual events (for reviews, see e.g. Lingnau & Downing, 2015; Wurm & Caramazza, 2021; Pitcher & Ungerleider, 2021). The same applies to several statements throughout the manuscript (e.g. page 21, line 455f: 'despite the parietal cortex' heavy involvement in action observation).

(03) Overall, the introduction might profit from some examples regarding open questions that one might want to address with the current data set.

(04) Overall, many of the results would be easier to understand if the authors provided brief statements regarding the purposes of the analyses, and if they pointed to the corresponding sections in the methods section for details. To give a concrete example, on page 11, line 246, the authors write that they accessed multivariate reliability (using RSA) to determine the upper and lower estimate of the noise ceilings (which I really appreciate). However, at this point the reader does not know yet what kind of models are entered into the RSA, and what the purpose of these noise ceilings is. Admittedly most of these points are explained later on in the methods section, but it would save the reader a lot of time and effort if once in a while some insights regarding the purpose and general logic of an analysis was provided upfront.

(05) Results, page 11: The authors state that they observed statistically significant reliability values across occipital and parietal cortex. It seems to me that Figure 3 also suggests an involvement of temporal regions, which brings me back to the point addressed in one of my

previous comments regarding the processing of visual events in the ventral stream. The same comment applies to the paragraph on page 14, where the dorsal visual stream and parietal cortex is highlighted.

(06) Results, page 15f, frame shuffling: I really liked this analysis. That said, I was wondering to which degree the performance of the DNN (rather than the brain) in predicting DNN activations for unseen unshuffled sequences is impaired by the shuffling. I'm assuming the authors could examine this and determine if they find an increase of the impact of shuffling across the different blocks of the DNN that resembles the pattern obtained across the different ROIs. I do not consider this analysis to be essential for the manuscript, but it could be something worth looking into if the authors wish to do so.

Minor points

(1) Figure 4, captions: The authors write that 'All videos are shown to both a DNN and a human.' The authors may want to state which DNN they used – they do so in panel B, so I'm assuming they used the same DNN here, but why not make that explicit from the start.

(2) I appreciate the level of detail the authors provide to access the quality of the (f)MRI data. That said, some of the reported image quality metrics might not be known to the readers of Nature Communications, so some more guidance regarding the interpretation of these measures might be helpful.

(3) Figure 4, captions, panel C: It did not become entirely clear to me why the difference in predictive performance was computed between block 1 and 4 – why these particular blocks? Can the authors motivate this choice?

(4) Methods, page 6, line 129: It might help to inform the reader already here which of these labels were generated by humans, and which were generated by a DNN.

(5) Methods, page 7, line 161: What was the reason to resample the data from a TR of 1.75 to 1 s, instead of collecting data at a TR of 1 s right away?

(6) On page 14, line 301f, the authors write that they release the model to aid investigations of visual event understanding. Here might be a good location to state where the model can be found.

(7) Figure 7: How similar are the models for scenes, actions and text caption?

(8) Figure 7, panel A: At this point in the manuscript it was not clear to me yet what video metadata are. Also here, the authors may want to refer to the corresponding methods section.

(9) Results, page 21, line 447f, RSA: here I was wondering how the neural RDM was constructed. The authors may want to provide the reader with a link to the corresponding methods section.

(10) Methods, Stimuli, page 27, line 577f: How was the semantic overlap between a pair of videos determined?

(11) Methods, Functional localizers, page 27, line 597f: It would be helpful to provide the reader with more details about these videos. Can the authors show some example snapshots? Moreover, I was wondering whether the authors can provide a justification regarding the use of videos rather than static images, in particular for stimulus categories such as objects and scenes.

(12) Related to the previous point: Was there a fixation cross provided during the functional localizer?

(13) Methods, Main experiment, page 28, line 626f: How were the null events modelled in the analysis?

(14) Methods, Main experiment, page 28, line 632f: I don't fully get the meaning of the statement 'with the restriction of excluding successive repetitions of test runs within one session'. Could the authors clarify?

(15) Methods, Main experiment and functional localizer: I did not find any information regarding a fixation period at the beginning and end of each run – is this simply an omission in the text, or was no such fixation period included?

(16) Page 29, Metadata, line 638: Do the five annotations refer to each stimulus or each rater (or something else)?

(17) Related to the previous point, on line 640, the authors write that the five annotations were collected to ensure comprehensive coverage, and to form a group consensus. I might have misunderstood, but this seems to be a bit in conflict with the statement on line 645f, which states that the authors did not enforce that all 5 object labels describe the same object.

(18) Page 29, Metadata, Action labels: Why did the authors select one action label that best described the video, but up to five different object labels (see comment above)? The authors argue that this is to reflect the participant's limited ability to perceive every day action due to limited

stimulus presentation time and their central fixation. Why would this apply to actions but not to objects?

(19) Page 30, line 686f, Memorability score and decay rate: how is the memorability decay rate defined?

(20) Page 32, line 730f: What makes this template asymmetrical?

(21) Page 32, line 735f: The authors state that the fieldmap was estimated on the basis of a phase-difference map using a dual-echo GRE sequence, but I could not find such a sequence among the sequences that were listed in the section 'fMRI data acquisition' on page 31.

(22) Related to the previous point, the authors state that the fieldmap was co-registered to the target EPI reference run. Which run did they use as the target?

(23) Page 33, line 759: Please define DVARS.

(24) Page 33, line 765f: How was the set of physiological regressors extracted?

(25) Page 34, line 813f: The authors may want to state more explicitly what were their regressors of interest.

(26) Page 37, line 884f: The reader might be curious to be provided with a few more details regarding how the authors used the Glasser atlas, given that there exist a number of different ways in which one can extract them.

(27) Page 37, line 909f: the raw beta values of what?

(28) Page 38, line 918f: Since not all readers might be familiar with voxel-wise upper and lower noise ceilings, could the authors provide a reference?

(29) Page 38, line 933f: what is LSEP loss?

(30) Page 38, line 937f: How were some of the details of the analysis determined (e.g. the split of the input video into 8 segments, sampling 1 frame from each segment; initialization of the learning rate and the weight decay with values of 0.02 and 0.0001, respectively)?

(31) Page 39, line 951: Please provide a reference for the encoding model procedure.

(32) Page 39, line 956: The statement regarding PCA on the DNN activations would profit from a few more details. As an example, what does the N refer to – number of components?

(33) Page 40, line 993f: What is the justification to extract the top 100 components for each block?

(34) Page 40, line 997f: What was the reasoning behind averaging beta values across TRs 5-9 (i.e., why these specific TRs)?

(35) Page 41, Encoding and variance partitioning analysis procedure: I found the entire first paragraph of this section hard to understand. It might be easier to follow if the authors provided the reader with a brief statement upfront regarding the purpose of this analysis. The authors also may want to double-check the readability of this paragraph (as an example, what exactly do the authors mean by 'feature maps of shape' (line 1019f)?)

(36) Related to the previous point, on page 41, line 1029, the authors write 'To test our hypothesis', but at this point I really was not clear about what that might be.

(37) Page 42, RSA-based decoding analysis procedure: Also here, a brief statement regarding the purpose of the analysis would be helpful. I also didn't fully follow the reasoning by the statement made in the last two sentences of the first paragraph (lines 1060-1663).

(38) Page 43, line 1096: What is the reason to use one-sided t-tests for the Memorability analysis procedure, and two-sided t-tests for the RSA-based decoding procedure?

References

Lingnau, A., & Downing, P. E. (2015). The lateral occipitotemporal cortex in action. *Trends in Cognitive Sciences*, 19(5), 268-277.

Pitcher, D., & Ungerleider, L. G. (2021). Evidence for a third visual pathway specialized for social perception. *Trends in Cognitive Sciences*, 25(2), 100-110.

Wurm, M. F., & Caramazza, A. (2021). Two 'what' pathways for action and object recognition. *Trends in Cognitive Sciences*.

Reviewer #2 (Remarks to the Author):

In this paper, the authors introduce a new repository of data collecting whole-brain fMRI responses from 10 subjects to over 1000 short (3s) naturalistic video clips of visual events. They assessed the quality of preprocessing and showcased some exploratory results to highlight the potential of BMD in addressing temporal events, high-level concepts, and memorability topics using BOLD signals. Overall, this dataset is timely for fMRI-based neural encoding research. My comments are as follows:

1. In Figure 1c, the differences between various blocks appear relatively small. How do the authors address the influence of previous stimulus blocks, especially considering hemodynamics and memory effects? There needs to be a justification for using the canonical HRF approaches to mitigate memory impacts.

2. In relation to Figure 4, the evaluation hinges on the choice of TSM ResNet50. Can the authors elucidate the roles of different blocks in TSM ResNet50 from a computer vision standpoint? This clarification might help in understanding the alignment between the two network types. Moreover, the stationarity of dynamics with the TR remains unclear, making it difficult to discern whether a region reconfigures its role or maintains a singular mode corresponding to a TSM ResNet50 block. Furthermore, given the plethora of existing research on static image encoding, this section doesn't offer new insights.

3. For Figure 5, I'm primarily concerned about the extent to which the results hinge on the choice of TSM. What kind of temporal dynamics is encoded by frame order? For instance, even when the order is shuffled, the variance—potentially another form of dynamics—remains unchanged. Given the temporal resolution of BOLD and the brain's processing speed for image sequences, the concept of "dynamics" requires further elucidation and discussion.

4. Pertaining to Figure 6, how reliable are the TR peak estimations? Given the size of the unique variance, I'm skeptical about the validity of the claims presented. Additionally, it seems the only definitive conclusion is that BOLD signals can differentiate between images with a 1-2TR delay. Contrasting experiments using the same stimuli might be beneficial.

5. As for Figure 7, while the findings are intriguing, they aren't novel. A more extensive static dataset might provide clearer insights. I recommend the authors explore meta-information that can uniquely be derived from the videos.

In summary, this dataset is crucial and beneficial for the broader research community. Concerning the manuscript, if the submission is to Scientific Data, its current content seems apt. However, if intended for Nature Communications, the content should emphasize the dynamic aspects and condense sections relating to static image stimuli.

Dear Reviewers:

Thank you for your thoughtful feedback that led to what we believe is a greatly improved manuscript. We summarize the major revisions following your suggestions:

(1) Significant textual edits throughout the manuscript to clarify procedures, highlight the dataset's potential, and better motivate existing analyses. These edits include a new supplementary discussion section titled, "The added value of brain responses to a short video dataset versus a static image dataset" [reviewer 1 major comments 1-5, minor comments 1-38; reviewer 2 major comments 1, 4, 5].

(2) We ran additional crowd-sourced experiments to include human-annotated (instead of top-5 ResNet50 predictions) scene and object metadata. These new annotations are more accurate than AI generated labels, provide human ground truth values, and overlap with additional computer science/neuroscience datasets (Places365 and THINGS) for even broader appeal among researchers. All provided metadata are now sourced from human experiments. As a consequence of this change, we revisited the "Semantic metadata reveal strong similarity between sentence-level descriptions and visual brain activity" section to include the new object and scene labels. This inclusion provides greater insight into how well ROI representations reflect spatial (object and scene), temporal (action), spatiotemporal (objects+scenes+actions), and relational (sentence descriptions) information [reviewer 1 major comments 4, minor comments 4, 7, 8, 9, 16, 17, 18, 37; reviewer 2 major comment 5].

(3) We ran additional experiments related to the analyses presented in Figure 4 and Figure 5 to measure the generalization of our results and better elucidate the temporal dynamics captured in the BOLD signal [reviewer 1 major comment 6; reviewer 2 major comment 2 and 3].

(4) We release another version of the dataset to further enhance impact and adoption, detailed in Supplement 2, output from an additional fMRIPrep preprocessing pipeline. We provide single trial beta estimates of this data version using GLMsingle (Prince et al., 2022) in MNI volumetric space and beta estimates using GLMsingle (Prince et al., 2022) in cortical surface space converted to the HCP preprocessing format using Ciftify (Dickie et al., 2019; Glasser et al., 2013). We believe these additional spaces, registrations, and preprocessed formats will greatly facilitate this dataset's wide adoption in the field. We additionally provide an analysis predicting these single trial brain responses from motion energy features (Adelson & Bergen, 1985; Nishimoto et al., 2011) and show the highest accuracies are in motion-sensitive regions, including MT [reviewer 1 major comment 3; reviewer 2 major comment 1, 5].

We respond to your comments below in bold. Revised text is tracked in the revised manuscript in **red**. Any revised text from the manuscript reproduced below will also be shown in **red bold font**. Your reviewer comments are reproduced in **unbolded green text**, our responses in the rebuttal in black normal font.

Reviewer #1 (Remarks to the Author):

This manuscript presents a rich fMRI dataset collected from ten human participants while watching 1,102 short naturalistic video clips collected from the Moments in Time database depicting visual events. The authors provide a range of different quality measures of the data and a number of regions of interests in early visual cortex and the ventral visual stream. In addition, the authors explore to which degree the

dataset can be used to explore different stages involved in the processing of visual events. To this aim, they present a number of modelling approaches in combination with representational similarity analysis and encoding and variance partitioning to capture higher-level visual aspects of the stimuli, and word- and sentence-level descriptions. These models are derived from different layers of a deep neural network and human ratings. In addition, the authors examined to which degree the data capture information regarding memorability scores and temporal order.

There is a lot to like about this manuscript. The authors use a number of state-of-the-art approaches carried out in a rigorous way demonstrating the wide range of ways in which this dataset can be used. In light of the growing interest in the use of naturalistic stimuli in Cognitive Neuroscience and reproducibility, I am convinced that this dataset is going to be of interest to a wide audience struggling to find a good balance between ecological validity and experimental control. That said, I think this manuscript could profit from a somewhat deeper embedding into the existing literature – please find below suggestions for threads the authors may want to pick up. Moreover, given the large number of different analyses reported in this manuscript, I believe that the readability of the manuscript could be improved substantially if the authors offered some guidance from time to time regarding the questions they aimed to address, and if occasionally some more details are added, and/ or explicit links to the corresponding methods sections are established. I hope that my comments and suggestions below can help the authors to further improve this otherwise really interesting and rich manuscript.

signed by Angelika Lingnau

Major comments

(01) In the introduction (first paragraph), the authors state that the processing of visual events engages regions performing visual object recognition, temporal integration and memory. While this is certainly true, the authors may want to acknowledge that many more processes are known to contribute to the processing of visual events, including information about scenes, action semantics, body posture and movement kinematics (in line with some of the analyses using metadata labels reported in the manuscript).

Thank you for highlighting the large number of relevant processes in visual event perception. We agree that the original text unintentionally read like other processes were not involved and did not adequately communicate the vast number of processes engaged in visual event understanding. The revised text aims to communicate the broad complexity of visual event understanding while explicitly highlighting specific processes that you point out and are related to analyses presented later in the manuscript. The first paragraph of the introduction now reads:

“Understanding visual events is a hallmark of human intelligence that engages a distributed and functionally diverse network of cortical regions. For example, extracting the meaning of even a simple event, such as a person opening a door, requires at a minimum parsing the scene into relevant components (a person, door, indoor room) (Carandini, 2005; DeYoe & Van Essen, 1988; DiCarlo et al., 2012; Felleman & Van Essen, 1991; Logothetis & Sheinberg, 1996; Ress & Heeger, 2003) and integrating these components over time for accurate action recognition (the door is being opened, not closed) (Fairhall et al., 2014; Hasson, Yang, et al., 2008; Lingnau & Downing, 2015; Orlov & Zohary, 2018; Wurm & Caramazza, 2022), including socially relevant cues (is the person opening the door angry or friendly?) (McMahon et al., 2023; Pitcher et al., 2011; Pitcher &

Ungerleider, 2021). Part of the event is also encoded into brain memory regions for later recall (Bainbridge, 2019; Bylinskii et al., 2022; Han et al., 2015; Hasson, Furman, et al., 2008; Schneider, 2013). Due to the richness of this cortical network, ecologically-valid visual event understanding is challenging to study with experimental rigor.”

(02) In the introduction (4th paragraph), the authors state that they aimed to predict brain responses in the dorsal visual and parietal cortices. The authors may want to motivate this choice, also in light of a growing literature that points towards the importance of ventral stream regions for the processing of visual events (for reviews, see e.g. Lingnau & Downing, 2015; Wurm & Caramazza, 2021; Pitcher & Ungerleider, 2021). The same applies to several statements throughout the manuscript (e.g. page 21, line 455f: ‘despite the parietal cortex’ heavy involvement in action observation).

Thank you for pointing out this body of literature that even further highlights the importance using video stimuli to study visual perception. Originally, focusing on predicting dorsal visual and parietal cortices was seen as a major differentiator to still image datasets, where these regions do not or only weakly respond. However, we agree that predicting ventral regions’ response to videos presents great opportunities to better understand visual perception. We clarify that we do also predict brain responses in all cortex, including ventral cortex. We updated the manuscript in the following places:

Introduction – we change the wording to emphasize, to your point, that regions evoked by both visual events and still images can be functionally different:

“Short video stimuli supplement work using still images by driving a greater extent (Bartels & Zeki, 2004; Konen & Kastner, 2008; Press et al., 2001; Schultz & Pilz, 2009; Yildirim et al., 2019) and different pattern (Buccino et al., 2004; Kret et al., 2011; Lingnau & Downing, 2015; Wurm & Caramazza, 2022) of neural responses.”

Introduction – we clarify the sentence describing the DNN prediction analysis that your comment refers to:

“Aided by BMD’s large number of stimuli and widespread reliable activity, we use a video-computable deep neural network (DNN) trained on an action recognition task to predict brain responses through all cortex, including the dorsal visual and parietal cortices.”

Results – Metadata RSA Analysis – we emphasize that regions throughout visual cortex are involved in action recognition. We remove the line ‘despite the parietal cortex heavy involvement in action observation’ and instead highlight the result of the action metadata’s higher correlations in the non-parietal regions. The motivation of the analysis now reads:

“It is unclear how these varying levels of complexity and content are reflected in ROIs while viewing visual events, especially given that regions throughout the ventral visual, dorsal visual, and parietal cortices have all been implicated in processing temporal aspects of videos (Buccino et al., 2004; Konen & Kastner, 2008; Lingnau & Downing, 2015; Silver & Kastner, 2009; Wurm & Caramazza, 2022) but also diverse feature preferences (Buccino et al., 2004;

Kanwisher, 2010; Kanwisher et al., 1997; Konen & Kastner, 2008; Lafer-Sousa et al., 2016; Ratan Murty et al., 2021)."

And the results of this analysis are rephrased to say:

"Action labels (blue) correlate with ventral and dorsal visual regions (Lingnau & Downing, 2015; Wurm et al., 2017) more strongly than parietal regions (Hardwick et al., 2018)."

(03) Overall, the introduction might profit from some examples regarding open questions that one might want to address with the current data set.

We agree the introduction can benefit from more concrete examples of open questions that can be uniquely addressed with this dataset. In response to your first comment, we revised the first paragraph of the Introduction to better emphasize the plethora of visual processes engaged in visual event understanding. This revised paragraph also points the reader to processes, or combinations of processes, that can be addressed with BMD. We also add a few sentences to the last paragraph of the introduction to highlight methodological and cognitive open questions that can be addressed with this dataset. Lastly, we add a supplementary discussion section titled, "The added value of brain responses to a short video dataset versus a static image dataset" to detail unique research opportunities afforded by a short video dataset like BMD. The last paragraph of the introduction now reads:

"Together, BMD's 1,102 thoroughly sampled short video stimuli isolate brain responses to ecologically-valid but experimentally controlled dynamic content. This dataset is well-suited to address open questions as diverse as developing methodologies to model rapid event BOLD signals (Misaki et al., 2013; Prince et al., 2022; Wittkuhn & Schuck, 2021), characterizing interactions between visual processing pathways (Lingnau & Downing, 2015; Mineault et al., 2021; Pitcher & Ungerleider, 2021; Wurm & Caramazza, 2022), and bridging the gap between still image and longform movie perception (Alijo et al., 2020; Allen et al., 2022; Hanke et al., 2016; Hebart et al., 2019) (see Supplementary 1 section, "The added value of brain responses to a short video dataset versus a static image dataset" for further discussion). Crucially, BMD's shared experimental design across subjects enables robust and generalizable conclusions, and its preparation in BIDS format facilitates easy adoption among researchers. Its scale of stimuli and subjects, reliable brain activity, and stimuli metadata enable a wide range of interdisciplinary analyses to reveal the neural mechanisms underlying visual event understanding."

We facilitate these endeavors by providing an additional preprocessed version of the dataset with more output spaces and different ROI definitions to appeal to more researchers. We describe this in the last paragraph of the main text under the section, "(f)MRI data processing, response modeling, and ROI definition":

"All together, this preprocessing suite ensures a low threshold for researchers to interact with the brain data at their desired processing level. All results presented here in the main text use version A of the dataset. We provide another preprocessed version of BMD (version B) to make it available in more output spaces (version B is detailed in Supplementary 2)."

Lastly, the supplementary 2 discussion section, “The added value of brain responses to a short video dataset versus a static image dataset” reads:

The added value of brain responses to a short video dataset versus a static image dataset

“We emphasize that a short video (e.g., 3 second duration, as in BMD) fMRI dataset is not better or worse than a static image fMRI dataset; rather, they are different in terms of stimulus features and corresponding brain responses that may make one better suited to answer specific research questions. Most obvious, short videos contain a temporal dimension that static images do not, allowing the video to communicate crucial contextual information about how spatial components in our environment move (or not) and spatially relate to each other over time. The benefit of this temporal dimension is clear in our everyday lives – we can interpret transitions between states (a door is being opened, not closed), direction (a steering wheel is being turned to left, not right or still), reactions (the child laughed when shown the picture), motion (the baby is crawling slowly, not fast), and more.

The contextual value of a video’s temporal dimension is reflected in BMD’s own action and sentence text description metadata. Concerning action labels, images can only be labelled with a limited subset of actions or else be highly constrained in order to capture a specific action. For example, the action of a baseball player “hitting” the ball can only be captured with an image if the photo were taken at very specific instant in time, otherwise the action may be “standing” or “swinging”. Even a short video like in BMD easily captures these actions without heavily constraining the space of possible videos that correspond to “hitting”. Concerning text descriptions, short videos can capture temporal sequences of events that an image cannot.

Examples of such captions sampled from some of BMD’s first videos include (emphasis our own):

- Video 0001: "A mallard is in the water alone *swimming around and putting its beak in.*"
- Video 002: "A man *is showing another man how to move feet back and forth.*"
- Video 005: "A woman guides a little boy's arms *up and down as other kids stretch* around him."
- Video 006: "a chess tournament is going on this is focused on two players one *is moving their queen and taking something* to put the king in checkmate"

Static frames of these videos cannot capture the temporal facts that the mallard is “putting its beak in”, the man “is showing another man how”, “a woman guides...as other kids stretch”, and a chess player “is moving their queen and taking something to put the king in checkmate.” This temporal information adds valuable context that often makes one’s understanding of the 3s video vastly different compared to any one of its single static frames.

But do these differences in short videos and static images translate to differences in fMRI brain responses? Yes, previous work has found that videos evoke a greater extent (Bartels & Zeki, 2004; Konen & Kastner, 2008; Press et al., 2001; Schultz & Pilz, 2009; Yildirim et al., 2019) and pattern (Buccino et al., 2004; Kret et al., 2011; Lingnau & Downing, 2015; Wurm & Caramazza, 2022) of cortex responding to videos than images throughout occipitotemporal, dorsal visual, and parietal cortex. In this manuscript we describe our highly reliable activations throughout cortex (Figure 3) with notably high reliability in parietal cortex, a region of the brain that weakly responds to static

images. These highly reliable brain responses are not just a result of increased participant engagement or stimulus saliency; we even show that BMD brain responses capture temporal information from the videos (Figure 5, Figure 6, Supplementary 1 Figure 9, Supplementary 2 Figure 2) despite the BOLD response's temporal sluggishness and fMRI's low sampling rate.

In the neighboring field of computer vision, researchers have long recognized that videos and images demand different modeling approaches (Ahn et al., 2023; Bertasius et al., 2021; Lin et al., 2019; Tong et al., 2022; Wang et al., 2016) and training datasets (Goyal et al., 2017; Kay et al., 2017; Miech et al., 2019; Monfort et al., 2020; Soomro et al., 2012) for strong task performance. Videos continue to be at the forefront of ground breaking computer vision research due to their creative, cross-domain, and practical applications in text-to-video generation (Ho et al., 2022; Singer et al., 2022; Wu et al., 2023), video understanding with large language models (Ju et al., 2022; Maaz et al., 2023; Zhang et al., 2023), and efficient action recognition and pose estimation (Liu et al., 2023; Qing et al., 2024; Zheng et al., 2023).

Taken together, short video fMRI datasets offer unique opportunities to advance the field of computational neuroscience where static image fMRI datasets cannot. They can advance methodologies around estimating BOLD signals in response to rapid stimulus presentations (Misaki et al., 2013; Prince et al., 2022; Wittkuhn & Schuck, 2021), elucidate cognitive functions concerning temporal integration (Fairhall et al., 2014; Hasson et al., 2008; Orlov & Zohary, 2018), test temporally specific cognitive objective functions (Doerig et al., 2022; Kanwisher et al., 2023), and detail how multiple visual pathways interact to achieve an understanding of an event (Lingnau & Downing, 2015; Mineault et al., 2021; Pitcher & Ungerleider, 2021; Wurm & Caramazza, 2022). As neuroscience and computer science research become increasingly intertwined (Allen et al., 2022; Chen et al., 2023; Cichy et al., 2019, 2021), BMD is well-suited to integrate with state-of-the-art video modeling work from the computer vision community. Importantly, a short video dataset like BMD can make these scientific advancements while staying connected to the vast body of still image work by sharing event-related paradigms, multivariate and univariate methodologies, representational similarity analyses, and/or encoding and decoding techniques. Short video datasets offer more ecological validity than static images while retaining experimental control and offer tremendous potential to advance our understanding of the human visual system.”

(04) Overall, many of the results would be easier to understand if the authors provided brief statements regarding the purposes of the analyses, and if they pointed to the corresponding sections in the methods section for details. To give a concrete example, on page 11, line 246, the authors write that they accessed multivariate reliability (using RSA) to determine the upper and lower estimate of the noise ceilings (which I really appreciate). However, at this point the reader does not know yet what kind of models are entered into the RSA, and what the purpose of these noise ceilings is. Admittedly most of these points are explained later on in the methods section, but it would save the reader a lot of time and effort if once in a while some insights regarding the purpose and general logic of an analysis was provided upfront.

We updated the manuscript accordingly to better motivate analyses and point the reader to the methods sections:

In “Reliable univariate and multivariate response profiles”:

“Neural encoding and decoding analyses often use single-voxel responses to measure information content in an ROI (Naselaris et al., 2011), and hypothesis models of neural representations derived from computational or behavioral models can be easily compared to the brain in multivariate analysis frameworks (Kriegeskorte & Kievit, 2013). These reliability measures provide intuition on data quality and can be used at the discretion of the researcher to normalize results with respect to the noise in BMD’s data.”

In the “Modeling visual event understanding with a video-computable deep neural network for action recognition” section, we describe the most pertinent details in the text and point the reader to the Methods section for additional details:

“We train the model on an action recognition task using the same dataset from which the BMD stimuli were sampled, the Moments in Time dataset (Monfort et al., 2020, 2022) (BMD stimuli were excluded from model training (see Methods section “DNN Block to Cortex Correspondence Procedure” for encoding model details))”

“Using a voxelwise encoding model approach (Naselaris et al., 2011) (Figure 6A), we observe a correspondence between DNN block depth and predictivity performance along the visual processing hierarchy and beyond (see Methods section “DNN Block to Cortex Correspondence Procedure” for encoding model details).”

In the frame shuffling analysis, we make sure to point readers to the methods section:

“By using the shuffled and unshuffled activations from these blocks to predict the brain responses, we can assess the effect of correct temporal ordering on our BMD brain responses (see Methods section “Shuffling Analysis to Determine Importance of Temporal Order” for details).”

Similarly, in the variance partitioning analysis we add:

“We used variance partitioning to identify the unique contribution of the first and third video epochs’ predictions to the real fMRI responses (Figure 6C) (see Methods section “Encoding and Variance Partitioning Analysis Procedure” for details).”

In the Metadata RSA analysis, we add a stronger motivation for the analysis in line with the work on action recognition in LOTC you cited above. We believe this motivation makes the results much more interesting to researchers:

“Varying levels of semantic information content, from static objects and scenes (e.g., “duck”, “water”) to temporal actions (e.g., “swimming”) to complex relations between parts (e.g., “the duck is swimming on the water”), can describe a visual event. It is unclear how these varying levels of complexity and content are reflected in ROIs while viewing visual events, especially given that regions throughout the ventral visual, dorsal visual, and parietal cortices have all been implicated in processing temporal aspects of videos (Buccino et al., 2004; Konen

& Kastner, 2008; Lingnau & Downing, 2015; Silver & Kastner, 2009; Wurm & Caramazza, 2022) but also diverse feature preferences (Buccino et al., 2004; Kanwisher, 2010; Kanwisher et al., 1997; Konen & Kastner, 2008; Lafer-Sousa et al., 2016; Ratan Murty et al., 2021)."

Also, in this section we point readers to the Methods for additional details:

"We use representational similarity analysis (RSA) (Kriegeskorte, 2008) to correlate the metadata representations (Figure 7B) with neural representations to measure how similarly the different metadata descriptions are reflected in brain activity of dynamic videos (see methods section "Metadata RSA Analysis Procedure" for more information)."

Lastly, the memorability analysis section, we point readers to the Methods for more details:

"Under the hypothesis that stimuli with higher memorability scores elicit a greater magnitude of brain response (Bainbridge et al., 2017; Bainbridge & Rissman, 2018; Jaegle et al., 2019), we correlate a vector of video memorability scores with a vector of each voxel's corresponding brain responses (beta values) (Figure 8A) (see Methods section "Memorability Analysis Procedure" for details)."

(05) Results, page 11: The authors state that they observed statistically significant reliability values across occipital and parietal cortex. It seems to me that Figure 3 also suggests an involvement of temporal regions, which brings me back to the point addressed in one of my previous comments regarding the processing of visual events in the ventral stream. The same comment applies to the paragraph on page 14, where the dorsal visual stream and parietal cortex is highlighted.

We agree that we should better highlight the ventral and temporal regions' involvement in processing videos.

The text you first refer to under "Reliable univariate and multivariate fMRI response profiles" has been updated to:

"In both the whole-brain univariate and multivariate reliability analyses, we observe statistically significant reliability values across the occipital, temporal, and parietal cortices, even extending into the frontal lobe."

The text you refer to under "Modeling visual event understanding with a video-computable deep neural network for action recognition" has been updated to include additional citations emphasizing different neural responses between images and videos:

"However, modeling visual events has been limited by the lack of a suitable dataset that accounts for complex distributed processes across the whole brain (Yildirim et al., 2019), drastic differences to image understanding (Buccino et al., 2004; Krekelberg et al., 2003; Kret et al., 2011; Lingnau & Downing, 2015; Schultz & Pilz, 2009; Senior et al., 2000; Shirai & Imura,

2014; Wurm & Caramazza, 2022), and the temporal boundaries of a visual event (Aliko et al., 2020; Hasson, Yang, et al., 2008; Nishimoto et al., 2011; Seeliger et al., 2019).”

In the same section, we delete the word “notably” to de-emphasize dorsal and parietal cortices:

“We observe that predictivity of DNN Block 4 becomes significantly greater than DNN Block 1 beginning in the ventral visual cortex and extending notably into dorsal visual cortex and parietal cortex (Figure 6C), reflecting the increase of feature complexity in the representations across the visual processing hierarchy (see Supplementary Figure S7 for results on all layers and ROIs).”

Again, in the same section, we rearrange and add to the last paragraph to stress the involvement of ventral visual and temporal cortices in event perception:

“This result extends previous research demonstrating a hierarchical correspondence between DNNs and brains from still image stimuli (Cichy, Pantazis, et al., 2016; Eickenberg et al., 2017; Kriegeskorte, 2015; Kubilius et al., 2019; D. L. K. Yamins et al., 2014) to dynamic video stimuli, a non-trivial outcome given that many cortical regions in the ventral visual and temporal cortex and beyond respond to stimulus features uniquely present in videos and not images (e.g., movement kinematics, temporal interactions) (Lingnau & Downing, 2015; Pitcher et al., 2011; Wurm et al., 2017; Wurm & Caramazza, 2022). These results also help clarify previously conflicting results about whether or not DNNs trained on action recognition tasks accurately predict dorsal stream regions (Bakhtiari et al., 2021; Güçlü & van Gerven, 2017; Mineault et al., 2021), showing that they accurately predict responses not only in the dorsal visual stream but also in the parietal cortex.”

Note that we do keep the focus on dorsal visual stream and parietal regions in the last sentence because the “previously conflicting results” specifically concern those regions.

(06) Results, page 15f, frame shuffling: I really liked this analysis. That said, I was wondering to which degree the performance of the DNN (rather than the brain) in predicting DNN activations for unseen unshuffled sequences is impaired by the shuffling. I’m assuming the authors could examine this and determine if they find an increase of the impact of shuffling across the different blocks of the DNN that resembles the pattern obtained across the different ROIs. I do not consider this analysis to be essential for the manuscript, but it could be something worth looking into if the authors wish to do so.

Thank you for this suggestion, it is definitely a good analysis to do. Since this analysis you propose does not do any direct comparisons with the brain data, there is no need to perform regression onto voxels. Thus, we can simply compute the correlations (Pearson) between the activations from the “unshuffled” and “shuffled” inputs at each block.

Our method is as follows: we extract the activations for each of the 1,102 videos at each of the four blocks in the TSM ResNet50 model trained on the M4 dataset (same model as in the manuscript). We extract activations under two conditions: (1) the frame order is unshuffled and (2) the frame order is randomly shuffled (repeated with ten different random seeds). In all blocks, the activations are

flattened and averaged over the frames. At each of the four blocks, we then correlate the unit activations between the (1) unshuffled activations and (2) each of the ten shuffled activations. For each randomly shuffled seed, the correlations are averaged over the layer units. The final correlations at each block averaged over the ten random seeds are:

Block 1: 0.9984845300607047
Block 2: 0.9842916420921666
Block 3: 0.9282806628390048
Block 4: 0.8082267527419473

These results show decreasing activation similarity between the unshuffled and shuffled activations through the four blocks, with a notably steep drop in activation similarity by block 4. This result suggests that block 4 is most impacted by frame shuffling. We find a similar pattern in brain prediction performance, where in figure 5B, block 4 shows a dramatically larger “unshuffled minus shuffled” brain prediction performance throughout cortex compared to blocks 1-3. Using the model’s sequential processing of activations through the four blocks as a proxy for a cortical processing hierarchy (Cichy, Pantazis, et al., 2016; Eickenberg et al., 2017; Kriegeskorte, 2015; Kumbhani et al., 2019; D. L. K. Yamins et al., 2014), we find a similar trend where block 4 has the largest impact in all cortical ROIs and especially in the high-level regions (figure 5C).

We believe this result is a valuable evaluation of the DNN itself and will be of interest to other researchers. We summarize these results in the corresponding methods section “Shuffling analysis to determine importance of temporal order”:

“We additionally correlate (Pearson) the unshuffled activations with each of the ten shuffled activations at each of the four blocks to examine the effect of frame shuffling on the model itself without the brain data. We average the correlations across activation units and the ten random seeds and find the following block correlations:

Block 1: 0.998
Block 2: 0.984
Block 3: 0.928
Block 4: 0.808

These results show decreasing activation similarity between the unshuffled and shuffled activations through the four blocks, with a notably steep drop in activation similarity by block 4. This result suggests that block 4 is most impacted by frame shuffling. Subsequently, block 4 also has the largest impact in brain prediction performance across cortex (figure 5B) and ROIs (figure 5C). “

We additionally want to make you aware that the ten sets of features from the shuffled inputs originally used to make Figure 5 were misplaced, so we re-extracted them for this revision in order to do this analysis and subsequent analyses for Reviewer 2. Due to the stochasticity of the random shuffling, the numerical values of the feature maps – and subsequently the brain prediction

performance – are different. In order to be fully transparent and reproducible, we decided it would be best to update the main manuscript Figure 5 to reflect these newly extracted feature maps from the shuffled inputs. The overall pattern of the results nor our conclusions changed as a result of this. We reproduce the figure and caption below:

“Figure 5: Importance of temporal order in predicting fMRI responses.

A) Frame shuffling procedure: We predict the real fMRI responses using both the DNN activations of the original (unshuffled) frame order and the DNN activations of the randomly shuffled frame order. A difference in prediction accuracy between the DNN activations of the unshuffled and shuffled frames indicates the preservation of correct temporal order in the fMRI response. **B) Whole-brain prediction difference:** Difference in the correlation averaged over participants between the shuffled framed prediction accuracy and unshuffled frame prediction accuracy across the whole brain at different DNN layers (TSM model). **C) ROI-based prediction difference:** Difference in the correlation between the shuffled frame prediction accuracy and unshuffled frame prediction accuracy at different ROIs and DNN layers (TSM

model). A colored asterisk above a box plot indicates significant difference between the unshuffled and shuffled prediction accuracy at that DNN block (one sample two-sided t-test against a population mean of 0, **FDR correction across 22 ROIs x 4 blocks = 88 comparisons**, $p < 0.05$). The box plot encompasses the first and third data quartiles and the median (horizontal line). The whiskers extend to the minimum and maximum values within 1.5 times the interquartile range, and values falling outside that range are considered outliers (denoted by a diamond). The overlaid points show the value at each observation ($n=10$ for all ROIs except TOS ($n=8$) RSC ($n=9$)).”

Minor points

(1) Figure 4, captions: The authors write that ‘All videos are shown to both a DNN and a human.’ The authors may want to state which DNN they used – they do so in panel B, so I’m assuming they used the same DNN here, but why not make that explicit from the start.

We add the specific type of DNN used in Figure 4’s panel A text. It now reads:

“All videos are shown to both a TSM ResNet50 DNN and a human.”

(2) I appreciate the level of detail the authors provide to access the quality of the (f)MRI data. That said, some of the reported image quality metrics might not be known to the readers of Nature Communications, so some more guidance regarding the interpretation of these measures might be helpful.

We agree that more motivation and explanation behind the IQMs would benefit the readers. Interpretation that clearly links IQM values to quality of downstream analysis results (e.g., encoding model prediction) is often not possible. These IQMs measure properties before or during preprocessing such as gray-white matter contrasts, motion, SNR, and general quality measures of scans. Such measures are valuable to estimate data quality “close to the scanner” before any neural modeling (e.g., GLM) or analyses takes place. We believe the following univariate and multivariate reliability measures, which are post-GLM and much closer to the downstream analysis results, complement these IQM metrics well. The full MRIQC report is available in the dataset release, and we additionally point the readers to the supplementary material where all 12 IQMs are explained in greater detail and the link to the MRIQC website that provides details for all 112 IQMs.

We made the following changes to the manuscript under the section “(f)MRI image scans of high quality across subjects and task” to convey the above points:

“This yielded a comprehensive set of 44 functional and 68 structural image quality metrics (IQMs) (full report available), of which we present a representative set of six structural (Figure 2A) and six functional MRI metrics (Figure 2B) (see **Supplementary for more details on the IQMs and Supplementary Figure S1 for the resting state and functional localizer scans). In brief, since no set of metrics can completely describe data quality, the selection of the IQMs to display here focused on metrics especially relevant to functional or structural scans (e.g., **Temporal SNR for functional scans and Contrast to Noise ratio for structural scans**), metrics shared between functional and structural scans for a more cohesive set (e.g., **SNR and Full-Width Half Maximum Smoothness**), and metrics common across other literature and analysis**

packages to increase familiarity with readers (e.g., SNR, Framewise Displacement, AFNI Outlier Ratio, AFNI Quality Index)."

In the supplementary section "Structural and functional scan quality assessment", we add the following text:

"We present a representative subset of 6 IQMs to summarize the quality of our structural scans and another subset of 6 IQMs to summarize the quality of our functional scans (see MRIQC documentation for details on all 112 IQMs: <https://mriqc.readthedocs.io/en/latest/measures.html>)."

(3) Figure 4, captions, panel C: It did not become entirely clear to me why the difference in predictive performance was computed between block 1 and 4 – why these particular blocks? Can the authors motivate this choice?

We choose features from block 1 to reflect low-level feature processing and block 4 to reflect high-level feature processing. The difference in prediction performance between these two blocks summarizes the dominance of low-level or high-level features encoded within an ROI. We added relevant text and citations to the manuscript, reproduced below:

"As a proxy for low-level and high-level features, we extract the video features at early (Block 1) and late (Block 4) blocks of the DNN, respectively (Cichy, Khosla, et al., 2016; Kubilius et al., 2019; D. L. Yamins et al., 2013; D. L. K. Yamins et al., 2014). We then estimate the dominance of low-level and high-level feature processing across ROIs by computing the difference in predictivity between DNN Block 1 and DNN Block 4."

(4) Methods, page 6, line 129: It might help to inform the reader already here which of these labels were generated by humans, and which were generated by a DNN.

We ran additional human crowd-sourced experiments to record new scene and object labels. Now, all video labels (object, scene, action, text description, and spoken transcription) are generated by humans. The text has been updated accordingly:

"Revealing how the brain mediates visual event understanding benefits from detailed, human-labeled descriptions of a visual event. Thus, we used human crowd-sourced experiments to annotate each clip with five word-level scene, object, and action labels, five sentence-level text descriptions, one spoken transcription, one behavioral memorability score, and one memorability decay rate (Figure 1A).

Each category of metadata labels was collected in separate experiments. The possible scene, object, and action labels were sampled from the Places365 (Zhou et al., 2018), THINGS (Hebart et al., 2019, 2023), and Multi-Moments in Time (Monfort et al., 2022) datasets, respectively, for their carefully crafted label coverage and extensive overlap with computer science resources. Due to most videos clearly containing more than one object, the

annotators in the object label experiment were instructed to label three different objects for a total of 15 object labels per video. The 1,102 BMD stimuli cover 305 of 365 possible scene labels, 1,002 of 1,854 possible object labels, and 261 of 292 possible action labels. The sentence text descriptions (13.06 mean +/- 2.800 std words per sentence) were typed free-form by the annotator to detail salient interactions in the video and summarize the pertinent content. The spoken transcription, collected through free-form audio recordings and transcribed (audio not available due to privacy considerations), tends to be more verbose with additional emotional and linguistic subtleties present in speech but often not in text (26.72 mean +/- 17.55 std words per transcription).

In the Methods section under “Metadata”, we make the following changes when describing the object and scene labels:

“Metadata

Visual events consist of complex combinations of objects, locations, actions, and more. To capture the many dimensions of visual events, we characterized each video with a set of seven metadata categories: object labels, scene labels, action labels, text descriptions, a spoken transcription, a memorability score, and an index of memorability decay rate (Figure 1A). Five object, scene, action, and text description labels were collected for each stimulus to ensure comprehensive coverage (the five annotators’ labels reflect their unique interpretations) and form a group consensus (the five annotators’ labels can be used to converge on a single label). Labels for each metadata category were collected in different human crowd-sourced experiments. An annotator was allowed to label up to 20 different videos but no more than one label per video to encourage diversity of human annotators. While crowd-source workers were not restricted from participating in multiple metadata experiments, this was unlikely due to the experiments being collected at different times and the large population pool from which the crowd-source workers were drawn.

Object labels

For each video, we obtained at least 5 sets of up to three different object labels in a human crowd-sourced experiment on Prolific. Each annotator was instructed to select up to three different object labels visible in the video. They selected at least one object label, and if they believed no more objects were present in the video, they were allowed to select a “No more objects in the video” option up to two times. This option thus encouraged accurate labels and carried information on the density of objects in the video. Each object label was one of 1,854 possible labels from the THINGS dataset to encourage overlap with computational neuroscience work and leverage the additional THINGS metadata on each label (e.g., animacy, size, indoor/outdoor). The object label selections can be different or the same across annotators. One author manually reviewed the labels to ensure the labels assigned to the video were sensible (i.e., participants were not choosing labels at random).

Scene labels

For each video, we obtained at least 5 scene labels using a human crowd-sourced experiment on Prolific. Each of the five different annotators were instructed to select a scene label that

best describes the scene of the video. All scene labels came from the Places365 dataset (Zhou et al., 2018) for its broad scene coverage and overlap with computer vision resources. The scene label selections can be different or the same across videos. One author manually reviewed the labels to ensure the labels assigned to the video were sensible (i.e., participants were not choosing labels at random)."

(5) Methods, page 7, line 161: What was the reason to resample the data from a TR of 1.75 to 1 s, instead of collecting data at a TR of 1 s right away?

Thank you for this question. We do not acquire the data directly at a TR of 1 second due to physical limitations and desired scanner protocols. We calculated that an acquisition TR of 1.75 seconds would allow us to achieve our desired acquisition performance with theoretically higher SNR than a 1 second acquisition TR. Additionally, an acquisition TR of 1.75 seconds also allows us to densely sample BOLD responses at different delays with respect to stimulus onset. For example, since 1.75 does not evenly divide into the 4 second trial length, the BOLD response to a single video across three repetitions might be sampled at onset delays of [0.5, 2.25, 4, 5.75,...], [0, 1.75, 3.5, 5.25, ...], and [1, 2.75, 4.5, 6.25,...] seconds for the three repetitions. Such a jittered sampling during acquisition allows for more accurate interpolation since the true acquired time points are sampled at different delays relative to the stimulus onset. An acquisition TR of 1 second would result in sampling the BOLD signal at [0, 1, 2, 3, 4, ...] seconds every time, making the sampling less dense. Finally, the resampling from 1.75 second to 1 second TR allows the BOLD estimates to be time-locked to stimulus onset, as required by FIR modeling.

We added the following text under "(f)MRI data preprocessing, response modeling, and ROI definition" to make this reasoning clearer:

"We first processed the data using the standardized fMRIPrep tool (Esteban et al., 2019) to achieve reproducible and transparent results. During acquisition, we sampled fMRI data at a TR of 1.75s to achieve a densely sampled time course with respect to the onset of the stimulus (K. Kay et al., 2020); since 1.75 does not evenly divide into the 4 second trial length, the BOLD response to a single video across three repetitions might be sampled at onset delays of [0.5, 2.25, 4, 5.75,...], [0, 1.75, 3.5, 5.25, ...], and [1, 2.75, 4.5, 6.25,...] seconds for the three (or ten) repetitions. In the preprocessing we then temporally resampled the data from a TR of 1.75s to a TR of 1s in order to acquire BOLD responses exactly time-locked to the onset of each trial (stimulus presentation happens every 4s, i.e., a multiple of 1s, but not 1.75s). To accommodate variability in the hemodynamic responses at different locations in cortex, to resolve temporal structure of the stimuli, and to address response overlap from the rapid event-related design, we used Finite Impulse Response (FIR) basis functions."

(6) On page 14, line 301f, the authors write that they release the model to aid investigations of visual event understanding. Here might be a good location to state where the model can be found.

We updated the manuscript to include the corresponding github link. The sentence now reads:

"We release this model to aid investigations of visual event understanding (model available at: <https://github.com/pbw-Berwin/M4-pretrained>)."

(7) Figure 7: How similar are the models for scenes, actions and text caption?

Below are the spearman correlations between the scene, object, action, scene+object+action, and text description RDMs from the test set videos used in Figure 7:

Scene < object: 0.1808
Scene < action: 0.1133
Scene < scene+object+action: 0.6641
Scene < text description: 0.1969
object < action: 0.0884
object < scene+object+action: 0.6137
object < text description: 0.2618
action < scene+object+action: 0.5238
action < text description: 0.1560
scene+object+action < text description: 0.3145

The purpose of these 5 models is to convey increasing amounts of information content from individual labels (“scenes”, “objects”, “actions”) to the combination of individual labels (“scene+object+action”) to a full text description (“text description”). We expect the correlations between the individual label models to be lowest as they are capturing distinct pieces of information about the video. The correlations between the individual label models and the combination “scene+object+action” model should be highest, as this combination model explicitly encompasses the individual label models. We also expect and observe a high correlation between the “text description” model and “scene+object+action” model since both capture information regarding the video’s scene, action, and objects.

We include these model correlation results in the methods section, with the relevant text reproduced below:

“We compute the correlations (Spearman) between the metadata RDMs to measure the similarity of their information content. As expected, the correlations between the individual single-word object, scene, and action labels were generally lowest, since these labels highlight explicitly different components of the video. The correlations between the individual single-word object, scene, and action labels with the text descriptions were generally next highest, also expected because the text description likely contains information about each of the single-word labels plus extra information. The combined object+scene+action RDM with the text description RDM was higher still given even more overlapping information. The highest similarities were between the single-word object, scene, and action RDMs and the combined object+scene+action RDM because of explicitly overlapping information content (computed using the same embedding) but less extra information than the text description.

**[scene, object]: 0.1808
[scene, action] 0.1133
[scene, scene+object+action]: 0.6641
[scene, text description]: 0.1969
[object, action]: 0.0884
[object, scene+object+action]: 0.6137**

[object, text description]: 0.2618
[action, scene+object+action]: 0.5238
[action, text description]: 0.1560
[scene+object+action, text description]: 0.3145”

(8) Figure 7, panel A: At this point in the manuscript it was not clear to me yet what video metadata are. Also here, the authors may want to refer to the corresponding methods section.

Thank you for pointing out the confusion here. We have been using the terms “annotation” and “metadata” interchangeably. Thus, the video metadata, or video annotations, are the object, scene, action, text description, spoken transcription, memorability score, and memorability decay rate labels. We change the uses of the word “annotations” to “metadata” throughout the manuscript to be consistent and avoid confusion. We also edit Figure 7 (now panel B, not A) to be more explicit as to what the metadata encompasses. We additionally point readers to the corresponding methods section for more details on the metadata collection in the text:

“... The combined objects+scenes+actions description concatenates the individual object, scene, and action label information to serve as an intermediate between the word-level and sentence-level descriptions (see Methods section “Metadata” for details on the metadata collection).”

(9) Results, page 21, line 447f, RSA: here I was wondering how the neural RDM was constructed. The authors may want to provide the reader with a link to the corresponding methods section.

The neural RDMs for each voxel in a brain mask shared across all subjects were constructed using a searchlight method (Kriegeskorte et al., 2006). Specifically, a spherical searchlight with a four voxel radius centered at voxel v indexed the response profiles of all voxels (n_{voxels}) within the searchlight. This indexing resulted in a matrix of size $102 \times n_{\text{voxels}}$ for the test set, denoting the responses to the 102 videos at each of the voxels in the searchlight. The pairwise correlation (Pearson) between video i and video j 's response vector (size $1 \times n_{\text{voxels}}$) was computed, and the distance ($1 - \text{Pearson correlation}$) was entered into the RDM at index (i, j) . This process repeated for each voxel and subject separately. The RDMs at each voxel were of size 102×102 ($n_{\text{videos}} \times n_{\text{videos}}$) and symmetrical across the diagonal. Subsequent correlations with metadata model RDMs only used the lower (or equivalently upper) triangle of the RDM, not including the diagonal.

We make the following additions to the main text and Figure 7 to make the neural RDM construction more clear and additionally point readers to the methods section:

“We compute a neural representational dissimilarity matrix (RDM) at each voxel by computing the pairwise distances between vector embeddings obtained from a searchlight procedure (four voxel radius) (Kriegeskorte et al., 2006) (Figure 7A). We compute one RDM for each metadata category by calculating the pairwise distances between vector embeddings obtained by feeding the text-based labels through natural language processing models (FastText, Bojanowski et al., 2017, for the single-word **object, scene, and action labels, and Sentence-BERT, Reimers & Gurevych, 2019, for sentence-level text descriptions) (Figure 7B). We use representational**

similarity analysis (RSA) (Kriegeskorte, 2008) to **correlate the metadata representations (Figure 7C) with neural representations to measure how similarly the different metadata descriptions are reflected in brain activity of dynamic videos (see methods section “Metadata RSA Analysis Procedure” for more information).**”

(10) Methods, Stimuli, page 27, line 577f: How was the semantic overlap between a pair of videos determined?

Semantic overlap between pairs of videos was determined by manual inspection. In the supplementary (Figure S8, reproduced below), we show a t-sne plot of the averaged sentence text description embeddings between the train (blue) and test (orange) videos. The lack of clustering in either the train or test videos further suggests the semantic coverage of the BMD stimuli is equally diverse in the train and test sets.

We make the following edits to the methods section “Stimuli” for clarification:

“Specifically, the testing set videos were chosen randomly from the 1,102 videos, and then checked manually to ensure no semantic overlap, in terms of objects plus actions, occurred between any pair of testing set videos. If semantic overlap was found between a pair of videos as determined by an author, one of these videos was swapped with a video randomly selected from the pool of remaining videos and incorporated into the testing set.”

We additionally include this point about manual inspection in the caption of Supplementary Figure S8 below:

Figure S8: Distributions of stimuli metadata between training and testing sets

The training and testing sets consist of 1,000 and 102 different videos, respectively. An author manually inspected the pairs of testing set videos to ensure no high-level semantic overlap, in terms of objects and actions. **A) Object, scene, and action label frequency of occurrence:** The barplot depicts the frequency of occurrence, (between 0 and 1) of, from left to right, the single-word object, scene, and action labels of the 1,102 video stimuli used in the BOLD Moments Dataset. The frequency bars for each label are separated by training (blue) and testing (orange) splits to show their similar frequency of distributions. **B) Text description and spoken transcription t-sne distances.** The scatterplot shows the t-sne components (n=2 components, perplexity=10, number of iterations=1000) of each text description or spoken transcription embedding. The 6 sentence descriptions per video (5 text descriptions and 1 spoken transcription) serve as a useful proxy for the video's content. The t-sne plot shows the training and testing set stimuli cover similar spaces of video content. **C) Memorability distribution.** The distribution of the memorability scores and memorability decay rates (1 per video) between the training and testing splits are highly similar and approximately normal. Note that the positive memorability decay rates, while theoretically implausible, reflect the true experimental results detailed in the Memento10k dataset. Users may want to set positive values to 0 depending on the analysis.

(11) Methods, Functional localizers, page 27, line 597f: It would be helpful to provide the reader with more details about these videos. Can the authors show some example snapshots? Moreover, I was wondering whether the authors can provide a justification regarding the use of videos rather than static images, in particular for stimulus categories such as objects and scenes.

Subjects freely viewed colored, naturalistic blocks of video. Each video block was 18 seconds long, composed of showing 6 3-second videos of the same category back-to-back. We also release the localizer videos with the rest of our data. Please see Figure 1C for snapshots of the videos used in the localizer experiment.

We chose to use a video-based, as opposed to an image-based, localizer because (1) it more closely matches the short video stimuli used in the main experiment, (2) video-based localizers have been shown to be more effective in activating ROIs (Fox et al., 2009), and (3) this particular video-based localizer has been successfully used in prior studies to localize numerous ROIs, including object and scene ROIs (Julian et al., 2012; Lafer-Sousa et al., 2016; Pitcher et al., 2011; Ratan Murty et al., 2021).

We edited the paragraph to include a more detail on the videos, reproduced below:

“Subjects freely viewed colored, naturalistic videos (18s length, composed of 6 3-second videos) corresponding to one of five categories (faces, bodies, scenes, objects, and scrambled objects) in order to functionally localize each subject’s category selective regions (Julian et al., 2012; Lafer-Sousa et al., 2016; Pitcher et al., 2011; Ratan Murty et al., 2021). Subjects viewed the videos freely and performed a one-back vigilance task to ensure attention to the task (videos are available with the dataset release).”

(12) Related to the previous point: Was there a fixation cross provided during the functional localizer?

There was no fixation cross provided during the functional localizer. The subjects freely viewed the videos according to established protocols from previous work using this dynamic functional localizer (Julian et al., 2012; Lafer-Sousa et al., 2016; Pitcher et al., 2011; Ratan Murty et al., 2021). See point 11 right above for the changes to the manuscript mentioning this.

(13) Methods, Main experiment, page 28, line 626f: How were the null events modelled in the analysis?

Thank you for pointing out this oversight. We address this comment in our response to your comment #25. In the subsequent section “General linear model” and subsection “Main experiment”, we add the sentence:

“The stimulus presentation trials, but not the null response trials, were included as regressors of interest.”

(14) Methods, Main experiment, page 28, line 632f: I don't fully get the meaning of the statement 'with the restriction of excluding successive repetitions of test runs within one session'. Could the authors clarify?

Within a main experimental scanning session (sessions 2-4), we showed the participant a total of 13 runs – 10 training runs and 3 testing runs composed of videos only from the training and testing sets, respectively. The order of these runs was randomized except for the constraint that two testing runs cannot be shown back-to-back. This is what “with the restriction of excluding successive repetitions of test runs within one session” refers to. This purpose of this restriction is to increase the number of videos between repetitions to reduce potential memory effects. The paragraph now reads:

“The testing and training set videos were presented within training and testing runs, where each test run consisted of 113 trials and each train run consisted of 100 trials. The order of these runs was randomized except for the constraint that two testing runs cannot be shown back-to-back in the same session in order to reduce potential memory effects caused by the same stimulus being presented within a short period of time. Each session contained 3 test runs and 10 training runs and lasted approximately 100 minutes.”

(15) Methods, Main experiment and functional localizer: I did not find any information regarding a fixation period at the beginning and end of each run – is this simply an omission in the text, or was no such fixation period included?

Thank you for the comment, this indeed was an omission in the text and should be included. Each functional localize run began and ended with an 18 second fixation block consisting of a blank gray screen (Figure 1C). The ending fixation also contained an additional 19 seconds of gray screen that was not used in subsequent modeling. The main experimental runs began with 4 seconds of fixation with a fixation cross before the first stimulus was presented. The main experimental runs ended with 13 or 12.5 seconds (for test and train runs, respectively) of fixation after the last stimulus trial ended. The 0.5 second difference in fixation period duration between the test and train runs is due to the runs' different number of trials (113 and 100 trials) and the 1.75 second acquisition TR not being an even factor of the 4 second trial length. We added the following information to the methods section for the functional localizer:

“Subject accuracy on the one-back task was 0.941 +/- 0.011 (mean ± SD). Each run began and ended with a 18 second fixation (null) block. The end of the run contained an additional 19 seconds of gray screen (not modeled in the GLM) for a total duration of 268 volumes.”

We added the following information to the methods section for the main experiment:

“Each session contained 3 test runs and 10 training runs and lasted approximately 100 minutes. Each run began with 4 seconds of fixation and ended with 13 seconds (testing runs) or 12.5 seconds (training runs) of fixation. The 0.5 second difference of the duration of the ending fixation period between the testing and training runs is due to their different number of trials (113 and 100) and the acquisition TR of 1.75 seconds not being an even factor the 4 second trial length. In total, testing and training runs consisted of 268 and 238 volumes, respectively.”

We additionally added information to the total duration of the resting state runs:

“Within each run, participants were instructed to keep their eyes closed, to not think of anything specific, but to remain awake. The duration of resting state runs is 212 volumes.”

(16) Page 29, Metadata, line 638: Do the five annotations refer to each stimulus or each rater (or something else)?

The five annotations (metadata) refer to five different raters assigning labels for a stimulus. We agree the language we use is confusing. As described in response to your above “minor comment” R1.8, we make consistent our usage of the word “metadata” to increase clarity. The paragraph you note now reads:

“Five object, scene, action, and text descriptions labels were collected for each stimulus to ensure comprehensive coverage and form a group consensus.”

(17) Related to the previous point, on line 640, the authors write that the five annotations were collected to ensure comprehensive coverage, and to form a group consensus. I might have misunderstood, but this seems to be a bit in conflict with the statement on line 645f, which states that the authors did not enforce that all 5 object labels describe the same object.

We assume there is only one primary interpretation of each BMD video. Thus, each scene label, action label, and text description will be describing the same concept. Taking an example of a video of someone skiing, among the five human crowd-source workers some may describe the scene as a “snowfield” while others may describe the scene as a “mountain”, but there is no ambiguity about the concept itself – no one will mistake it for a “ocean”. By collecting five scenes from five different human raters per video, we successfully capture a “comprehensive coverage” of different – but equally valid – labels. If a researcher wishes to assign simply one label per video, they have the option to use the five labels for an accurate “group consensus” – they may take the mode of the five labels, top 3 most similar, or some other means.

In contrast to scene and action labels, a video often contains multiple distinct objects and we cannot assume each crowd-source human worker would choose to label the same object. To address this nuance for the object labels (new to this revision), each of the five crowd-source human workers labelled three visually distinct objects in the video. In this way we can better obtain both “comprehensive coverage” and a “group consensus”, depending on the researcher’s analysis.

We make this point clearer in the manuscript with the relevant text reproduced below. In the main text under section “Semantic and behavior metadata on visual events”, we write:

Due to most videos clearly containing more than one object, the annotators in the object label experiment were instructed to label three different objects for a total of 15 object labels per video.

In the methods section under “Metadata”, we add:

“Five object, scene, action, and text description labels were collected for each stimulus to ensure comprehensive coverage (the five annotators’ labels reflect their unique interpretations) and form a group consensus (the five annotators’ labels can be used to converge on a single label). Labels for each metadata category were collected in different human crowd-sourced experiments. An annotator was allowed to label up to 20 different videos but no more than one label per video to encourage a diverse sampling of human annotators within videos and throughout the stimulus set. While crowd-source workers were not restricted from participating in multiple metadata experiments, this was unlikely due to the experiments being collected at different times and the large population pool from which the crowd-source workers were drawn.”

(18) Page 29, Metadata, Action labels: Why did the authors select one action label that best described the video, but up to five different object labels (see comment above)? The authors argue that this is to reflect the participant’s limited ability to perceive every day action due to limited stimulus presentation time and their central fixation. Why would this apply to actions but not to objects?

Thank you for the opportunity to clarify. We ask each of the five human crowd-source workers to select only one action label that best describes the video because we believe the three second video length is short enough to capture only one action in series. Thus, the vast majority of BMD videos can be described by one primary action. We concede, however, that a complex video may have multiple actions taking place in parallel. In this edge case, the five labels would likely capture the multiple different actions. The five labels can also be used to form a “group consensus” (e.g., mode, top 3 most similar vector embeddings, etc.) to identify the most salient action.

Because most videos contain multiple, easily identifiable objects, we asked each of the five crowd-source human workers to label 3 different objects. Using this experimental design to collect action labels would multiply costs for little benefit.

We argue that limited presentation repetitions, short 3 second stimulus duration, and central fixation limits the identification multiple actions and not objects because actions must unfold over time while objects do not. Actions are a more abstract composition of movement, while objects are visually distinct parts of a video with easily identifiable boundaries.

In summary, we are confident that asking each of the five crowd-source workers to label each video with one action label and 3 distinct objects was a cost-effective and accurate method to collect this metadata.

We clarify this point in the text, reproduced below in the methods section under “Action labels”:

“The 5 action labels were selected by workers on the crowd-sourcing platform Prolific. We restricted the possible action labels to be from one of 292 possible action labels that broadly encompass meaningful human actions (Monfort et al., 2022). The participant viewed these 292 possible action labels, watched the video, and selected one action label that best described the video. Each of the 5 action labels per video were produced by different participants. Given that an action, by definition, unfolds across time and the BMD stimuli have a short 3 second duration, the majority of the videos contain one primary action. Additionally, the limited number of stimulus repetitions and the central fixation further limits the fMRI scanner participant’s ability to perceive a video’s small, inconsequential actions if they are present. In the edge case that a complex video captures multiple salient actions, the five annotations can capture these multiple actions. Note that annotators labeled up to 3 objects in a video (described above) because objects, unlike actions, have clear visual boundaries, often occur in multiple instances in a video, and have no temporal dimension. Two authors manually reviewed the labels to ensure the labels assigned to the video were sensible (i.e., participants were not choosing labels at random).”

(19) Page 30, line 686f, Memorability score and decay rate: how is the memorability decay rate defined?

(Newman et al., 2020) used a crowd-sourced human behavioral experiment where the participant viewed videos in a continuous stream and were asked if they had seen this image before. Under this paradigm, the memorability score is “the fraction of times that [a] repeated video was correctly detected” after controlling for the lag in videos (i.e., the number of videos in between consecutive viewings of the same video). A memorability decay rate describes how a video’s memorability score changes over time (i.e., lags). Detailed in the same report, the decay rate is an output from an Inflated 3D (I3D) Convolutional Neural Network, called SemanticMemNet. The network is trained to predict a video’s memorability score and text caption from the video frames and optical flow. The model then regresses a memorability decay rate.

We point readers to (Newman et al., 2020) for detailed descriptions of the memorability decay rate’s derivation, and we add text to the methods section to describe the relationship between memorability score and memorability decay rate in more detail:

“The participant’s responses were then used to calculate a video’s memorability score from 0 (no recall) to 1 (perfect recall) and memorability decay rate from 0 (no decay) to $-\infty$ (instantaneous decay). Specifically, a video’s memorability score is the fraction of correct identifications in the memory game normalized to a lag of 80 videos in between consecutive presentations. A video’s memorability decay rate describes how a video’s

memorability score changes over different lags. The memorability decay rate was regressed from the output of SemanticMemNet (Newman et al., 2020), an Inflated 3D (I3D) Convolutional Neural Network trained to predict a video's memorability score and text caption from the video frames and optical flow. A video's memorability score (m) and decay rate (α) can be used to predict its memorability at any time lag t (the number of videos in between the first and second presentation) according to the following equation (Newman et al., 2020):

$$m_t = m_T + \alpha(t - T)$$

where T is a set lag of 80. The memorability scores given in the stimuli annotations were computed at a lag of $t=80$."

(20) Page 32, line 730f: What makes this template asymmetrical?

The asymmetry in the template is the result of the natural left-right hemisphere asymmetry in the template of the human brain used. The 2009c ICBM MNI152 templates are released in both a symmetric and asymmetric version. Both templates are essentially averages over 152 subjects, but the asymmetric version preserves the left-right hemisphere anatomical asymmetry naturally seen in the human brain. The symmetric version undergoes additional processing steps in order to achieve left-right hemisphere anatomical symmetry. The asymmetric version is the default template in many fMRI analysis pipelines, including fMRIPrep used here. See (Fonov et al., 2009) for more details on the construction of these templates.

We add in the methods under the section "Preprocessing":

"All data from all sessions was then preprocessed using the standardized fMRIPrep preprocessing pipeline. For all results unless stated otherwise, we use the standard MNI152NLin2009cAsym volumetric space for its frequent use in other work and preservation of natural left-right asymmetries in the brain. As recommended by fMRIPrep to increase transparency and reproducibility in MRI preprocessing, we copy their generated preprocessing text in its entirety below:"

(21) Page 32, line 735f: The authors state that the fieldmap was estimated on the basis of a phase-difference map using a dual-echo GRE sequence, but I could not find such a sequence among the sequences that were listed in the section 'fMRI data acquisition' on page 31.

Thank you for pointing out this oversight. We added the following sentence to the "fMRI data acquisition" section describing the fieldmap sequence:

"Dual echo fieldmaps (TR = 636 ms, TE1 = 5.72 ms, TE2 = 8.18 ms, flip angle = 60°, FOV read = 190 mm, FOV phase = 100%, bandwidth = 260 Hz/Px, resolution = 2.5 × 2.5 × 2.5 mm, slice gap = 10%, slices = 54, ascending interleaved acquisition) were acquired at the beginning of every session to post-hoc correct for spatial distortion of functional scans induced by magnetic field inhomogeneities."

(22) Related to the previous point, the authors state that the fieldmap was co-registered to the target EPI reference run. Which run did they use as the target?

The target EPI reference volume was not calculated from a specific run, but was calculated using fMRIPrep's custom methodology, explained in detail here (<https://fmripred.org/en/stable/workflows.html#bold-ref>). The relevant text in this documentation states, "This workflow estimates a reference image for a BOLD series. If a single-band reference ("sbref") image associated with the BOLD series is available, then it is used directly. If not, a reference image is estimated from the BOLD series as follows: When T1-saturation effects ("dummy scans" or non-steady state volumes) are detected, they are averaged and used as reference due to their superior tissue contrast. Otherwise, a median of motion corrected subset of volumes is used."

In our case, the latter methodology applies - the reference image was estimated from the BOLD series using averaged T1-saturation effects. This reference EPI is the one mentioned in the fMRIPrep boilerplate text that you cite, reproduced here, "The *fieldmap* was then co-registered to the target EPI (echo-planar imaging) reference run and converted to a displacements field map (amenable to registration tools such as ANTs) with FSL's *fugue* and other *SDCflows* tools."

(23) Page 33, line 759: Please define DVARS.

In DVARS, the "D" refers to the temporal derivative of timecourses, and "VARS" refers to the root mean square variance over voxels. You can see the code implementation and helper comments here:

<https://github.com/nipy/nipype/blob/b066807402ee07d908ce4f1b24ca69fef6b91809/nipype/algorithms/confounds.py#L1009C16-L1009C16>

fMRIPrep asks its users to faithfully copy their automatically generated boilerplate text in their manuscript so the preprocessing details can be completely transparent and reproducible. Hence, we hesitate to edit the section of fMRIPrep's boilerplate text that you are referencing here, but we believe the boilerplate text (reproduced below) does a sufficient job of pointing readers to the appropriate reference for more information.

"FD and DVARS are calculated for each functional run, both using their implementations in *Nipype* (following the definitions by Power et al., 2014)."

(24) Page 33, line 765f: How was the set of physiological regressors extracted?

Physiological regressors were extracted by masking the timeseries in the cerebrospinal fluid (CF) and white matter (WM) ROIs. The theory behind the CompCor component-based noise correction used in fMRIPrep is that signals in the CF and WM are non-neuronal and dominated by physiological noise, such as cardiac or respiratory rhythms (Behzadi et al., 2007). Identifying similarities between the gray matter signal (which contains neuronal signal of interest plus physiological noise) with the CF and WM signal will then characterize physiological signals in the gray matter.

Similar to the above comment, we hesitate to edit the section of fMRIPrep's boilerplate text that this comment is referencing. We believe the boilerplate text (reproduced below) sufficiently points readers to the relevant Behzadi and colleagues (2007) publication for more information.

“The three global signals are extracted within the CSF, the WM, and the whole-brain masks. Additionally, a set of physiological regressors were extracted to allow for component-based noise correction (*CompCor*, Behzadi et al., 2007).”

(25) Page 34, line 813f: The authors may want to state more explicitly what were their regressors of interest.

Thank you for the comment, the regressors of interest are important to make explicit. The relevant methods text for the Functional Localizer GLM now reads:

“To model the hemodynamic response to the localizer videos, the preprocessed fMRI data, video, and fixation baseline onsets and durations were included in a general linear model (GLM). The fixation and five category (faces, objects, scenes, bodies, and scrambled objects) blocks were included as regressors of interest. Motion and run regressors were included as regressors of no interest. All regressors were convolved with a hemodynamic response function (canonical HRF) to calculate beta estimates.”

The relevant methods text for the Main Experimental GLM now reads:

“We modeled the BOLD response with respect to each video onset from 1 to 9 seconds in 1 second steps (corresponding to the resolution of the resampled time series). The stimulus presentation trials, but not the null response trials, were included as regressors of interest. Within this time interval the voxel-wise time course of activation was high-pass filtered (removing signal with $f < 1/128$ Hz) and serial correlations due to aliased biorhythms or unmodelled neuronal activity were accounted for using an autoregressive AR(1) model.”

(26) Page 37, line 884f: The reader might be curious to be provided with a few more details regarding how the authors used the Glasser atlas, given that there exist a number of different ways in which one can extract them.

The Glasser atlas registered to a template MNI volume was downloaded and then resampled to the BMD functional voxel resolution with nearest neighbor interpolation using SPM 12's *reslice* function. The resampled Glasser atlas was then used to index the voxels in the BMD data belonging to the corresponding ROI. We update the manuscript to read:

“Atlas versions registered in MNI volume space were downloaded and resampled to BMD's functional voxel resolution with nearest neighbor interpolation using SPM 12's “reslice” function. The resampled atlases were then used to index ROIs in the BMD volume data.”

(27) Page 37, line 909f: the raw beta values of what?

We mean to say that we use the FIR beta estimates to the stimulus presentations before any additional processing, such as z-scoring or averaging. We clarify the manuscript to read:

“For each subject separately, the raw FIR beta estimates to the video stimuli at each voxel were z-scored across stimuli, averaged across TRs 5-9, and averaged across stimuli repetitions to result in a (n_stimuli x 1) vector of beta values.”

(28) Page 38, line 918f: Since not all readers might be familiar with voxel-wise upper and lower noise ceilings, could the authors provide a reference?

We agree that the upper and lower noise ceiling procedure would benefit from references. The first sentence in the methods section under “Univariate split-half reliability analysis” now reads:

“To select voxels with high signal-to-noise ratio, we defined a selection criteria based on split-half trial reliability (Lage-Castellanos et al., 2019; Schrimpf, Kubilius, Hong, et al., 2020).”

When introducing the upper and lower noise ceiling estimates in the methods section under “Multivariate searchlight-based reliability analysis”, we add the sentence:

“The upper and lower noise ceilings of the multivariate searchlight results were computed (Carlin & Kriegeskorte, 2017; Nili et al., 2014).”

(29) Page 38, line 933f: what is LSEP loss?

LSEP loss stands for log-sum-exp pairwise loss. It is proposed in (Li et al., 2017) and modified in (Monfort et al., 2022) as an appropriate loss function to train on multi-label and class imbalanced datasets, such as actions.

We add the following text to the methods section “Action recognition TSM ResNet50 model training”:

“We trained our model on the M4 (Multi-Moments minus Memento) training dataset for 120 epochs by using LSEP (log-sum-exp pairwise) loss (Monfort et al., 2022). LSEP loss was first proposed in (Li et al., 2017) and modified in (Monfort et al., 2022) as an appropriate loss function to train on multi-label and class imbalanced datasets, such as actions.”

(30) Page 38, line 937f: How were some of the details of the analysis determined (e.g. the split of the input video into 8 segments, sampling 1 frame from each segment; initialization of the learning rate and the weight decay with values of 0.02 and 0.0001, respectively)?

The hyperparameters for the TSM ResNet50 training, including but not limited to the sampling of the input video, learning rate, and weight decay, were chosen to closely follow the original TSM paper (Section 5.1 in (Lin et al., 2019) <https://arxiv.org/pdf/1811.08383.pdf>). We scale up the learning rate to 0.02 (0.01 in the TSM paper) because our batch size is 128, which is 2x of the batch size 64 in TSM paper. These hyperparameters are well within established practices for DNN training. We made the following addition to the manuscript methods section, “Action recognition TSM ResNet50 model training”:

“We chose the model hyperparameters to closely follow those used in Lin and colleagues (2019). Specifically, during the training phase, our model split the input video into 8 segments and sampled 1 frame from each segment. We used SGD optimizer to optimize our model. The learning rate followed the cosine learning rate schedule and was initialized as 0.02. The weight decay was set to be 0.0001 and the batch size 128.”

(31) Page 39, line 951: Please provide a reference for the encoding model procedure.

We add references for the encoding model procedure in the first sentence of the methods section, “DNN block to cortex correspondence procedure.” The sentence now reads:

“We used an encoding model procedure to quantify the correspondence between DNN Blocks and regions of cortex (Kriegeskorte & Douglas, 2019; Naselaris et al., 2011; Schrimpf, Kubilius, Hong, et al., 2020).”

(32) Page 39, line 956: The statement regarding PCA on the DNN activations would profit from a few more details. As an example, what does the N refer to – number of components?

We added more details to the PCA procedure and rearranged text in the paragraph for clarity. The paragraph now reads:

“We ran inference on the TSM ResNet50 model using the 1,102 videos used in the fMRI experiment and extracted the activations for each video. The activations for a given block were extracted after the nonlinearity. We then used an encoding model procedure. In detail, we standardized the DNN activations (using the mean and standard deviation of the training videos) and performed principal component analysis (PCA) using the top 100 components of the DNN activations to ensure fair comparison of activations of different embedding sizes. The PCA procedure was fit on the 1,000 training set video activations and applied to the 102 testing set video activations. For each voxel v , we fit a linear model from the training set DNN activations (size $(n_{\text{training_videos}} \times n_{\text{PCA_components}})$) to the training set fMRI responses (beta values z-scored across stimuli and averaged over TRs 5-9) averaged over the 3 trial repetitions (size $(n_{\text{training_videos}} \times n_{\text{voxels}})$). We then predicted testing set voxel responses (size $(n_{\text{testing_videos}} \times n_{\text{voxels}})$) by applying the linear fit on the testing set DNN activations (size $(n_{\text{testing_videos}} \times n_{\text{PCA_components}})$). We evaluated the performance of the prediction by correlating (Pearson) the predicted testing set voxel responses with the true testing set fMRI responses of each of the 10 testing set repetitions (size $(n_{\text{testing}} \times n_{\text{voxels}})$). The final performance of the prediction is the average correlation of the 10 repetitions. The noise-normalized correlation is this 10-repetition average Pearson correlation divided by the voxel’s split-half correlation value (the Pearson correlation value before Spearman-Brown). We only predicted the values of the voxels that met the split-half reliability criteria, as described in the “Univariate Split-Half Reliability Analysis” methods section above, in order to model meaningful signal.”

We additionally realized that this encoding procedure originally used 500 PCA components as opposed to 100 components used for analyses presented in Figure 5 and Figure 6. We fixed this miscommunication by re-running the same analysis in Figure 4 with 100 PCA components. The number of components generally do not affect the trend of the results, but it can affect magnitude. Hence, we see in Figure 4C that V1v was no longer statistically significant, and hV4 and V3ab are now significant, but the hierarchical results remain. We reproduce Figure 4 and Supplementary Figure S7 (see panel A for the TSM ResNet50 model, the other models in panel B and C were added in response to a comment from Reviewer 2 comment #2) with captions below for convenience:

Figure 4: Evaluation of biologically-similar video-based encoding model

(A) Voxelwise encoding model procedure: All videos are shown to both a **TSM ResNet50 DNN** and a human. Training set video embeddings are extracted from a block b of the DNN and used to learn a voxelwise mapping function to the human responses. This mapping is then applied to the testing set video embeddings to predict the brain response at each voxel. **(B) Whole-brain encoding accuracy across blocks:** We use the encoding model procedure with each of the four blocks of a TSM ResNet50 model trained to recognize actions in videos to predict the neural response at each voxel in the whole brain. The brain figures show the subject-average noise-normalized predictive correlation (divided by the voxel's upper noise ceiling) at each voxel. **(C) ROI-based encoding accuracy difference:** Difference in predictive performance between block 1 and block 4 at each of the 22 ROIs. Predictive performance at

each voxel is measured as the noise-normalized correlation between the brain responses and the predicted responses, averaged over all reliable voxels in each ROI. Significant ROIs are denoted with an asterisk and a color (blue for Block 1, red for Block 4, gray is not significant) corresponding to the significant layer (one sample two-sided t-test against a population mean of 0, Bonferroni corrected across 22 ROIs, $p < 0.05$). The box plot encompasses the first and third data quartiles and the median (horizontal line). The whiskers extend to the minimum and maximum values within 1.5 times the interquartile range, and values falling outside that range are considered outliers (denoted by a diamond). The overlaid points show the value at each observation ($n=10$ for all ROIs except TOS ($n=8$) RSC ($n=9$)).

Figure S7: Encoding model performance on BMD
A) TSM ResNet50 trained on M4: Features were extracted after block's 1 (blue), 2 (orange), 3 (green), and 4 (red) in the ResNet 50 architecture. B) TSM MobileNetV2 trained on Kinetics-400: Features were extracted after the first bottleneck layer (blue),

third bottleneck layer (orange), sixth bottleneck layer (green), and last 2D convolutional layer before the average pool (red) in the MobileNetV2 architecture. C) TimeSformer S+T trained on HowTo100M: Of the model's twelve layers, features were extracted after the first (blue), fourth (orange), eighth (green), and twelfth (red) layers. The box plot on the left side in each panel shows the noise-normalized predictivity of four of each architecture's features at each of the 22 ROIs. The features were extracted at early (blue), intermediate (orange and green), and late (red) processing stages in each architecture to capture increasingly high-level degrees of transformations. The box plot on the right side in each panel shows the brain prediction difference between each architecture's latest and earliest layers for each subject and ROI. For the box plots on the right, a blue or red colored box plot denotes a significant difference in correlations from 0 ($p < 0.05$, two-sided one-sample t-test, Bonferroni corrected for $n=22$ comparisons), and gray denotes no significance. The box plots encompass the first and third data quartiles and the median (horizontal line). The whiskers extend to the minimum and maximum values within 1.5 times the interquartile range, and values falling outside that range are considered outliers (denoted by a diamond). The overlaid points show the value at each observation ($n=10$ for all ROIs except TOS ($n=8$) RSC ($n=9$)).

(33) Page 40, line 993f: What is the justification to extract the top 100 components for each block?

The top 100 components were extracted from each block's activations to reduce the number of features, make all blocks have the same number of features, and capture a sufficient amount of variance. We make the following edits to the manuscript:

"Then we performed PCA to extract the top 100 components of each block's activations to reduce the number of features and equate their dimensionality while preserving the variance in the activations."

(34) Page 40, line 997f: What was the reasoning behind averaging beta values across TRs 5-9 (i.e., why these specific TRs)?

TRs 5-9 (representing seconds 5-9 after stimulus onset) reflect the peak of a typical BOLD response. During FIR modeling, modeling TRs 5-9 together empirically gave reasonable estimates.

We add the following text to section you mention:

"As brain responses, we used beta values z-scored across stimuli and averaged over TRs 5-9 (TRs 5-9 reflect the peak of a typical BOLD response)."

(35) Page 41, Encoding and variance partitioning analysis procedure: I found the entire first paragraph of this section hard to understand. It might be easier to follow if the authors provided the reader with a brief statement upfront regarding the purpose of this analysis. The authors also may want to double-check the readability of this paragraph (as an example, what exactly do the authors mean by 'feature maps of shape' (line 1019f)?)

We edited this section to make it more readable and used terminology consistent with previous sections. The phrase “feature maps of shape” that you point out is intended to be read together with its following parenthetical “(1000 training videos × 100 features × 2 video seconds).” We edit the text to avoid this confusing use of parentheses. The section now reads:

“We use an encoding model and variance partitioning analysis to identify any unique variance explained by the first and third video seconds in the brain activity. In this way, we measure if the brain activity captures temporal content. The encoding algorithm involved two steps. In the first step, **we** fed the first and third video second frames of the 1000 training and 102 testing videos to an AlexNet architecture (Krizhevsky et al., 2017) pre-trained on the ILSVRC-2012 image classification challenge (Russakovsky et al., 2015), and we extracted the corresponding **activations** at each layer. We then applied the following operations to the **activations** of both video seconds (first and third), independently: we appended the feature maps of all layers, averaged them across frames, standardized them (using the mean and standard deviation of the training videos feature maps) and downsampled them to 100 components through principal components analysis (PCA) (computed on the training videos feature maps). This resulted in the training **activations** of size **1000 × 100 × 2** and testing **activations** of size **102 × 100 × 2 (number of videos × features × video seconds)**. In the second step, we linearly mapped the stimuli videos feature space onto voxel space, thus predicting the fMRI responses to videos. For each combination of **subjects (10), fMRI voxels (N), and fMRI TRs (9)**, we trained the weights of a linear **regression model** to predict the fMRI training data (averaged over the three **repetitions**) using the training **activations** of both video seconds independently as predictors, and then multiplied the learned weights with the test **activations**. This resulted in two synthetic fMRI test data instances (**size 10 subjects × 102 test videos × 9 fMRI TRs × N fMRI voxels**), one for each video second.

To test our hypothesis **that the brain activity captures information unique to the first and third video seconds**, we ran a variance partitioning analysis between the biological fMRI test data and the two instances of synthetic fMRI test data. At each subject, TR, and voxel we ran a searchlight (Kriegeskorte et al., 2006) to calculate the portion of the biological fMRI test data (averaged over the ten repeats) uniquely explained by, respectively, the synthetic fMRI test data of the first or third video seconds. **The unique biological fMRI variance explained by the first/third video second fMRI synthetic data consisted in the adjusted R² score of a linear regression trained to predict the biological fMRI data using both synthetic fMRI data instances as predictors, minus the adjusted R² score of a linear regression trained to predict the biological fMRI data using only the third/first synthetic fMRI data instance as predictor.** We then observed at which TRs the unique variance explained by the two versions of synthetic test data peaked, and subtracted the peak TR of the first video second synthetic data from the peak TR of the third video second synthetic data. Next, we created subject wise binary whole brain masks with ones in voxels that show TR peak differences in the range 1 to 3 and zeros elsewhere, summed the binary masks across subjects, and performed a binomial test with FDR correction to remove the non-significant voxels.

The variance partitioning analysis for the ROIs was similar but performed on the reliable (split-half reliability $p < 0.05$) voxels within each ROI. Again, this results in time courses that reveal how well the synthetic fMRI test data from either the first or third video second explains the real fMRI data at each of the nine TRs. To quantify this difference, we again subtracted the peak TRs of the first and third video second synthetic data.”

(36) Related to the previous point, on page 41, line 1029, the authors write ‘To test our hypothesis’, but at this point I really was not clear about what that might be.

We edited the section to be clearer about the purpose of the analysis and the hypothesis. We begin the section by introducing the purpose:

“We use an encoding model and variance partitioning analysis to identify any unique variance explained by the first and third video seconds in the brain activity. In this way, we measure if the brain activity captures temporal content.”

We also edit the first sentence of the paragraph you cite here to be more explicit in the hypothesis. It now reads:

“To test our hypothesis that the brain activity captures information unique to the first and third video seconds, we ran a variance partitioning analysis between the biological fMRI test data and the two instances of synthetic fMRI test data.”

(37) Page 42, RSA-based decoding analysis procedure: Also here, a brief statement regarding the purpose of the analysis would be helpful. I also didn’t fully follow the reasoning by the statement made in the last two sentences of the first paragraph (lines 1060-1663).

Thank you for the opportunity to clarify and improve the RSA metadata methods. We now introduce this section with more motivation:

“We performed Representational Similarity Analysis (RSA) (Kriegeskorte, 2008) between metadata RDMs and neural RDMs to examine the extent that representations defined by the semantic metadata of varying information content are reflected in neural activity. The analysis broadly consisted of correlating a Representational Dissimilarity Matrix (RDM) defined by the metadata (Figure 7B) with a RDM at each voxel in the brain defined by the brain responses (Figure 7A).”

Note that this section underwent large changes to include the new Object label metadata and perform an analysis that better targets the conclusions we were previously making. Since we included the Object label metadata, the last two sentences of the first paragraph you refer to have been deleted. We reproduce the entirety of the “Metadata RSA Analysis Procedure” below:

“We performed Representational Similarity Analysis (RSA) (Kriegeskorte, 2008) between metadata RDMs and neural RDMs to examine the extent that representations defined by the semantic metadata of varying information content are reflected in neural activity. The

analysis broadly consisted of correlating a Representational Dissimilarity Matrix (RDM) defined by the metadata (Figure 7B) with a RDM at each voxel in the brain defined by the brain responses (Figure 7A).

We defined five metadata RDMs – object, scene, action, object+scene+action, and text description RDMs - from each of the 102 testing set videos. The object, scene, action, and text description RDM was defined by first indexing the annotations from the first five annotators (if the video contained annotations from more than five annotators). We then feed the 5 object, 5 scene, 5 action, and 5 text caption metadata from each of the 102 testing set videos into a language model to generate vector embeddings for each label. In the case of the object labels, since each annotator labeled up to three different objects, we computed the word embedding of each object label individually then averaged them (object labels corresponding to “no more objects in video” were skipped) to obtain one object embedding per annotator. The object, scene, and action labels were fed into the FastText model (Bojanowski et al., 2017) to compute single-word embeddings (length 300) and the text descriptions were fed into the Sentence-BERT (Reimers & Gurevych, 2019) model to compute sentence-level embeddings (length 512). To minimize the effect of noise in the metadata labels, we averaged the 3 most similar vector embeddings together to result in a single vector embedding that represents the object, scene, action, or text caption for that video. At this step, the object+scene+action embedding was created by concatenating the individual object, scene, and action vector embeddings (length 900). We then computed the pairwise cosine distance between each video’s vector embedding to produce a single 102 x 102 Representational Dissimilarity Matrix (RDM) for the object, scene, action, object+scene+action, and text caption metadata. Figure 7B shows the rank-normalized (rank each distance value and divide by the maximum rank) RDM for the object, scene, action, object+scene+action, and text description RDMs.

We compute the correlations (Spearman) between the metadata RDMs to measure the similarity of their information content. As expected, the correlations between the individual single-word object, scene, and action labels were generally lowest, since these labels highlight explicitly different components of the video. The correlations between the individual single-word object, scene, and action labels with the text descriptions were generally next highest, also expected because the text description likely contains information about each of the single-word labels plus extra information. The combined object+scene+action RDM with the text description RDM was higher still given even more overlapping information. The highest similarities were between the single-word object, scene, and action RDMs and the combined object+scene+action RDM because of explicitly overlapping information content (computed using the same embedding) but less extra information than the text description.

[scene, object]: 0.1808
[scene, action] 0.1133
[scene, scene+object+action]: 0.6641
[scene, text description]: 0.1969
[object, action]: 0.0884
[object, scene+object+action]: 0.6137

[object, text description]: 0.2618
[action, scene+object+action]: 0.5238
[action, text description]: 0.1560
[scene+object+action, text description]: 0.3145

To define the RDMs at each voxel in the brain for each subject, we perform a searchlight analysis in the way described in the “multivariate searchlight-based reliability analysis” methods section. To summarize, we center a sphere (radius of 4 voxels) around voxel v and extract the beta values (TRs 5-9 averaged over repetitions and z-scored across conditions) for all testing set conditions at all voxels encompassed in the sphere. Each stimulus thus has a corresponding vector of beta values, one from each voxel within the searchlight sphere. We compute the 1 - Pearson R correlation between all pairs of stimulus vectors to obtain an RDM at the centered voxel v . We repeat this process for all voxels in the whole brain for each subject (Figure 7A).

We then correlate (Spearman’s R) the metadata RDM (cosine-distance, not rank-normalized) with the searchlight-based RDMs at each voxel for each of the 10 subjects separately. For the whole-brain analysis (Figure 7C), we compute a t-test (1-sample, 2-sided) against a null hypothesis of a correlation of 0 at each voxel then perform FDR correction ($q=0.05$, assuming positive correlation) on all p-values in the whole brain to obtain a set of significant voxels. We compute the noise-normalized correlation by dividing the correlation with the voxel’s upper noise ceiling and plot the 10-subject average noise-normalized correlation at each significant voxel (Figure 7C). For the ROI-based analysis (Figure 7D), after we correlate (Spearman’s R) the metadata RDM with the searchlight-based RDM at each voxel, we compute the average noise-normalized correlation within each ROI. For each ROI, we compute a one-way ANOVA test between the average noise-normalized correlations corresponding to the five semantic metadata models. If the p-value of the ANOVA test is significant ($p<0.05$, Bonferroni corrected with $n=22$ ROIs), we perform a pairwise Tukeys Honestly Significant Difference test ($\alpha=0.05$). Significant differences between a pair of metadata models are reported with the dual-colored bars under the ROI name in Figure 7D.”

(38) Page 43, line 1096: What is the reason to use one-sided t-tests for the Memorability analysis procedure, and two-sided t-tests for the RSA-based decoding procedure?

In the memorability literature, there are clear a-priori hypotheses that relate memorability effects to stronger (not weaker) correlations (Bainbridge et al., 2017; Bainbridge & Rissman, 2018; Bylinskii et al., 2022; Jaegle et al., 2019). Thus, we perform one sided tests for correlations greater than 0.

We add the following text in the methods section “Memorability analysis procedure”:

“A one-sided, as opposed to two-sided, test against a correlation of 0 was computed because there exists clear a-priori hypotheses that relate memorability effects to

larger (not smaller) magnitudes of response (Bainbridge et al., 2017; Bainbridge & Rissman, 2018; Bylinskii et al., 2022; Jaegle et al., 2019)."

References

Lingnau, A., & Downing, P. E. (2015). The lateral occipitotemporal cortex in action. *Trends in Cognitive Sciences*, 19(5), 268-277.

Pitcher, D., & Ungerleider, L. G. (2021). Evidence for a third visual pathway specialized for social perception. *Trends in Cognitive Sciences*, 25(2), 100-110.

Wurm, M. F., & Caramazza, A. (2021). Two 'what' pathways for action and object recognition. *Trends in Cognitive Sciences*.

Reviewer #2 (Remarks to the Author):

In this paper, the authors introduce a new repository of data collecting whole-brain fMRI responses from 10 subjects to over 1000 short (3s) naturalistic video clips of visual events. They assessed the quality of preprocessing and showcased some exploratory results to highlight the potential of BMD in addressing temporal events, high-level concepts, and memorability topics using BOLD signals. Overall, this dataset is timely for fMRI-based neural encoding research. My comments are as follows:

Thank you for reviewing our manuscript and for acknowledging its relevance to encoding research. However, we respectfully disagree with the comment that we showcase "exploratory results" and take the comment as an opportunity to clarify the added value of our work. We contend that all our results extend multiple lines of computational neuroscience work into the video domain while grounding themselves in past literature. This solid grounding is necessary for careful and principled advancement of visual neuroscience, out of which exciting methodological and computational advancements can likely be made - we detail some of these opportunities in the revised manuscript's introduction:

"This dataset is well-suited to address open questions as diverse as developing methodologies to model rapid event BOLD signals (Misaki et al., 2013; Prince et al., 2022; Wittkuhn & Schuck, 2021), characterizing interactions between visual processing pathways (Lingnau & Downing, 2015; Mineault et al., 2021; Pitcher & Ungerleider, 2021; Wurm & Caramazza, 2022), and bridging the gap between still image and longform movie perception (Aliko et al., 2020; Allen et al., 2022; Hanke et al., 2016; Hebart et al., 2019).

Perception of videos does not evoke the same extent (Bartels & Zeki, 2004; Konen & Kastner, 2008; Press et al., 2001; Schultz & Pilz, 2009; Yildirim et al., 2019) or pattern (Buccino et al., 2004; Kret et al., 2011; Lingnau & Downing, 2015; Wurm & Caramazza, 2022) of brain responses as images throughout the ventral visual, dorsal visual, and parietal cortices, and this manuscript lays the crucial groundwork to further understand these differences.

We sincerely thank you for your further critical yet constructive comments as they have pushed us and given us the opportunity to clarify the novel potential of our dataset and add insightful analyses.

1. In Figure 1c, the differences between various blocks appear relatively small. How do the authors address the influence of previous stimulus blocks, especially considering hemodynamics and memory effects? There needs to be a justification for using the canonical HRF approaches to mitigate memory impacts.

Thank you for this comment. You correctly notice our intertrial interval is small (1 second) between videos. This small intertrial interval was chosen to achieve a large number of stimuli presentations across ten subjects while balancing practical limitations of budget and participant scanning time.

We first clarify that we did not use a canonical HRF to model the BOLD response, but a FIR function that flexibly fits across timepoints. This flexibility ensures a more accurate estimation of the hemodynamics in a rapid event related experimental design like BMD and in voxels both inside and outside visual cortex where the canonical HRF may be less appropriate.

Furthermore in this version of the manuscript, we additionally release a preprocessed version (new to this revision, see Supplementary 2 Figure 1 below and the text in Supplementary 2 for preprocessing details) of the data using GLMsingle (Prince et al., 2022), a toolbox to estimate single trial beta values specifically in rapid event-related designs. GLMsingle was developed to accurately model rapid event-related designs. GLMsingle's (1) selection of an optimal HRF for each voxel, as you suggest in your comment, and (2) fractional ridge regression for custom regularization at each voxel most directly pertain to accurate beta estimates of a rapid event-related design like BMD.

Figure 1: Overview of preprocessing pipelines. Version A (left) of BMD was preprocessed with fMRIPrep into a standard volumetric output space, modeled with FIR functions, and supplemented with 22 ROI

definitions. Details are provided in the main manuscript. Version B (right) of BMD was preprocessed with fMRIPrep into two volume-based and two surface-based output spaces. Single trial beta estimates using GLMsingle and 47 ROI definitions were provided in the standard volume-based space. Data was transformed into a fifth output space (fsLR32k) using the Ciftify toolbox and scripts from the Human Connectome Project preprocessing pipeline.

To address memory effects of a participant having previously seen a stimulus, we present all main experimental stimuli across 4 sessions, each collected on different days, to reduce the chances a subject would remember a particular stimulus. We decidedly did not employ a memory task (as in the Natural Scenes Dataset (Allen et al., 2022)), so the participants were not encouraged to retain stimulus information in memory. Stimulus presentation was also randomized across runs and sessions so as to not unintentionally prime the subject based on context of previous stimuli. To address memory effects caused by carry-over of hemodynamic effects from one trial to the next, the FIR model achieves flexible, voxel-specific BOLD estimates. The randomization of stimulus presentations and averaging over each stimulus's multiple repetitions would theoretically average out unwanted effects of memory.

You bring up a good point to include in the manuscript, so we add the following main manuscript text under the section "(f)MRI data processing, response modeling, and ROI definition":

"We modeled the hemodynamic response to visual events using data from 1-9s after stimulus onset (to account for the hemodynamic lag) in 1s steps (i.e. 9 bins of 1s length each) for each trial separately. The FIR model's flexible BOLD estimates, randomization of stimulus presentations across runs and sessions, and ability to average over multiple repetitions reduce potential unwanted memory effects."

2. In relation to Figure 4, the evaluation hinges on the choice of TSM ResNet50. Can the authors elucidate the roles of different blocks in TSM ResNet50 from a computer vision standpoint? This clarification might help in understanding the alignment between the two network types. Moreover, the stationarity of dynamics with the TR remains unclear, making it difficult to discern whether a region reconfigures its role or maintains a singular mode corresponding to a TSM ResNet50 block. Furthermore, given the plethora of existing research on static image encoding, this section doesn't offer new insights.

Thank you for this comment. In response, we (a) repeat this analysis with two additional models to examine the extent the results depend on the specific Temporal Shift Module (TSM) ResNet50, (b) elucidate the roles of the different blocks of the TSM ResNet50, especially in regard to TSM, and (c) comment on the interpretation of results and its new insights with respect to static image encoding.

(a) Generalization of results to two additional models

You make a good point that the results may not generalize beyond the TSM ResNet50 architecture used in the main manuscript. We thus perform this same encoding model procedure with a TSM MobileNetV2 and TimeSformer model to sample different architectures and training diets (see table

R1 below). Compared to the TSM ResNet50 trained on M4 dataset (Multi-Moments in Time Minus Memento10k), the TSM MobileNetv2 retains residual connections and TSM frame processing, but it is pretrained on the Kinetics-400 dataset and optimized for efficiency with over 5x fewer parameters. The TimeSformer uses a very different transformer architecture, no TSM frame processing (although frame order is still taken into account), and is pretrained on the HowTo100M dataset.

Architecture	Training Dataset	Training Objective	Increasing Late Layer Dominance
TSM ResNet50	M4	Action classification	Yes
TSM MobileNetV2	Kinetics-400	Action Classification	No
TimeSformer	HowTo100M	Action Classification	Yes

Table R1

We run inference on these two models with the 1,102 BMD video stimuli and extract the features for the encoding analysis. We plot results in Figure R1 below (now the revised Supplementary 1 Figure S7). We find that all three models have good prediction performance (> 0.3 noise normalized correlation) throughout cortex, including in dorsal visual and parietal ROIs. This result thus does not hinge on our initial choice of TSM ResNet50. However, we notice the dominance of the early layers (blue) in the low-level visual ROIs and the dominance of the later layers (red) in the high-level ROIs layers (blue) only holds true for the TSM ResNet50 model (the model detailed in the manuscript) and the TimeSformer. The TSM MobileNetV2 does not show such a pattern, as the high-level features (red) all predict each ROI significantly better than low-level features (blue) (see Figure R1 below). Thus, the dominance of early model layers in early ROIs and later model layers in later ROIs is more nuanced than we previously suspected. We edit the Supplementary 1 Figure S7 to include encoding results from the two additional models and edit the main manuscript to refer to these results:

“We observe that predictivity of DNN Block 4 becomes significantly greater than DNN Block 1 beginning in the ventral visual cortex and extending into dorsal visual cortex and parietal cortex (Figure 6C), reflecting the increase of feature complexity in the representations across the visual processing hierarchy. We repeat this encoding analysis with different architectures and training diets to see how this result generalizes across models. Specifically, we extract features from a TSM MobileNetV2 and a TimeSformer architecture trained on Kinetics-400 and HowTo100M datasets, respectively. We find that both are good predictors of dorsal visual and parietal regions, but the TSM MobileNetV2 does not show increasing dominance of high-level model features along the cortical hierarchy regions (see supplementary Figure S7 for results on all architectures and blocks).

This result extends previous research demonstrating a hierarchical correspondence between DNNs and brains from still image stimuli (Cichy, Pantazis, et al., 2016; Eickenberg et al., 2017; Kriegeskorte, 2015; Kubilius et al., 2019; D. L. K. Yamins et al., 2014) to dynamic video stimuli, a non-trivial outcome given that many cortical regions in the ventral visual and temporal cortex and beyond respond to stimulus features uniquely present in videos and not images (e.g., movement kinematics, temporal interactions)

(Lingnau & Downing, 2015; Pitcher et al., 2011; Wurm et al., 2017; Wurm & Caramazza, 2022). The finding that a transformer-based TimeSformer model follows a similar pattern to the TSM ResNet50 while the TSM MobileNetV2 does not invites further inquiry into the effect of training diet, parameter count, and architecture on visual event understanding. Finally, these results also help clarify previously conflicting results about whether or not DNNs trained on action recognition tasks accurately predict dorsal stream regions (Bakhtiari et al., 2021; Güçlü & van Gerven, 2017; Mineault et al., 2021), showing that **all three architectures** accurately predict responses not only in the dorsal visual stream but also in the parietal cortex.”

Figure S7: Encoding model performance on BMD

A) TSM ResNet50 trained on M4: Features were extracted after block's 1 (blue), 2 (orange), 3 (green), and 4 (red) in the ResNet 50 architecture. B) TSM MobileNetV2 trained on Kinetics-400: Features were extracted after the first bottleneck layer (blue),

third bottleneck layer (orange), sixth bottleneck layer (green), and last 2D convolutional layer before the average pool (red) in the MobileNetV2 architecture. C) TimeSformer S+T trained on HowTo100M: Of the model's twelve layers, features were extracted after the first (blue), fourth (orange), eighth (green), and twelfth (red) layers. The box plot on the left side in each panel shows the noise-normalized predictivity of four of each architecture's features at each of the 22 ROIs. The features were extracted at early (blue), intermediate (orange and green), and late (red) processing stages in each architecture to capture increasingly high-level degrees of transformations. The box plot on the right side in each panel shows the brain prediction difference between each architecture's latest and earliest layers for each subject and ROI. For the box plots on the right, a blue or red colored box plot denotes a significant difference in correlations from 0 ($p < 0.05$, two-sided one-sample t-test, Bonferroni corrected for $n=22$ comparisons), and gray denotes no significance. The box plots encompass the first and third data quartiles and the median (horizontal line). The whiskers extend to the minimum and maximum values within 1.5 times the interquartile range, and values falling outside that range are considered outliers (denoted by a diamond). The overlaid points show the value at each observation ($n=10$ for all ROIs except TOS ($n=8$) RSC ($n=9$)).

We note that we performed this analysis in the main manuscript with a ResNet50 model (as opposed to a more recent, better performing model) because of its well-established use in both computer science and computer vision literature, similarity to the human brain (Kietzmann et al., 2019; Koivisto et al., 2011; Kubilius et al., 2019; Pascual-Leone & Walsh, 2001; Schrimpf, Kubilius, Lee, et al., 2020; Silvanto, Cowey, et al., 2005; Silvanto, Lavie, et al., 2005), 2D architecture to bridge knowledge with still image models, and ability to add a temporal shift module (TSM) for analyses isolating temporal processing. We expand on these points below in (b).

(b) The roles of the different blocks of the TSM ResNet50

Regarding the roles of different blocks in the TSM ResNet50 model from a computer vision standpoint, each block does not have a pre-determined, engineered role. The model architecture processes the visual input with a series of convolutional operations (and recurrent connections) that effectively aggregates spatial information in increasingly large receptive fields. The model is trained to optimize a behavioral task – in this case, temporally relevant action classification – and updates the weights throughout the network accordingly to achieve increasingly better performance at the task. The earlier blocks learn to extract more low-level visual features, such as lines and edges, and the higher blocks learn to extract more high-level visual features, such as shapes and objects and here, actions.

The temporal shift module (TSM) (see Figure R2, panel C) merely changes how the information between input frames is shared (Lin et al., 2019). The more naïve approach (Figure R2, panel A) does not share information between frames. The uni-directional TSM approach (Figure R2 panel C) that we employ here shifts activations along the temporal direction so past frame information is shared with the current frame. TSM is inserted inside the residual branch of each residual block of the ResNet50. This addition of the TSM to the network is computationally free, adding no parameters to the network (the cost comes in a slight increase in latency due to shifting the activations in memory, but this cost does not concern the question at hand). Additionally, TSM is an addition to a 2D ResNet50 (the frames are convolved with 2D kernels), thus further bridging

research between still images and videos. TSM is not an architecture in itself, but merely an add-on to existing architectures to explicitly control how spatial information is being shared across frames. Taken together, TSM is an excellent method to isolate the effect of temporal processing of model features and subsequently brain prediction performance.

Figure R2: This figure is copied in its entirety from Figure 1 of the original Temporal Shift Module (TSM) paper (Lin et al., 2019). This figure schematizes differences in data movement between tensors with no shift between frames (a), bi-directional shift between frames in both temporal dimensions (bi-directional TSM), and (c) uni-directional shift between the past frame and current frame in the forward temporal dimension (uni-directional TSM). We use the uni-directional TSM (c) in our analyses.

(c) Interpretation of the results and new insights

You definitely pose an interesting question about a region reconfiguring its role over time, so we add that as a limitation (reproduced below). The brain activity we predict is averaged over TRs 5-9, the peak of the BOLD signal, so any reconfiguration of role by a region is smoothed over time. We believe the best way to study possible reconfigurations over a short time frame would be to collect high temporal resolution M/EEG data and analyze it alongside the fMRI data (EEG data collection for this short video stimulus set is currently taking place for a separate project). The added paragraph to the methods section, “DNN block to cortex correspondence procedure”, reads:

“We note that while the temporal dimension in videos invites exciting modeling opportunities, it also adds complexities in the fMRI data that may make modeling difficult. For example, regions may reconfigure their roles over the duration of the video or integrate features over time in a manner that cannot be resolved with fMRI. Thus, the extent that models can predict fMRI brain responses to videos may be inherently limited by the temporal resolution of fMRI and best be modeled alongside millisecond-level temporal resolution neuroimaging data (M/EEG).”

We respectfully disagree with the comment that this section does not offer new insights. The perceptual differences between image and video encoding are pervasive through occipitotemporal, dorsal visual, and parietal cortex (Lingnau & Downing, 2015; Pitcher et al., 2011; Pitcher & Ungerleider, 2021; Wurm & Caramazza, 2022). It would be empirically and theoretically unjustified to assume results from static image encoding would directly translate to video encoding. Thanks to this comment, we further found some biologically similar patterns of brain prediction performance seem to depend on model training diet and architecture - an important research direction to understand visual event understanding. As stated in the main text, we clarify previous conflicting

work that the objective of action recognition can highly predict regions in the dorsal visual and parietal cortices. This analysis makes the necessary step to extend results into the video domain while being bound back to previous work.

3. For Figure 5, I'm primarily concerned about the extent to which the results hinge on the choice of TSM. What kind of temporal dynamics is encoded by frame order? For instance, even when the order is shuffled, the variance—potentially another form of dynamics—remains unchanged. Given the temporal resolution of BOLD and the brain's processing speed for image sequences, the concept of "dynamics" requires further elucidation and discussion.

Similar to the previous comment #2, thank you for challenging us to test the generalizability of our results to different architectures. In response to this comment we (a) repeat this frame shuffling analysis using a non-TSM architecture and (b) perform another analysis isolating temporal integration to add to the discussion around the concept of "dynamics."

(a) Frame-shuffling using non-RSM architecture

We recognize the value in determining the influence of the TSM component on the results, so we repeat this analysis on a TSM MobileNetV2 model trained on the Kinetics-400 dataset and a more recent (non-TSM) TimeSformer model trained on the HowTo100M dataset. The TSM MobileNetV2 has the TSM component and residual connections in common with the TSM ResNet50, but contains over 5x fewer parameters and is less accurate. The TimeSformer, on the other hand, has about 5x more parameters than a ResNet50 and leverages a transformer-based architecture for spatial and temporal video processing (but does not use TSM). These two models are a good test to determine if the results found with the TSM ResNet50 can generalize to distinct video-computable models.

We show the results in the new Supplementary 1 Figure S10 (reproduced below). The TSM ResNet50 is the only model of the three that shows a decrease in brain prediction accuracy when the input frames are shuffled (see the color-coded asterisks along the x-axis as indicator of significance).

“Figure S10: The effect of frame shuffling on brain prediction performance across different architectures
We compute the difference in the correlation between the shuffled frame prediction accuracy and unshuffled frame prediction accuracy at all 22 ROIs and four layers of a (A)

TSM ResNet50, (B) TSM MobileNetV2, and (C) TimeSformer model. Features were extracted at increasing levels of depth in each model (blue, orange, green, red) that reflect higher levels of model processing stages. Only the TSM ResNet50 architecture trained on the M4 dataset (Multi-moments Minus Memento10k) showed evidence of robust differences across cortex between shuffled and unshuffled input. Colored asterisks along the x-axis plot indicates significant difference between the unshuffled and shuffled prediction accuracy at that DNN block (one sample two-sided t-test against a population mean of 0, FDR correction across 22 ROIs x 4 blocks = 88 comparisons, $p < 0.05$). The box plot encompasses the first and third data quartiles and the median (horizontal line). The whiskers extend to the minimum and maximum values within 1.5 times the interquartile range, and values falling outside that range are considered outliers (denoted by a diamond). The overlaid points show the value at each observation (n=10 for all ROIs except TOS (n=8) RSC (n=9))."

While we find this result intriguing, we are not concerned that the TSM MobileNetV2 and TimeSformer models are not as strongly impacted by frame shuffling for two reasons. First, the TSM ResNet50 was specifically chosen for its biological similarity to computations in the human visual system (Kietzmann et al., 2019; Koivisto et al., 2011; Kubilius et al., 2019; Pascual-Leone & Walsh, 2001; Schrimpf, Kubilius, Lee, et al., 2020; Silvano, Cowey, et al., 2005; Silvano, Lavie, et al., 2005), and thus it can be expected to suffer from shuffled inputs like a human would. The other two models are notably different in their biological similarity. The TSM MobileNetV2 is engineered for low latency (fast) and lightweight (few parameters) usage, not task accuracy, and the TimeSformer's transformer architecture and high parameter count may lessen the impact of frame shuffling. Second, a null result in different architectures does not change the fact we see robust effects throughout most other ROIs and blocks in the TSM ResNet50 experiment. This analysis definitely brings up multiple interesting research avenues to explore biological similarities of models and humans through a temporal lens.

We add discussion of this result to the main text section, "fMRI responses capture temporal event structure":

"Lastly, we repeat this frame shuffling analysis on a TSM MobileNetV2 (Lin et al., 2019; Sandler et al., 2018) and TimeSformer (Bertasius et al., 2021) architecture trained on Kinetics-400 (W. Kay et al., 2017) and HowTo100M (Miech et al., 2019) datasets, respectively, to see if the frame-shuffling results generalize to models of varying architectures, training diets, parameter counts, and task performance. We find that the TSM ResNet50 is the only model of the three that sees robust effects in most ROIs and DNN blocks (see Supplementary 1 Figure S10 for all results), implying that biological similarity of the model may be closely tied to the effects of frame shuffling."

(b) Additional frame-shuffling analysis to determine dynamics

We agree that relating the word "dynamics" specifically to the frame shuffling analysis is vague and inexact. Some temporal dynamics of the brain that this frame shuffling analysis is affecting may be related to temporal integration across frames, next frame prediction, residual feedback connections, or some other mechanism. While this fMRI dataset is not well suited to disentangle the exact nature of temporal dynamics we observe here, we do have the opportunity to isolate the

effect of temporal integration (Fairhall et al., 2014; Hasson, Yang, et al., 2008) by leveraging our TSM model. This analysis is complementary to the frame shuffling analysis and further elucidates the temporal dynamics at play here.

We compare the brain prediction performance of a Temporal Shift Module (TSM) ResNet50 (Lin et al., 2019) and a Temporal Segment Network (TSN) ResNet50 (Wang et al., 2016) to isolate the dynamic of temporal integration. The difference between these two architectures lies in how information between frames is shared. In Figure R2 above (reproduced from (Lin et al., 2019)), panel A describes a TSN implementation where information is not shared between frames. Figure R2 panel C describes a uni-directional TSM implementation, used here, where information from the past frame is shared with the current frame. The backbone ResNet50 architecture, number of model parameters, objective function, learning algorithm, unshuffled input frame order, and training diet are all held constant. We train both a TSM ResNet50 model and a TSN ResNet50 model on a 10,000-video subset of the 1 million+ M4 dataset that the model presented in the manuscript was trained on. This smaller scale was necessary as training on the full dataset took over three months and 16 V100 GPUs – resources we did not have for this revision.

We first note that our TSM network outperforms our TSN network in action classification accuracy (mean average precision = 0.281 and 0.268, respectively). This result is in line with the abundance of networks that benefit from TSM (Lin et al., 2019) – and hence TSM’s popularity - but this fact is still worth mentioning given how few differences exist between our TSM and TSN networks.

Now we ask: does the TSM model’s superior task performance translate to superior brain prediction performance? We reason that if the TSM achieves superior brain prediction performance over its TSN counterpart then the brain activity is capturing information related to temporal integration, as this is the only difference between the TSM and TSN architectures.

Using the same encoding model procedure in the analyses presented in Figures 4 and 5, we compute the difference in brain prediction accuracy for each subject at each ResNet50 Block and ROI (Supplementary Figure S9, new to this revision, reproduced below). We find that early visual regions being predicted by late Blocks 3 and 4 experience the strongest affect. However, this trend does not follow an equivalent pattern observed in the frame shuffling analysis, where significant differences were seen in early visual ROIs from all DNN blocks and late visual ROIs from primarily late model blocks. We conclude that the TSM vs TSN analysis and the frame shuffling analysis are perturbing different temporal dynamics.

“Figure S9: The effect of Temporal Shift Module (TSM) on brain prediction performance (A) TSM vs TSN prediction performance: The difference in subject brain prediction performance of a TSM ResNet50 and Temporal Segment Network (TSN) ResNet50 each trained on a 10,000-video subset of the M4 dataset (Multi-moments Minus Memento) was computed at each of the four Blocks for each ROI. TSM results in increased brain prediction performance most prominently in early visual ROIs. In both panels, colored asterisks along the x-axis plot indicates significant difference between the unshuffled and shuffled prediction accuracy at that DNN block (one sample two-sided t-test against a population mean of 0, FDR correction across 22 ROIs x 4 blocks = 88 comparisons, $p < 0.05$). The box plot encompasses the first and third data quartiles and the median (horizontal line). The whiskers extend to the minimum and maximum values within 1.5 times the interquartile range, and values falling outside that range are considered outliers (denoted by a diamond). The overlaid points show the value at each observation ($n=10$ for all ROIs except TOS ($n=8$) RSC ($n=9$)).”

We add this figure to Supplementary 1 and add text to the main manuscript section, “fMRI responses capture temporal event structure”, to further discuss temporal dynamics:

“In order to further elucidate the type of temporal dynamics in the brain that this frame shuffling analysis is affecting, we perform an additional small-scale experiment comparing the difference in prediction accuracy between a Temporal Shift Module (TSM) (Lin et al., 2019) ResNet50 and a Temporal Segment Network (TSN) (L. Wang et al., 2016) ResNet50. Since the only difference between the two models is how spatial information is shared between frames, this analysis effectively isolates temporal integration. We find that implementation of TSM significantly improves encoding accuracy in early visual ROIs primarily from DNN blocks 3 and 4 (see Supplementary 1 Figure S9 for results). This

pattern is different from the one observed from the frame shuffling analysis (Figure 5C), suggesting that the brain activity is capturing various forms of temporal dynamics.”

Your comment here gave us the opportunity to explore the rich temporal dynamics that BMD captured and the intriguing limitations of these effects. We believe these additional analyses and discussion spurred by this comment adds a lot of value to the manuscript and will be of a lot of interest to the researchers interested in using this dataset.

4. Pertaining to Figure 6, how reliable are the TR peak estimations? Given the size of the unique variance, I'm skeptical about the validity of the claims presented. Additionally, it seems the only definitive conclusion is that BOLD signals can differentiate between images with a 1-2TR delay. Contrasting experiments using the same stimuli might be beneficial.

Thank you for the comment. Regarding the reliability of the TR peak estimations (Figure 6E, upper ROI panel blue and orange boxplots and supplementary Figure 6, reproduced below), we provide box and whisker plots overlaid with individual data points to be transparent about the observed effect's reliability. The peak TR for first (blue star) and third epoch (orange star) is the TR with the highest subject averaged unique variance and that reached statistical significance (difference in unique variance against 0, $p < 0.05$, one-sample two-sided t-test, FDR corrected for 9TRs x 2 video epochs = 18 comparisons). All ROI plots are additionally provided in the supplementary material (reproduced below).

Figure S6: Encoding the temporal dynamics of the BOLD signal.

(A) Whole-brain analysis: Each voxel shows the percentage of subjects with a TR peak difference of 2 TRs at that specific voxel. Only significant voxels are plotted ($p < 0.05$, binomial test, FDR corrected). The effect of interest is showing predominantly in the visual cortex. (B) ROI analysis: Unique variance explained by the first and third video epoch (second) synthetic fMRI data, at each TR. Red asterisks along the x-axis indicate unique variance scores significantly greater than 0 ($p < 0.05$, one-sample one-side t-test, FDR corrected across 9 TRs x 2 video epochs = 18 comparisons). Large blue/orange stars indicate the TR with the highest subject averaged unique variance for the first/third video epochs, respectively. The box plot encompasses the first and third data quartiles and the median (horizontal line). The whiskers extend to the minimum and maximum values within 1.5 times the interquartile range, and values falling outside that range are considered outliers (denoted by a diamond). The overlaid points show the value at each observation ($n=10$ for all ROIs except TOS ($n=8$) RSC ($n=9$)). Y-axis and X-axis labels are shared horizontally and vertically, respectively.

This procedure to compute a “peak TR” takes into account results from all 10 subjects, and the unique variance calculation is a rather robust method of isolating our desired effect. Other potential methods to determine a “peak TR”, like an average TR weighted by a difference in unique variance magnitude, will result in nonsensical floating point TRs (e.g., TR=3.4) and/or add unnecessary complexity to the analysis. We believe the difference between the “peak TRs” at the subject level (Figure 6E, bottom panel green boxplots) combined with additional thorough statistical tests ($p < 0.05$, one-sample two-sided t-test) is the most accurate way to measure temporal encoding in the BOLD signal in an ROI.

The size of the effect is numerically small, as expected, yet is significant even after a robust set of statistical tests. We put our result in the context of a recent breakthrough finding published in Nature Communications (Wittkuhn & Schuck, 2021), where the researchers reliably decode the sequence and content of five *highly distinct and uncorrelated images* (cat, chair, face, house, shoe) rapidly presented within a few hundred milliseconds and a couple seconds. In contrast, we investigated whether a sequence of *highly similar and conceptually related* video frames totaling 3 seconds (as in BMD) can be disentangled. We show that yes, early and late TRs of brain responses are better explained by early and late video frames. This result both extends results from static images to dynamic videos and opens new lines of research to investigate the limits of BOLD’s encoding of rapidly presented stimuli. We add this point to the main text:

Together, these results support the hypothesis that early and late TRs of the BOLD signal better code early and late video snapshots, respectively. Previous work showed that the content and sequence order of distinct images presented in under a second can be reliably decoded in the BOLD signal (Wittkuhn & Schuck, 2021). Here we add to this line of work to show that BMD can differentiate between two visually and conceptually similar stimuli of one second duration (i.e., first and third video second) separated by another highly similar stimulus of one second duration (the second video second) with the most pronounced effects focused in the early visual cortex (Fairhall et al., 2014; Kiebel et al., 2008). This invites future research to use BMD’s temporally well-defined stimuli to explore how visual event information is integrated over shorter time periods, bridging an important gap to temporal integration studies of longform movies and BOLD encoding of rapidly presented stimuli.

In summary, we agree that the conclusion of this analysis is that the BOLD signal can reliably differentiate between “images” 1 second in duration separated by another “image” of 1 second duration (i.e., a three second video). However, each “image” is visually and conceptually highly similar and presented as a continuous stream. We believe these results go beyond previous literature (Wittkuhn & Schuck, 2021) and show that the BOLD response is sensitive to dynamic content presented in videos at the level of seconds.

We agree that conducting another fMRI experiment to compare and contrast brain activity evoked from the same movie stimuli and still frames would add more insights into temporal encoding of the BOLD signal. However, the feat would be immense: It would effectively double the data and thus the amount of information to be conveyed (from 8 to 2*8 main figures). It is thus

unfortunately impractical to recruit the same subjects, purchase enough scanner time, and analyze as well as succinctly communicate results for this revision.

5. As for Figure 7, while the findings are intriguing, they aren't novel. A more extensive static dataset might provide clearer insights. I recommend the authors explore meta-information that can uniquely be derived from the videos.

Thank you for the excellent suggestion to explore additional metadata and giving us the opportunity to clarify the novelty of our findings. In response to your comment, we (a) acquired new meta-information from human behavior and relate it to the brain to strengthen the analysis presented in Figure 7, (b) added motion-energy features of the videos as another crucial meta-information and related them to brain responses, and (c) theoretically clarified the relevance of a video vs. a static dataset in a supplementary discussion section. We detail the three points below.

(a) New meta-information from human behavior

We collected new meta-information from human behavior and their relationship to the brain in order to better support our findings presented in Figure 7. Our previous five object and scene labels were based on a neural network model's top 5 predictions. New to this revision, we ran two additional large-scale crowd-sourced experiments to gather human annotated object and scene labels that are more accurate and grounded in human truth. We use the THINGS (Hebart et al., 2023) and Places365 (Zhou et al., 2018) datasets to define the set of possible object and scene labels, respectively, further enabling cross-talk between BMD and other computational neuroscience work leveraging these datasets. We describe this metadata in detail under the main manuscript section, "Semantic and behavioral metadata on visual events", under the methods section, "Metadata", and in response to Reviewer 1 minor comments 4, 8, 16, 17, and 18. Below we reproduce the object and scene metadata outlined in the methods section:

"Object labels

For each video, we obtained at least 5 sets of up to three different object labels in a human crowd-sourced experiment on Prolific. Each annotator was instructed to select up to three different object labels visible in the video. They selected at least one object label, and if they believed no more objects were present in the video, they were allowed to select a "No more objects in the video" option up to two times. This option thus encouraged accurate labels and carried information on the density of objects in the video. Each object label was one of 1,854 possible labels from the THINGS dataset (Hebart et al., 2023) to encourage overlap with computational neuroscience work and leverage the additional THINGS metadata on each label (e.g., animacy, size, indoor/outdoor). The object label selections can be different or the same across annotators. One author manually reviewed the labels to ensure the labels assigned to the video were sensible (i.e. participants were not choosing labels at random).

Scene labels

For each video, we obtained at least 5 scene labels using a human crowd-sourced experiment on Prolific. Each of the five different annotators were instructed to select a scene label that best describes the scene of the video. All scene labels came from the Places365 dataset (Zhou et al., 2018) for its broad scene coverage and overlap with

computer vision resources. The scene label selections can be different or the same across videos. One author manually reviewed the labels to ensure the labels assigned to the video were sensible (i.e. participants were not choosing labels at random)."

Regarding the analysis in Figure 7 (reproduced below), the new object and scene labels allow us to make a more controlled examination of visuo-semantic representations in human cortex. We correlate the Representational Dissimilarity Matrices (RDMs) of the object, scene, action, a concatenation of "object+scene+action", and sentence text description labels with the neural RDMs. We compute the pairwise significance between means of the metadata categories (using one-way ANOVA and Tukey's HSD tests) in each ROI to better target the conclusions we were previously making.

We better motivate this analysis in the main text, making sure to emphasize the increased functional diversity of regions throughout cortex in response to video perception:

"Varying levels of semantic information content, from static objects and scenes (e.g., "duck", "water") to temporal actions (e.g., "swimming") to complex relations between parts (e.g., "the duck is swimming on the water"), can describe a visual event. It is unclear how these varying levels of complexity and content are reflected in ROIs while viewing visual events, especially given that regions throughout the ventral visual, dorsal visual, and parietal cortices have all been implicated in processing temporal aspects of videos (Buccino et al., 2004; Konen & Kastner, 2008; Lingnau & Downing, 2015; Silver & Kastner, 2009; Wurm & Caramazza, 2022) but also diverse feature preferences (Buccino et al., 2004; Kanwisher, 2010; Kanwisher et al., 1997; Konen & Kastner, 2008; Lafer-Sousa et al., 2016; Ratan Murty et al., 2021)."

We find that the sentence text description representations best correlate in nearly all ROIs by a wide margin. We then ask if the reason for the sentence text description's higher correlations is due to finer granularity of labels (e.g., text descriptions were collected free form, while category labels were constrained by a vocabulary). The higher (or equally high) correlations of the concatenated "object+scene+action" representation compared to the individual object, scene, or action representation suggests that the text description's high correlations are driven by cortex's objective of complex scene analysis, not a finer description of labels. We reproduce the description of our novel findings below:

"Overall, the sentence text description results in stronger (or equally strong) correlation values than the other four semantic descriptions in all ROIs. Additionally, the three concatenated single-word labels (object+scene+action) results in stronger (or equally strong) correlation values than the individual single-word labels across all regions, even in category-selective regions. Both of these results are consistent with the idea that complex scene analysis, rather than simpler tasks such as object recognition, is the objective of the visual brain (Doerig et al., 2022). One might suspect that the category-selective ventral regions would best correlate with their respective metadata (e.g., PPA, RSC, and TOS for scene metadata), reasoning that the text description and object+scene+action labels, while including the pertinent category information, contain mostly irrelevant and

distracting *extra-category* content (Kosakowski et al., 2022; Ratan Murty et al., 2021, but see Bonner & Epstein, 2021). “

In summary of point (a), the analysis presented in Figure 7 offers novel insights into visuo-semantic representations of the brain that even an extensive static dataset cannot accomplish due to our video-evoked brain activity and use of “action”, “object+scene+action”, and “text description” temporal metadata. We are able to share similar methodologies to previous image-only work (Doerig et al., 2022) to ground our insights in existing literature while presenting new ones. We further highlight how the added value of short video fMRI datasets over static image datasets in part (c) of this response.

[redacted]

Motion energy prediction of brain activity: C) The boxplots depict the correlation (Pearson) between the predicted brain responses with the true responses of the testing set using stimulus features computed from a motion energy model. The boxes show the median response across subjects (horizontal line), 25th and 75th percentile (lower and upper box boundary), and whiskers extending to maximum and minimum values within 1.5 times the interquartile range. Individual subject results are shown as black points, and outliers are shown as diamonds (n=10 subjects for all ROIs).

This result shows the motion energy features best predicted brain activity in motion-selective ROIs of hV4, V3ab, IPS0, and MT. This result provides more evidence that BMD captures motion information, and we provide the motion energy features as additional metadata. We refer to this analysis in the Methods section under “Metadata”, the supplementary 1 discussion (detailed next in point (c)), and in greater detail in the supplementary 2. The text in the methods section reads:

Motion energy features

We provide video-computable motion features of each video using a motion energy model (Adelson & Bergen, 1985; Nishimoto et al., 2011; Watson & Ahumada, 1985). The motion energy model uses a set of spatial and temporal Gabor filters to extract a video’s motion and direction. We use these features to predict brain activity in motion-selective regions of MT (Born & Bradley, 2005; Nishimoto & Gallant, 2011), hV4 (Kamitani & Tong, 2006; Roe et al., 2012), V3AB (Konen & Kastner, 2008; Smith et al., 1998), and IPS0 (Konen & Kastner, 2008). This analysis uses version B of the dataset and is detailed in Supplementary 2 section 5, “Motion energy features computation and encoding model.”

The text in Supplementary 2 describing the motion energy methods reads:

“5 Motion Energy Features Computation and Encoding Model

Motion energy features were used to predict brain activity in response to BMD’s 3 second naturalistic videos. The motion energy model (Adelson & Bergen, 1985; Nishimoto et al., 2011; Watson & Ahumada, 1985) consists of a series of spatial and temporal Gabor filters intended to capture local motion and direction in a video stimulus, thus making it a highly interpretable method to model video dynamics. The motion energy encoding model accuracy (Figure 2C) shows highest prediction accuracy in motion selective ROIs, namely MT (Born & Bradley, 2005; Nishimoto & Gallant, 2011), hV4 (Kamitani & Tong, 2006; Roe et al., 2012), V3AB (Konen & Kastner, 2008; Smith et al., 1998), and IPS0 (Konen & Kastner, 2008). These results support that single trial beta estimates of BMD’s 3 second naturalistic videos capture motion information.

Motion energy features for each BMD video stimulus was computed using the MATLAB code available here: https://github.com/gallantlab/motion_energy_matlab (Nishimoto et al., 2011; Nishimoto & Gallant, 2011). For each 268x268 video, the frames were converted from RGB to LAB color space, and only the L (luminance) channel was retained. The luminance channel was then passed through a three-dimensional bank of spatiotemporal Gabor filters consisting of two spatial dimensions and one temporal dimension. Similar to the filter bank used in (Nishimoto et al., 2011) to model naturalistic movies, the three-dimensional filters are defined at five spatial frequencies (0, 2, 4, 8, 16, and 32 cycles/image), three temporal frequencies (0, 2, and 4Hz), and eight directions (0, 45, 90,

135, 180, 225, 270, and 315 degrees) with the exception that the 0 Hz temporal filter is defined at only 0, 45, 90, and 135 degrees directions and the 0 cycles/image spatial filter is defined at 0 degree orientation. Local motion-energy features were computed by taking the square root of the sum of the squared outputs of each pair of filters with orthogonal phases. The logarithm of the output from these filters was computed to scale large values, and the temporal dimension of the output was downsampled to 1 second to match the fMRI sampling rate (i.e., the interpolated TR of 1 second) of the BOLD time series. The output was then z-scored across time. In total, this procedure resulted in a matrix of size 3 x 6555 (seconds x motion energy features).

The motion energy features were then used in a voxelwise linear encoding model (Naselaris et al., 2011) to predict the brain activity (beta estimates) in 47 regions of interest (ROIs) from the version B preprocessed data in MNI152NLin2009cAsym space (Figure 2C). Specifically, the motion energy features for each video were concatenated along the three seconds and underwent principal component analysis (PCA) to reduce dimensionality to the top 100 components. PCA was fit to the training videos and applied to both the training and testing videos. A linear model was then fit to the training video features to predict the response at the voxel. The learned weights of the linear model were then applied to the testing video features. The encoding model accuracy was computed as the correlation of the vector of predicted responses of the test set with the vector of true responses of the test set.”

(c) Relevance of the video vs. the static dataset

We clarify the relevance of a short video vs static image dataset in a new discussion in Supplementary 1 titled, “The added value of brain responses to a short video dataset versus a static image dataset.” This section explains how videos are different from images, evidence that these differences are reflected in fMRI data (from previous work and BMD), and what open questions a short video dataset might thus be well-suited to address. It reads:

The added value of brain responses to a short video dataset versus a static image dataset

“We emphasize that a short video (e.g., 3 second duration, as in BMD) fMRI dataset is not better or worse than a static image fMRI dataset; rather, they are different in terms of stimulus features and corresponding brain responses that may make one better suited to answer specific research questions. Most obvious, short videos contain a temporal dimension that static images do not, allowing the video to communicate crucial contextual information about how spatial components in our environment move (or not) and spatially relate to each other over time. The benefit of this temporal dimension is clear in our everyday lives – we can interpret transitions between states (a door is being opened, not closed), direction (a steering wheel is being turned to left, not right or still), reactions (the child laughed when shown the picture), motion (the baby is crawling slowly, not fast), and more.

The contextual value of a video’s temporal dimension is reflected in BMD’s own action and sentence text description metadata. Concerning action labels, images can only be labelled with a limited subset of actions or else be highly constrained in order to capture a

specific action. For example, the action of a baseball player “hitting” the ball can only be captured with an image if the photo were taken at very specific instant in time, otherwise the action may be “standing” or “swinging”. Even a short video like in BMD easily captures these actions without heavily constraining the space of possible videos that correspond to “hitting”. Concerning text descriptions, short videos can capture temporal sequences of events that an image cannot. Examples of such captions sampled from some of BMD’s first videos include (emphasis our own):

- Video 0001: "A mallard is in the water alone *swimming around and putting its beak in.*"
- Video 002: "A man *is showing another man how to move feet back and forth.*"
- Video 005: "A woman guides a little boy's arms *up and down as other kids stretch around him.*"
- Video 006: "a chess tournament is going on this is focused on two players one *is moving their queen and taking something to put the king in checkmate*"

Static frames of these videos cannot capture the temporal facts that the mallard is “putting its beak in”, the man “is showing another man how”, “a woman guides...as other kids stretch”, and a chess player “is moving their queen and taking something to put the king in checkmate.” This temporal information adds valuable context that often makes one’s understanding of the 3s video vastly different compared to any one of its single static frames.

But do these differences in short videos and static images translate to differences in fMRI brain responses? Yes, previous work has found that videos evoke a greater extent (Bartels & Zeki, 2004; Konen & Kastner, 2008; Press et al., 2001; Schultz & Pilz, 2009; Yildirim et al., 2019) and pattern (Buccino et al., 2004; Kret et al., 2011; Lingnau & Downing, 2015; Wurm & Caramazza, 2022) of cortex responding to videos than images throughout occipitotemporal, dorsal visual, and parietal cortex. In this manuscript we describe our highly reliable activations throughout cortex (Figure 3) with notably high reliability in parietal cortex, a region of the brain that weakly responds to static images. These highly reliable brain responses are not just a result of increased participant engagement or stimulus saliency; we even show that BMD brain responses capture temporal information from the videos (Figure 5, Figure 6, Supplementary 1 Figure 9, Supplementary 2 Figure 2) despite the BOLD response’s temporal sluggishness and fMRI’s low sampling rate.

In the neighboring field of computer vision, researchers have long recognized that videos and images demand different modeling approaches (Ahn et al., 2023; Bertasius et al., 2021; Lin et al., 2019; Tong et al., 2022; Wang et al., 2016) and training datasets (Goyal et al., 2017; Kay et al., 2017; Miech et al., 2019; Monfort et al., 2020; Soomro et al., 2012) for strong task performance. Videos continue to be at the forefront of ground breaking computer vision research due to their creative, cross-domain, and practical applications in text-to-video generation (Ho et al., 2022; Singer et al., 2022; Wu et al., 2023), video understanding with large language models (Ju et al., 2022; Maaz et al., 2023; Zhang et al., 2023), and efficient action recognition and pose estimation (Liu et al., 2023; Qing et al., 2024; Zheng et al., 2023).

Taken together, short video fMRI datasets offer unique opportunities to advance the field of computational neuroscience where static image fMRI datasets cannot. They can advance methodologies around estimating BOLD signals in response to rapid stimulus presentations (Misaki et al., 2013; Prince et al., 2022; Wittkuhn & Schuck, 2021), elucidate cognitive functions concerning temporal integration (Fairhall et al., 2014; Hasson et al., 2008; Orlov & Zohary, 2018), test temporally specific cognitive objective functions (Doerig et al., 2022; Kanwisher et al., 2023), and detail how multiple visual pathways interact to achieve an understanding of an event (Lingnau & Downing, 2015; Mineault et al., 2021; Pitcher & Ungerleider, 2021; Wurm & Caramazza, 2022). As neuroscience and computer science research become increasingly intertwined (Allen et al., 2022; Chen et al., 2023; Cichy et al., 2019, 2021), BMD is well-suited to integrate with state-of-the-art video modeling work from the computer vision community. Importantly, a short video dataset like BMD can make these scientific advancements while staying connected to the vast body of still image work by sharing event-related paradigms, multivariate and univariate methodologies, representational similarity analyses, and/or encoding and decoding techniques. Short video datasets offer more ecological validity than static images while retaining experimental control and offer tremendous potential to advance our understanding of the human visual system.”

In summary, this dataset is crucial and beneficial for the broader research community. Concerning the manuscript, if the submission is to Scientific Data, its current content seems apt. However, if intended for Nature Communications, the content should emphasize the dynamic aspects and condense sections relating to static image stimuli.

Thank you again for your comments as they have directly resulted in a significantly improved manuscript. The additional analyses and textual clarifications we provide as a result of this review will be valuable to the researchers excited to use this dataset. We hope we have convinced you of BMD’s novelty, its temporal richness, and the many opportunities it presents to further our understanding of the human visual system.

References:

- Adelson, E. H., & Bergen, J. R. (1985). Spatiotemporal energy models for the perception of motion. *JOSA A*, 2(2), 284–299. <https://doi.org/10.1364/JOSAA.2.000284>
- Ahn, D., Kim, S., Hong, H., & Ko, B. C. (2023). *STAR-Transformer: A Spatio-Temporal Cross Attention Transformer for Human Action Recognition*. 3330–3339. https://openaccess.thecvf.com/content/WACV2023/html/Ahn_STAR-Transformer_A_Spatio-Temporal_Cross_Attention_Transformer_for_Human_Action_Recognition_WACV_2023_paper.html
- Aliko, S., Huang, J., Gheorghiu, F., Meliss, S., & Skipper, J. I. (2020). A naturalistic neuroimaging database for understanding the brain using ecological stimuli. *Scientific Data*, 7(1), Article 1. <https://doi.org/10.1038/s41597-020-00680-2>
- Allen, E. J., St-Yves, G., Wu, Y., Breedlove, J. L., Prince, J. S., Dowdle, L. T., Nau, M., Caron, B., Pestilli, F., Charest, I., Hutchinson, J. B., Naselaris, T., & Kay, K. (2022). A massive 7T fMRI dataset to bridge cognitive neuroscience and artificial intelligence. *Nature Neuroscience*, 25(1), Article 1. <https://doi.org/10.1038/s41593-021-00962-x>
- Bainbridge, W. A. (2019). Chapter One - Memorability: How what we see influences what we remember. In K. D. Federmeier & D. M. Beck (Eds.), *Psychology of Learning and Motivation* (Vol. 70, pp. 1–27). Academic Press. <https://doi.org/10.1016/bs.plm.2019.02.001>

- Bainbridge, W. A., Dilks, D. D., & Oliva, A. (2017). Memorability: A stimulus-driven perceptual neural signature distinctive from memory. *NeuroImage*, *149*, 141–152.
<https://doi.org/10.1016/j.neuroimage.2017.01.063>
- Bainbridge, W. A., & Rissman, J. (2018). Dissociating neural markers of stimulus memorability and subjective recognition during episodic retrieval. *Scientific Reports*, *8*(1), Article 1.
<https://doi.org/10.1038/s41598-018-26467-5>
- Bakhtiari, S., Mineault, P., Lillicrap, T., Pack, C., & Richards, B. (2021). The functional specialization of visual cortex emerges from training parallel pathways with self-supervised predictive learning. *Advances in Neural Information Processing Systems*, *34*, 25164–25178.
<https://proceedings.neurips.cc/paper/2021/hash/d384dec9f5f7a64a36b5c8f03b8a6d92-Abstract.html>
- Bartels, A., & Zeki, S. (2004). Functional brain mapping during free viewing of natural scenes. *Human Brain Mapping*, *21*(2), 75–85. <https://doi.org/10.1002/hbm.10153>
- Behzadi, Y., Restom, K., Liu, J., & Liu, T. T. (2007). A component based noise correction method (CompCor) for BOLD and perfusion based fMRI. *NeuroImage*, *37*(1), 90–101.
<https://doi.org/10.1016/j.neuroimage.2007.04.042>
- Bertasius, G., Wang, H., & Torresani, L. (2021). Is Space-Time Attention All You Need for Video Understanding? *Proceedings of the 38th International Conference on Machine Learning*, 813–824. <https://proceedings.mlr.press/v139/bertasius21a.html>

- Bojanowski, P., Grave, E., Joulin, A., & Mikolov, T. (2017). Enriching Word Vectors with Subword Information. *Transactions of the Association for Computational Linguistics*, 5, 135–146.
https://doi.org/10.1162/tacl_a_00051
- Bonner, M. F., & Epstein, R. A. (2021). Object representations in the human brain reflect the co-occurrence statistics of vision and language. *Nature Communications*, 12(1), Article 1.
<https://doi.org/10.1038/s41467-021-24368-2>
- Born, R. T., & Bradley, D. C. (2005). Structure and Function of Visual Area Mt. *Annual Review of Neuroscience*, 28(1), 157–189.
<https://doi.org/10.1146/annurev.neuro.26.041002.131052>
- Buccino, G., Binkofski, F., Fink, G. R., Fadiga, L., Fogassi, L., Gallese, V., Seitz, R. J., Zilles, K., Rizzolatti, G., & Freund, H.-J. (2004). Action Observation Activates Premotor and Parietal Areas in a Somatotopic Manner: An fMRI Study. In *Social Neuroscience*. Psychology Press.
- Bylinskii, Z., Goetschalckx, L., Newman, A., & Oliva, A. (2022). Memorability: An Image-Computable Measure of Information Utility. In B. Ionescu, W. A. Bainbridge, & N. Murray (Eds.), *Human Perception of Visual Information* (pp. 207–239). Springer International Publishing. https://doi.org/10.1007/978-3-030-81465-6_8
- Carandini, M. (2005). Do We Know What the Early Visual System Does? *Journal of Neuroscience*, 25(46), 10577–10597. <https://doi.org/10.1523/JNEUROSCI.3726-05.2005>
- Carlin, J. D., & Kriegeskorte, N. (2017). Adjudicating between face-coding models with individual-face fMRI responses. *PLOS Computational Biology*, 13(7), e1005604.
<https://doi.org/10.1371/journal.pcbi.1005604>

- Chen, Z., Qing, J., & Zhou, J. H. (2023). *Cinematic Mindscapes: High-quality Video Reconstruction from Brain Activity* (arXiv:2305.11675). arXiv. <https://doi.org/10.48550/arXiv.2305.11675>
- Cichy, R. M., Dwivedi, K., Lahner, B., Lascelles, A., Iamshchinina, P., Graumann, M., Andonian, A., Murty, N. A. R., Kay, K., Roig, G., & Oliva, A. (2021). *The Algonauts Project 2021 Challenge: How the Human Brain Makes Sense of a World in Motion*. <https://doi.org/10.48550/ARXIV.2104.13714>
- Cichy, R. M., Khosla, A., Pantazis, D., Torralba, A., & Oliva, A. (2016). Comparison of deep neural networks to spatio-temporal cortical dynamics of human visual object recognition reveals hierarchical correspondence. *Scientific Reports*, *6*(1), 27755. <https://doi.org/10.1038/srep27755>
- Cichy, R. M., Pantazis, D., & Oliva, A. (2016). Similarity-Based Fusion of MEG and fMRI Reveals Spatio-Temporal Dynamics in Human Cortex During Visual Object Recognition. *Cerebral Cortex*, *26*(8), 3563–3579. <https://doi.org/10.1093/cercor/bhw135>
- Cichy, R. M., Roig, G., Andonian, A., Dwivedi, K., Lahner, B., Lascelles, A., Mohsenzadeh, Y., Ramakrishnan, K., & Oliva, A. (2019). *The Algonauts Project: A Platform for Communication between the Sciences of Biological and Artificial Intelligence*. <https://doi.org/10.48550/ARXIV.1905.05675>
- DeYoe, E. A., & Van Essen, D. C. (1988). Concurrent processing streams in monkey visual cortex. *Trends in Neurosciences*, *11*(5), 219–226. [https://doi.org/10.1016/0166-2236\(88\)90130-0](https://doi.org/10.1016/0166-2236(88)90130-0)
- DiCarlo, J. J., Zoccolan, D., & Rust, N. C. (2012). How Does the Brain Solve Visual Object Recognition? *Neuron*, *73*(3), 415–434. <https://doi.org/10.1016/j.neuron.2012.01.010>

- Dickie, E. W., Anticevic, A., Smith, D. E., Coalson, T. S., Manogaran, M., Calarco, N., Viviano, J. D., Glasser, M. F., Van Essen, D. C., & Voineskos, A. N. (2019). ciftify: A framework for surface-based analysis of legacy MR acquisitions. *NeuroImage*, *197*, 818–826. <https://doi.org/10.1016/j.neuroimage.2019.04.078>
- Doerig, A., Kietzmann, T. C., Allen, E., Wu, Y., Naselaris, T., Kay, K., & Charest, I. (2022). *Semantic scene descriptions as an objective of human vision*. <https://doi.org/10.48550/ARXIV.2209.11737>
- Eickenberg, M., Gramfort, A., Varoquaux, G., & Thirion, B. (2017). Seeing it all: Convolutional network layers map the function of the human visual system. *NeuroImage*, *152*, 184–194. <https://doi.org/10.1016/j.neuroimage.2016.10.001>
- Esteban, O., Markiewicz, C. J., Blair, R. W., Moodie, C. A., Isik, A. I., Erramuzpe, A., Kent, J. D., Goncalves, M., DuPre, E., Snyder, M., Oya, H., Ghosh, S. S., Wright, J., Durnez, J., Poldrack, R. A., & Gorgolewski, K. J. (2019). fMRIPrep: A robust preprocessing pipeline for functional MRI. *Nature Methods*, *16*(1), 111–116. <https://doi.org/10.1038/s41592-018-0235-4>
- Fairhall, S. L., Albi, A., & Melcher, D. (2014). Temporal Integration Windows for Naturalistic Visual Sequences. *PLoS ONE*, *9*(7), e102248. <https://doi.org/10.1371/journal.pone.0102248>
- Felleman, D. J., & Van Essen, D. C. (1991). Distributed hierarchical processing in the primate cerebral cortex. *Cerebral Cortex (New York, N.Y.)*, *1*(1), 1–47. <https://doi.org/10.1093/cercor/1.1.1-a>

Fonov, V., Evans, A., McKinstry, R., Almlı, C., & Collins, D. (2009). Unbiased nonlinear average age-appropriate brain templates from birth to adulthood. *NeuroImage*, *47*, S102.

[https://doi.org/10.1016/S1053-8119\(09\)70884-5](https://doi.org/10.1016/S1053-8119(09)70884-5)

Fox, C. J., Iaria, G., & Barton, J. J. S. (2009). Defining the face processing network: Optimization of the functional localizer in fMRI. *Human Brain Mapping*, *30*(5), 1637–1651.

<https://doi.org/10.1002/hbm.20630>

Glasser, M. F., Sotiropoulos, S. N., Wilson, J. A., Coalson, T. S., Fischl, B., Andersson, J. L., Xu, J., Jbabdi, S., Webster, M., Polimeni, J. R., Van Essen, D. C., & Jenkinson, M. (2013). The minimal preprocessing pipelines for the Human Connectome Project. *NeuroImage*, *80*,

105–124. <https://doi.org/10.1016/j.neuroimage.2013.04.127>

Goyal, R., Ebrahimi Kahou, S., Michalski, V., Materzynska, J., Westphal, S., Kim, H., Haenel, V., Freund, I., Yianilos, P., Mueller-Freitag, M., Hoppe, F., Thureau, C., Bax, I., & Memisevic, R. (2017). *The “Something Something” Video Database for Learning and Evaluating Visual Common Sense*. 5842–5850.

https://openaccess.thecvf.com/content_iccv_2017/html/Goyal_The_Something_Something_ICCV_2017_paper.html

Güçlü, U., & van Gerven, M. A. J. (2017). Increasingly complex representations of natural movies across the dorsal stream are shared between subjects. *NeuroImage*, *145*, 329–336.

<https://doi.org/10.1016/j.neuroimage.2015.12.036>

Han, J., Chen, C., Shao, L., Hu, X., Han, J., & Liu, T. (2015). Learning Computational Models of Video Memorability from fMRI Brain Imaging. *IEEE Transactions on Cybernetics*, *45*(8),

1692–1703. <https://doi.org/10.1109/TCYB.2014.2358647>

- Hanke, M., Adelhöfer, N., Kottke, D., Iacovella, V., Sengupta, A., Kaule, F. R., Nigbur, R., Waite, A. Q., Baumgartner, F., & Stadler, J. (2016). A studyforrest extension, simultaneous fMRI and eye gaze recordings during prolonged natural stimulation. *Scientific Data*, *3*(1), Article 1. <https://doi.org/10.1038/sdata.2016.92>
- Hardwick, R. M., Caspers, S., Eickhoff, S. B., & Swinnen, S. P. (2018). Neural correlates of action: Comparing meta-analyses of imagery, observation, and execution. *Neuroscience & Biobehavioral Reviews*, *94*, 31–44. <https://doi.org/10.1016/j.neubiorev.2018.08.003>
- Hasson, U., Furman, O., Clark, D., Dudai, Y., & Davachi, L. (2008). Enhanced Intersubject Correlations during Movie Viewing Correlate with Successful Episodic Encoding. *Neuron*, *57*(3), 452–462. <https://doi.org/10.1016/j.neuron.2007.12.009>
- Hasson, U., Yang, E., Vallines, I., Heeger, D. J., & Rubin, N. (2008). A Hierarchy of Temporal Receptive Windows in Human Cortex. *Journal of Neuroscience*, *28*(10), 2539–2550. <https://doi.org/10.1523/JNEUROSCI.5487-07.2008>
- Hebart, M. N., Contier, O., Teichmann, L., Rockter, A. H., Zheng, C. Y., Kidder, A., Corriveau, A., Vaziri-Pashkam, M., & Baker, C. I. (2023). THINGS-data, a multimodal collection of large-scale datasets for investigating object representations in human brain and behavior. *eLife*, *12*, e82580. <https://doi.org/10.7554/eLife.82580>
- Hebart, M. N., Dickter, A. H., Kidder, A., Kwok, W. Y., Corriveau, A., Van Wicklin, C., & Baker, C. I. (2019). THINGS: A database of 1,854 object concepts and more than 26,000 naturalistic object images. *PLOS ONE*, *14*(10), e0223792. <https://doi.org/10.1371/journal.pone.0223792>

- Ho, J., Chan, W., Saharia, C., Whang, J., Gao, R., Gritsenko, A., Kingma, D. P., Poole, B., Norouzi, M., Fleet, D. J., & Salimans, T. (2022). *Imagen Video: High Definition Video Generation with Diffusion Models* (arXiv:2210.02303). arXiv.
<https://doi.org/10.48550/arXiv.2210.02303>
- Jaegle, A., Mehrpour, V., Mohsenzadeh, Y., Meyer, T., Oliva, A., & Rust, N. (2019). Population response magnitude variation in inferotemporal cortex predicts image memorability. *eLife*, *8*, e47596. <https://doi.org/10.7554/eLife.47596>
- Ju, C., Han, T., Zheng, K., Zhang, Y., & Xie, W. (2022). Prompting Visual-Language Models for Efficient Video Understanding. In S. Avidan, G. Brostow, M. Cissé, G. M. Farinella, & T. Hassner (Eds.), *Computer Vision – ECCV 2022* (pp. 105–124). Springer Nature Switzerland. https://doi.org/10.1007/978-3-031-19833-5_7
- Julian, J. B., Fedorenko, E., Webster, J., & Kanwisher, N. (2012). An algorithmic method for functionally defining regions of interest in the ventral visual pathway. *NeuroImage*, *60*(4), 2357–2364. <https://doi.org/10.1016/j.neuroimage.2012.02.055>
- Kamitani, Y., & Tong, F. (2006). Decoding seen and attended motion directions from activity in the human visual cortex. *Current Biology*, *16*(11), 1096–1102.
<https://doi.org/10.1016/j.cub.2006.04.003>
- Kanwisher, N. (2010). Functional specificity in the human brain: A window into the functional architecture of the mind. *Proceedings of the National Academy of Sciences*, *107*(25), 11163–11170. <https://doi.org/10.1073/pnas.1005062107>

- Kanwisher, N., Khosla, M., & Dobs, K. (2023). Using artificial neural networks to ask 'why' questions of minds and brains. *Trends in Neurosciences*, *46*(3), 240–254.
<https://doi.org/10.1016/j.tins.2022.12.008>
- Kanwisher, N., McDermott, J., & Chun, M. M. (1997). The Fusiform Face Area: A Module in Human Extrastriate Cortex Specialized for Face Perception. *The Journal of Neuroscience*, *17*(11), 4302–4311. <https://doi.org/10.1523/JNEUROSCI.17-11-04302.1997>
- Kay, K., Jamison, K. W., Zhang, R.-Y., & Uğurbil, K. (2020). A temporal decomposition method for identifying venous effects in task-based fMRI. *Nature Methods*, *17*(10), Article 10.
<https://doi.org/10.1038/s41592-020-0941-6>
- Kay, W., Carreira, J., Simonyan, K., Zhang, B., Hillier, C., Vijayanarasimhan, S., Viola, F., Green, T., Back, T., Natsev, P., Suleyman, M., & Zisserman, A. (2017). *The Kinetics Human Action Video Dataset* (arXiv:1705.06950). arXiv. <https://doi.org/10.48550/arXiv.1705.06950>
- Kiebel, S. J., Daunizeau, J., & Friston, K. J. (2008). A Hierarchy of Time-Scales and the Brain. *PLoS Computational Biology*, *4*(11), e1000209. <https://doi.org/10.1371/journal.pcbi.1000209>
- Kietzmann, T. C., Spoerer, C. J., Sörensen, L. K. A., Cichy, R. M., Hauk, O., & Kriegeskorte, N. (2019). Recurrence is required to capture the representational dynamics of the human visual system. *Proceedings of the National Academy of Sciences*, *116*(43), 21854–21863.
<https://doi.org/10.1073/pnas.1905544116>
- Koivisto, M., Railo, H., Revonsuo, A., Vanni, S., & Salminen-Vaparanta, N. (2011). Recurrent Processing in V1/V2 Contributes to Categorization of Natural Scenes. *Journal of Neuroscience*, *31*(7), 2488–2492. <https://doi.org/10.1523/JNEUROSCI.3074-10.2011>

- Konen, C. S., & Kastner, S. (2008). Representation of Eye Movements and Stimulus Motion in Topographically Organized Areas of Human Posterior Parietal Cortex. *Journal of Neuroscience*, 28(33), 8361–8375. <https://doi.org/10.1523/JNEUROSCI.1930-08.2008>
- Kosakowski, H. L., Cohen, M. A., Takahashi, A., Keil, B., Kanwisher, N., & Saxe, R. (2022). Selective responses to faces, scenes, and bodies in the ventral visual pathway of infants. *Current Biology*, 32(2), 265-274.e5. <https://doi.org/10.1016/j.cub.2021.10.064>
- Krekelberg, B., Dannenberg, S., Hoffmann, K.-P., Bremmer, F., & Ross, J. (2003). Neural correlates of implied motion. *Nature*, 424(6949), 674–677. <https://doi.org/10.1038/nature01852>
- Kret, M. E., Pichon, S., Grèzes, J., & de Gelder, B. (2011). Similarities and differences in perceiving threat from dynamic faces and bodies. An fMRI study. *NeuroImage*, 54(2), 1755–1762. <https://doi.org/10.1016/j.neuroimage.2010.08.012>
- Kriegeskorte, N. (2008). Representational similarity analysis – connecting the branches of systems neuroscience. *Frontiers in Systems Neuroscience*. <https://doi.org/10.3389/neuro.06.004.2008>
- Kriegeskorte, N. (2015). Deep Neural Networks: A New Framework for Modeling Biological Vision and Brain Information Processing. *Annual Review of Vision Science*, 1(1), 417–446. <https://doi.org/10.1146/annurev-vision-082114-035447>
- Kriegeskorte, N., & Douglas, P. K. (2019). Interpreting encoding and decoding models. *Current Opinion in Neurobiology*, 55, 167–179. <https://doi.org/10.1016/j.conb.2019.04.002>
- Kriegeskorte, N., Goebel, R., & Bandettini, P. (2006). Information-based functional brain mapping. *Proceedings of the National Academy of Sciences*, 103(10), 3863–3868. <https://doi.org/10.1073/pnas.0600244103>

- Kriegeskorte, N., & Kievit, R. A. (2013). Representational geometry: Integrating cognition, computation, and the brain. *Trends in Cognitive Sciences*, 17(8), 401–412.
<https://doi.org/10.1016/j.tics.2013.06.007>
- Krizhevsky, A., Sutskever, I., & Hinton, G. E. (2017). ImageNet classification with deep convolutional neural networks. *Communications of the ACM*, 60(6), 84–90.
<https://doi.org/10.1145/3065386>
- Kubilius, J., Schrimpf, M., Kar, K., Rajalingham, R., Hong, H., Majaj, N., Issa, E., Bashivan, P., Prescott-Roy, J., Schmidt, K., Nayebi, A., Bear, D., Yamins, D. L., & DiCarlo, J. J. (2019). Brain-Like Object Recognition with High-Performing Shallow Recurrent ANNs. *Advances in Neural Information Processing Systems*, 32.
<https://proceedings.neurips.cc/paper/2019/hash/7813d1590d28a7dd372ad54b5d29d033-Abstract.html>
- Lafer-Sousa, R., Conway, B. R., & Kanwisher, N. G. (2016). Color-Biased Regions of the Ventral Visual Pathway Lie between Face- and Place-Selective Regions in Humans, as in Macaques. *Journal of Neuroscience*, 36(5), 1682–1697.
<https://doi.org/10.1523/JNEUROSCI.3164-15.2016>
- Lage-Castellanos, A., Valente, G., Formisano, E., & Martino, F. D. (2019). Methods for computing the maximum performance of computational models of fMRI responses. *PLOS Computational Biology*, 15(3), e1006397. <https://doi.org/10.1371/journal.pcbi.1006397>
- Li, Y., Song, Y., & Luo, J. (2017). *Improving Pairwise Ranking for Multi-Label Image Classification*. 3617–3625.

- https://openaccess.thecvf.com/content_cvpr_2017/html/Li_Improving_Pairwise_Ranking_CVPR_2017_paper.html
- Lin, J., Gan, C., & Han, S. (2019). *TSM: Temporal Shift Module for Efficient Video Understanding*. 7083–7093.
- https://openaccess.thecvf.com/content_ICCV_2019/html/Lin_TSM_Temporal_Shift_Module_for_Efficient_Video_Understanding_ICCV_2019_paper.html
- Lingnau, A., & Downing, P. E. (2015). The lateral occipitotemporal cortex in action. *Trends in Cognitive Sciences*, 19(5), 268–277. <https://doi.org/10.1016/j.tics.2015.03.006>
- Liu, D., Li, Q., Dinh, A.-D., Jiang, T., Shah, M., & Xu, C. (2023). *Diffusion Action Segmentation*. 10139–10149.
- https://openaccess.thecvf.com/content/ICCV2023/html/Liu_Diffusion_Action_Segmentation_ICCV_2023_paper.html
- Logothetis, N. K., & Sheinberg, D. L. (1996). Visual object recognition. *Annual Review of Neuroscience*, 19, 577–621. <https://doi.org/10.1146/annurev.ne.19.030196.003045>
- Maaz, M., Rasheed, H., Khan, S., & Khan, F. S. (2023). *Video-ChatGPT: Towards Detailed Video Understanding via Large Vision and Language Models* (arXiv:2306.05424). arXiv.
- <https://doi.org/10.48550/arXiv.2306.05424>
- McMahon, E., Bonner, M. F., & Isik, L. (2023). Hierarchical organization of social action features along the lateral visual pathway. *Current Biology*, 33(23), 5035-5047.e8.
- <https://doi.org/10.1016/j.cub.2023.10.015>
- Miech, A., Zhukov, D., Alayrac, J.-B., Tapaswi, M., Laptev, I., & Sivic, J. (2019). *HowTo100M: Learning a Text-Video Embedding by Watching Hundred Million Narrated Video Clips*.

2630–2640.

https://openaccess.thecvf.com/content_ICCV_2019/html/Miech_HowTo100M_Learning_a_Text-Video_Embedding_by_Watching_Hundred_Million_Narrated_ICCV_2019_paper.html

Mineault, P., Bakhtiari, S., Richards, B., & Pack, C. (2021). Your head is there to move you around: Goal-driven models of the primate dorsal pathway. *Advances in Neural Information Processing Systems*, *34*, 28757–28771.

<https://proceedings.neurips.cc/paper/2021/hash/f1676935f9304b97d59b0738289d2e22-Abstract.html>

Misaki, M., Luh, W.-M., & Bandettini, P. A. (2013). Accurate decoding of sub-TR timing differences in stimulations of sub-voxel regions from multi-voxel response patterns. *NeuroImage*, *66*, 623–633. <https://doi.org/10.1016/j.neuroimage.2012.10.069>

Monfort, M., Pan, B., Ramakrishnan, K., Andonian, A., McNamara, B. A., Lascelles, A., Fan, Q., Gutfreund, D., Feris, R. S., & Oliva, A. (2022). Multi-Moments in Time: Learning and Interpreting Models for Multi-Action Video Understanding. *IEEE Transactions on Pattern Analysis and Machine Intelligence*, *44*(12), 9434–9445. <https://doi.org/10.1109/TPAMI.2021.3126682>

Monfort, M., Vondrick, C., Oliva, A., Andonian, A., Zhou, B., Ramakrishnan, K., Bargal, S. A., Yan, T., Brown, L., Fan, Q., & Gutfreund, D. (2020). Moments in Time Dataset: One Million Videos for Event Understanding. *IEEE Transactions on Pattern Analysis and Machine Intelligence*, *42*(2), 502–508. <https://doi.org/10.1109/TPAMI.2019.2901464>

- Naselaris, T., Kay, K. N., Nishimoto, S., & Gallant, J. L. (2011). Encoding and decoding in fMRI. *NeuroImage*, *56*(2), 400–410. <https://doi.org/10.1016/j.neuroimage.2010.07.073>
- Newman, A., Fosco, C., Casser, V., Lee, A., McNamara, B., & Oliva, A. (2020). Multimodal Memorability: Modeling Effects of Semantics and Decay on Video Memorability. In A. Vedaldi, H. Bischof, T. Brox, & J.-M. Frahm (Eds.), *Computer Vision – ECCV 2020* (pp. 223–240). Springer International Publishing. https://doi.org/10.1007/978-3-030-58517-4_14
- Nili, H., Wingfield, C., Walther, A., Su, L., Marslen-Wilson, W., & Kriegeskorte, N. (2014). A Toolbox for Representational Similarity Analysis. *PLOS Computational Biology*, *10*(4), e1003553. <https://doi.org/10.1371/journal.pcbi.1003553>
- Nishimoto, S., & Gallant, J. L. (2011). A Three-Dimensional Spatiotemporal Receptive Field Model Explains Responses of Area MT Neurons to Naturalistic Movies. *Journal of Neuroscience*, *31*(41), 14551–14564. <https://doi.org/10.1523/JNEUROSCI.6801-10.2011>
- Nishimoto, S., Vu, A. T., Naselaris, T., Benjamini, Y., Yu, B., & Gallant, J. L. (2011). Reconstructing Visual Experiences from Brain Activity Evoked by Natural Movies. *Current Biology*, *21*(19), 1641–1646. <https://doi.org/10.1016/j.cub.2011.08.031>
- Orlov, T., & Zohary, E. (2018). Object Representations in Human Visual Cortex Formed Through Temporal Integration of Dynamic Partial Shape Views. *Journal of Neuroscience*, *38*(3), 659–678. <https://doi.org/10.1523/JNEUROSCI.1318-17.2017>
- Pascual-Leone, A., & Walsh, V. (2001). Fast backprojections from the motion to the primary visual area necessary for visual awareness. *Science*, *292*(5516), 510–512.

- Pitcher, D., Dilks, D. D., Saxe, R. R., Triantafyllou, C., & Kanwisher, N. (2011). Differential selectivity for dynamic versus static information in face-selective cortical regions. *NeuroImage*, *56*(4), 2356–2363. <https://doi.org/10.1016/j.neuroimage.2011.03.067>
- Pitcher, D., & Ungerleider, L. G. (2021). Evidence for a Third Visual Pathway Specialized for Social Perception. *Trends in Cognitive Sciences*, *25*(2), 100–110. <https://doi.org/10.1016/j.tics.2020.11.006>
- Power, J. D., Mitra, A., Laumann, T. O., Snyder, A. Z., Schlaggar, B. L., & Petersen, S. E. (2014). Methods to detect, characterize, and remove motion artifact in resting state fMRI. *NeuroImage*, *84*, 320–341. <https://doi.org/10.1016/j.neuroimage.2013.08.048>
- Press, W. A., Brewer, A. A., Dougherty, R. F., Wade, A. R., & Wandell, B. A. (2001). Visual areas and spatial summation in human visual cortex. *Vision Research*, *41*(10), 1321–1332. [https://doi.org/10.1016/S0042-6989\(01\)00074-8](https://doi.org/10.1016/S0042-6989(01)00074-8)
- Prince, J. S., Charest, I., Kurzawski, J. W., Pyles, J. A., Tarr, M. J., & Kay, K. N. (2022). Improving the accuracy of single-trial fMRI response estimates using GLMsingle. *eLife*, *11*, e77599. <https://doi.org/10.7554/eLife.77599>
- Qing, Z., Zhang, S., Huang, Z., Wang, X., Wang, Y., Lv, Y., Gao, C., & Sang, N. (2024). MAR: Masked Autoencoders for Efficient Action Recognition. *IEEE Transactions on Multimedia*, *26*, 218–233. <https://doi.org/10.1109/TMM.2023.3263288>
- Ratan Murty, N. A., Bashivan, P., Abate, A., DiCarlo, J. J., & Kanwisher, N. (2021). Computational models of category-selective brain regions enable high-throughput tests of selectivity. *Nature Communications*, *12*(1), 5540. <https://doi.org/10.1038/s41467-021-25409-6>

- Reimers, N., & Gurevych, I. (2019). *Sentence-BERT: Sentence Embeddings using Siamese BERT-Networks*. <https://doi.org/10.48550/ARXIV.1908.10084>
- Ress, D., & Heeger, D. J. (2003). Neuronal correlates of perception in early visual cortex. *Nature Neuroscience*, 6(4), Article 4. <https://doi.org/10.1038/nn1024>
- Roe, A. W., Chelazzi, L., Connor, C. E., Conway, B. R., Fujita, I., Gallant, J. L., Lu, H., & Vanduffel, W. (2012). Toward a Unified Theory of Visual Area V4. *Neuron*, 74(1), 12–29. <https://doi.org/10.1016/j.neuron.2012.03.011>
- Russakovsky, O., Deng, J., Su, H., Krause, J., Satheesh, S., Ma, S., Huang, Z., Karpathy, A., Khosla, A., Bernstein, M., Berg, A. C., & Fei-Fei, L. (2015). ImageNet Large Scale Visual Recognition Challenge. *International Journal of Computer Vision*, 115(3), 211–252. <https://doi.org/10.1007/s11263-015-0816-y>
- Schneider, W. X. (2013). Selective visual processing across competition episodes: A theory of task-driven visual attention and working memory. *Philosophical Transactions of the Royal Society B: Biological Sciences*, 368(1628), 20130060. <https://doi.org/10.1098/rstb.2013.0060>
- Schrimpf, M., Kubilius, J., Hong, H., Majaj, N. J., Rajalingham, R., Issa, E. B., Kar, K., Bashivan, P., Prescott-Roy, J., Geiger, F., Schmidt, K., Yamins, D. L. K., & DiCarlo, J. J. (2020). *Brain-Score: Which Artificial Neural Network for Object Recognition is most Brain-Like?* (p. 407007). bioRxiv. <https://doi.org/10.1101/407007>
- Schrimpf, M., Kubilius, J., Lee, M. J., Ratan Murty, N. A., Ajemian, R., & DiCarlo, J. J. (2020). Integrative Benchmarking to Advance Neurally Mechanistic Models of Human Intelligence. *Neuron*, 108(3), 413–423. <https://doi.org/10.1016/j.neuron.2020.07.040>

- Schultz, J., & Pilz, K. S. (2009). Natural facial motion enhances cortical responses to faces. *Experimental Brain Research*, 194(3), 465–475. <https://doi.org/10.1007/s00221-009-1721-9>
- Seeliger, K., Sommers, R. P., Güçlü, U., Bosch, S. E., & Gerven, M. A. J. van. (2019). *A large single-participant fMRI dataset for probing brain responses to naturalistic stimuli in space and time* (p. 687681). bioRxiv. <https://doi.org/10.1101/687681>
- Senior, C., Barnes, J., Giampietroc, V., Simmons, A., Bullmore, E. T., Brammer, M., & David, A. S. (2000). The functional neuroanatomy of implicit-motion perception or ‘representational momentum.’ *Current Biology*, 10(1), 16–22. [https://doi.org/10.1016/S0960-9822\(99\)00259-6](https://doi.org/10.1016/S0960-9822(99)00259-6)
- Shirai, N., & Imura, T. (2014). Implied motion perception from a still image in infancy. *Experimental Brain Research*, 232(10), 3079–3087. <https://doi.org/10.1007/s00221-014-3996-8>
- Silvanto, J., Cowey, A., Lavie, N., & Walsh, V. (2005). Striate cortex (V1) activity gates awareness of motion. *Nature Neuroscience*, 8(2), Article 2. <https://doi.org/10.1038/nn1379>
- Silvanto, J., Lavie, N., & Walsh, V. (2005). Double Dissociation of V1 and V5/MT activity in Visual Awareness. *Cerebral Cortex*, 15(11), 1736–1741. <https://doi.org/10.1093/cercor/bhi050>
- Silver, M. A., & Kastner, S. (2009). Topographic maps in human frontal and parietal cortex. *Trends in Cognitive Sciences*, 13(11), 488–495. <https://doi.org/10.1016/j.tics.2009.08.005>
- Singer, U., Polyak, A., Hayes, T., Yin, X., An, J., Zhang, S., Hu, Q., Yang, H., Ashual, O., Gafni, O., Parikh, D., Gupta, S., & Taigman, Y. (2022). *Make-A-Video: Text-to-Video Generation*

- without Text-Video Data* (arXiv:2209.14792). arXiv.
<https://doi.org/10.48550/arXiv.2209.14792>
- Smith, A. T., Greenlee, M. W., Singh, K. D., Kraemer, F. M., & Hennig, J. (1998). The Processing of First- and Second-Order Motion in Human Visual Cortex Assessed by Functional Magnetic Resonance Imaging (fMRI). *Journal of Neuroscience*, *18*(10), 3816–3830.
<https://doi.org/10.1523/JNEUROSCI.18-10-03816.1998>
- Soomro, K., Zamir, A. R., & Shah, M. (2012). *UCF101: A Dataset of 101 Human Actions Classes From Videos in The Wild* (arXiv:1212.0402). arXiv.
<https://doi.org/10.48550/arXiv.1212.0402>
- Tong, Z., Song, Y., Wang, J., & Wang, L. (2022). VideoMAE: Masked Autoencoders are Data-Efficient Learners for Self-Supervised Video Pre-Training. *Advances in Neural Information Processing Systems*, *35*, 10078–10093.
- Wang, L., Xiong, Y., Wang, Z., Qiao, Y., Lin, D., Tang, X., & Van Gool, L. (2016). Temporal Segment Networks: Towards Good Practices for Deep Action Recognition. In B. Leibe, J. Matas, N. Sebe, & M. Welling (Eds.), *Computer Vision – ECCV 2016* (pp. 20–36). Springer International Publishing. https://doi.org/10.1007/978-3-319-46484-8_2
- Watson, A. B., & Ahumada, A. J. (1985). Model of human visual-motion sensing. *JOSA A*, *2*(2), 322–342. <https://doi.org/10.1364/JOSAA.2.000322>
- Wittkuhn, L., & Schuck, N. W. (2021). Dynamics of fMRI patterns reflect sub-second activation sequences and reveal replay in human visual cortex. *Nature Communications*, *12*(1), Article 1. <https://doi.org/10.1038/s41467-021-21970-2>

- Wu, J. Z., Ge, Y., Wang, X., Lei, S. W., Gu, Y., Shi, Y., Hsu, W., Shan, Y., Qie, X., & Shou, M. Z. (2023). *Tune-A-Video: One-Shot Tuning of Image Diffusion Models for Text-to-Video Generation*. 7623–7633. https://openaccess.thecvf.com/content/ICCV2023/html/Wu_Tune-A-Video_One-Shot_Tuning_of_Image_Diffusion_Models_for_Text-to-Video_Generation_ICCV_2023_paper.html
- Wurm, M. F., & Caramazza, A. (2022). Two ‘what’ pathways for action and object recognition. *Trends in Cognitive Sciences*, 26(2), 103–116. <https://doi.org/10.1016/j.tics.2021.10.003>
- Wurm, M. F., Caramazza, A., & Lingnau, A. (2017). Action Categories in Lateral Occipitotemporal Cortex Are Organized Along Sociality and Transitivity. *Journal of Neuroscience*, 37(3), 562–575. <https://doi.org/10.1523/JNEUROSCI.1717-16.2016>
- Yamins, D. L., Hong, H., Cadieu, C., & DiCarlo, J. J. (2013). Hierarchical Modular Optimization of Convolutional Networks Achieves Representations Similar to Macaque IT and Human Ventral Stream. *Advances in Neural Information Processing Systems*, 26. <https://proceedings.neurips.cc/paper/2013/hash/9a1756fd0c741126d7bbd4b692ccbd91-Abstract.html>
- Yamins, D. L. K., Hong, H., Cadieu, C. F., Solomon, E. A., Seibert, D., & DiCarlo, J. J. (2014). Performance-optimized hierarchical models predict neural responses in higher visual cortex. *Proceedings of the National Academy of Sciences*, 111(23), 8619–8624. <https://doi.org/10.1073/pnas.1403112111>

- Yildirim, I., Wu, J., Kanwisher, N., & Tenenbaum, J. (2019). An integrative computational architecture for object-driven cortex. *Current Opinion in Neurobiology*, *55*, 73–81. <https://doi.org/10.1016/j.conb.2019.01.010>
- Zhang, H., Li, X., & Bing, L. (2023). *Video-LLaMA: An Instruction-tuned Audio-Visual Language Model for Video Understanding* (arXiv:2306.02858). arXiv. <https://doi.org/10.48550/arXiv.2306.02858>
- Zheng, C., Wu, W., Chen, C., Yang, T., Zhu, S., Shen, J., Kehtarnavaz, N., & Shah, M. (2023). Deep Learning-based Human Pose Estimation: A Survey. *ACM Computing Surveys*, *56*(1), 11:1-11:37. <https://doi.org/10.1145/3603618>
- Zhou, B., Lapedriza, A., Khosla, A., Oliva, A., & Torralba, A. (2018). Places: A 10 Million Image Database for Scene Recognition. *IEEE Transactions on Pattern Analysis and Machine Intelligence*, *40*(6), 1452–1464. <https://doi.org/10.1109/TPAMI.2017.2723009>

Reviewer #1 (Remarks to the Author):

In my view the authors have done a great job in addressing the comments I raised in the previous round. I believe that the revised version has indeed greatly improved both in terms of readability and the embedding into the existing literature, and that it is going to be of interest for the broader research community, as also pointed out by R2. I only have a very minor remaining comment (which does not require an additional review from my side): on page 15, the authors provide the link to a GitHub depository, but that link does not appear to be valid anymore.

Reviewer #2 (Remarks to the Author):

I really appreciate the authors' effort in addressing my questions. They provide extensively new results to support the claims. Overall, the revised manuscript solved has improved a lot and addressed most of my concerns. I only have a few additional comments for the response.

1. The authors provide varied results about the "dynamics". For example, in the response to Comment 3 & 5, there are many investigations on identifying significant regions w.r.t the encoded dynamics. My questions here are 1) are they consistent? 2) how to link these results together? The authors mentioned some meta-information from human behavior and the temporal dynamics of the BOLD signal. The relationship between these two parts should be verified.
2. On page 47 in the response letter, the authors state that "We conclude that the TSM vs TSN analysis and the frame shuffling analysis are perturbing different temporal dynamics". What does the temporal dynamics mean indeed?
3. On page 49 in the response letter, the authors respond to the reliability of the TR peak estimation. The p-value here is not feasible. The p-value only gives that it may exist but it does not give how reliable it is. The authors should calculate the confidence interval instead. If it is overly wide, the discussion on the position of the TR peak may not be meaningful enough then.

Reviewer #2 (Remarks on code availability):

The codes are accessible under the directory and the codes are organized. Maybe it would be better if more instructions are provided within the scripts. However, I didn't install and run the application.

Dear Reviewers,

Thank you for assessing our previous submission, and we are pleased that we were able to address the majority of your comments. Below we address your remaining comments. Revised text is tracked in the revised manuscript in **red**. We also provide a clean version of the manuscript without any comments or tracked changes. Any revised text from the manuscript reproduced below will also be shown in **red bold font**. Your reviewer comments are reproduced in unbolded **green** text, and our responses are in black normal font.

Reviewer #1 (Remarks to the Author):

In my view the authors have done a great job in addressing the comments I raised in the previous round. I believe that the revised version has indeed greatly improved both in terms of readability and the embedding into the existing literature, and that it is going to be of interest for the broader research community, as also pointed out by R2. I only have a very minor remaining comment (which does not require an additional review from my side): on page 15, the authors provide the link to a GitHub depository, but that link does not appear to be valid anymore.

We are glad to hear that you were satisfied with the previous revisions, and thank you again for your helpful feedback. Regarding the GitHub repository for the ResNet50 model (GitHub - pbw-Berwin/M4-pretrained), the link is valid but the repository access settings were set to “private”, meaning only us authors have access to it. We have since made the repository “public” so you (and anyone) can view it. This repository’s privacy setting slipped our mind, and we apologize for not providing you with an alternative way to view the model training code and weights.

Reviewer #2 (Remarks to the Author):

I really appreciate the authors’ effort in addressing my questions. They provide extensively new results to support the claims. Overall, the revised manuscript solved has improved a lot and addressed most of my concerns. I only have a few additional comments for the response.

1. The authors provide varied results about the “dynamics”. For example, in the response to Comment 3 & 5, there are many investigations on identifying significant regions w.r.t the encoded dynamics. My questions here are 1) are they consistent? 2) how to link these results together? The authors mentioned some meta-information from human behavior and the temporal dynamics of the BOLD signal. The relationship between these two parts should be verified.

We respond to your comment in three points regarding our results’ (1) consistency, (2) linking the results together, and (3) verifying the relationship between our meta-information and temporal dynamics of the BOLD signal. We reproduce the frame shuffling, encoding model, and TSM vs TSN figures at the bottom of this comment for you to reference at your convenience.

(1) *Consistency*:

Our results in this manuscript, and more specifically in response to the previous revision comments, consistently tell us that BMD's brain responses capture temporal properties of the videos. We introduced a number of experiments (a) constraining the model's ability to process a video's temporal properties (e.g., TSM vs TSN), (b) perturbing different temporal properties of the stimuli themselves (e.g., frame shuffling), and (c) predicting brain activity using different modeling architectures and pretraining diets. Since these experiments themselves were not intended to be related, their results in terms of significant ROIs and block depth are not related. However, within an experiment, we agree that we should expect to observe consistent results.

Starting with (a) the TSM vs TSN analysis, the results neatly convey early visual ROIs are most affected by restricting the sharing of information across frames. We believe this was our most controlled experiment as the difference between the TSM and TSN networks only differed in the frame sharing aspect.

(b) In the frame shuffling analyses, we did expect to see the results of the MobileNetV2 trained on Kinetics-400 align more closely with the results of the ResNet50 trained on M4. We hypothesize that the pretraining dataset's temporal information content heavily influenced the results. Kinetics-400 has notoriously poor temporal information between frames (i.e., a model can easily predict the video's action from a single frame) while the Multi-Moments in Time dataset (used in M4) has notably high temporal information content. We expect that the TimeSformer results also did not align with the results of the ResNet50 trained on M4 because its highly non-linear attention layers reduced the impact of frame shuffling.

(c) The encoding model analysis showcased the prediction accuracies of the three models explained above in (b) and their hierarchical correspondence (first layer minus last layer). The results for the ResNet50 model trained on M4 and TimeSformer model trained on HowTo100M aligned more closely than that of the MobileNetV2 model. This suggests that the Kinetics-400 pretraining dataset again influenced MobileNetV2's performance. However, future large-scale experiments that are beyond the scope of this manuscript are needed to confirm this hypothesis

In sum, our results consistently show that BMD captures temporal information from the video, and that predicting this information depends on complex relationships between at least model architecture and pretraining datasets.

We edit a sentence in section "fMRI responses capture temporal event structure" to read:

We find that the TSM ResNet50 is the only model of the three that sees robust effects in most ROIs and DNN blocks (see Supplementary Fig. 10 for all results), implying that model architecture and the level of temporal information in the model training datasets may be closely tied to the effects of frame shuffling.

(2) Linking the results together:

Our set of experiments perturbed various aspects of temporal information contained in the video. For example, the TSM vs TSN analysis restricted models in how information can be

shared between frames, the frame shuffling analysis affected temporal properties of (at least) temporal continuity, motion direction, and camera (or head) motion, and the encoding models processed videos according to a temporally-based objective (action classification).

To unify these results, we focus on the results from the TSM ResNet50 model trained on M4 because the architecture is best motivated to be a model of the visual brain and BMD is a (non-overlapping) subset of the high temporal information M4 dataset.

In the highly controlled TSM vs TSN analysis, restricting the model's ability to share information across frames resulted in worse prediction in early visual ROIs by high-level blocks (blocks 3 and 4). This may have reduced the model's prediction accuracy in regions that relied on inter-frame information to parse low-level features like basic edges, for example. In higher-level categorically selective regions, such as EBA, information does not necessarily need to be shared across frames to accomplish its primary function of body recognition since that goal is relatively static.

The frame shuffling analysis affected a mix of early visual, ventral visual, and parietal cortex ROIs. Since frame shuffling affects multiple temporal properties of a video, we would expect it to impact more regions. Frame shuffling not only affects which information is shared across frames (and thus affects early visual regions, like above), but it also affects higher-level regions' ability to interpret the scene content. In Figure 7, we (and other work) provide evidence that the objective of the visual system is likely to be more similar to complex scene analysis than single-category recognition. A shuffled sequence of frames makes such scene analysis especially difficult.

Lastly, as an encoding model, we observe late (early) blocks are increasingly better (worse) brain predictors in hierarchically late ROIs. This result fits nicely with known properties of the visual system, such as increasingly large receptive fields that build toward more complex neural representations. One novelty in this result is that this trend extends deep into the parietal cortex towards an even more complex objective of action recognition.

In response part (3), we additionally link our metadata to temporal properties captured by BMD.

We add the following text to the Discussion to summarize the consistency of the results and how they link together:

For example, we perturb various temporal properties of the videos and use DNNs to predict the corresponding brain activity. We observe how frame shuffling, by interrupting (at least) the video's temporal continuity and motion direction, affected a mix of early visual, ventral visual, and parietal cortex ROIs (Fig. 5). Comparing the brain prediction performance between the Temporal Shift Module (TSM) architecture and the Temporal Shift Network (TSN) architecture (Supplementary Fig. 9) isolates the effect of sharing information across frames and shows significant differences predominantly in early visual ROIs. Our

encoding model optimized with a temporally-based objective function (action classification) demonstrates a correspondence in processing stages between video-computable DNN model blocks and cortical regions (Fig. 4), thereby both extending previous studies that analyzed responses to still images (Kubilius et al., 2019; Schrimpf, Kubilius, Lee, et al., 2020; D. L. Yamins et al., 2013) and showing we can accurately predict brain activity in the dorsal visual and parietal regions that are largely driven by a stimulus's dynamic properties (Gazzola & Keysers, 2009; Heim et al., 2012; Konen & Kastner, 2008; R. Peeters et al., 2009; R. R. Peeters et al., 2013). Lastly, we demonstrate that despite the sluggishness of the BOLD response, BMD allows tracking of visual information processing at the level of seconds (Fig. 6) (Hasson et al., 2004; Hasson, Yang, et al., 2008; Kiebel et al., 2008; Murray et al., 2014; Orlov & Zohary, 2018; Piasini et al., 2021). These analyses consistently demonstrate BMD's temporal content and thus present a unique opportunity to leverage existing models of social expressions (Hu et al., 2022; Kahou et al., 2016; Tzirakis et al., 2018), action recognition (Carreira & Zisserman, 2017; Feichtenhofer et al., 2016; Monfort et al., 2022), integration of temporal features (Bertasius et al., 2021; Ji et al., 2013; Y. Wang et al., 2023), and object detection (Fan et al., 2021; Shafiee et al., 2017) to study brain function.

(3) Relationship between meta-information and temporal dynamics of BOLD signal:

We first note the memorability analysis (Figure 8) exhibits patterns of correlations not seen in image-based memorability work, suggesting a relationship between the dynamic information in BMD's brain responses and the memorability meta-information. Second, we provide an additional supplementary analysis (Supplementary Figure 11, reproduced below) to verify the relationship between our semantic meta-information and the temporal dynamics of the BOLD signal. Building on the representational similarity analysis presented in Figure 7, we examine how well sentence descriptions of a *single video frame* correlate with the brain data compared to the sentence descriptions of the full 3 second video. In other words, we ask if our brain data (responses to dynamic videos) contains representations more similar to *dynamic video captions* or *static frame captions*. In brief, we find that the video captions significantly outperform the frame captions in high-level ventral and dorsal brain regions ($p < 0.05$, one sample t-test against a difference of 0, two-sided, Bonferroni corrected with $n=22$ ROIs). These results provide further evidence that the brain responses are not only capturing dynamic information content, but also this information content is reflected in our metadata. If BMD's brain responses were equivalent to that of a static image, we would expect the frame-level caption to perform just as well, if not better than, the video-level caption.

In detail, we use the state-of-the-art captioning model GIT (Wang et al., 2022) to generate 5 different captions of the middle frame of each video. We then feed these frame captions through the exact same analysis pipeline used in Figure 7. To exemplify the differences between the frame and video captions, we show you examples of the GIT frame captions and our provided video captions of the first test set video (video 1001):

Frame captions:

"a red monster truck driving through a parking lot."

"a red monster truck driving around a parking lot."

"a red monster truck driving down a track."

"a red monster truck driving through a parking lot"

"a red monster truck driving down a road."

Video captions:

"Giant monster truck in an open field with an audience reversing"

"What fun! The crowd has gathered around a fenced area where trucks on huge tires a parading around the fenced area."

"A large monster truck reverses on an outdoor concrete course with a crowd in the background."

"A large monster truck goes in reverse in front of a crowd"

"A monster pickup truck rolls backwards across the field at a rally."

We note that it is not immediately clear the video captions would outperform the frame captions. On one hand, the frame captions succinctly and accurately relate the object, action (to the best extent it can in a single frame), and scene, and the video captions contain potentially irrelevant or distracting information (e.g., "What fun!", "parading", "rally"). On the other hand, the video captions accurately capture the action (the truck here is "reversing", not "driving") and spatial information not immediately available in a frame (e.g., the "crowd" and "fenced area" are difficult to parse from the frame). Especially in the context of our results presented in Figure 7, we believe we have provided ample evidence our semantic metadata captures various aspects of BMD's dynamic brain responses. We additionally make available all GIT frame captions in the dataset release.

We add the following figure and caption to the supplementary 1 document:

Figure S11: Representational similarity of frame and video captions to fMRI responses.

A) ROI-based correlation: We correlate (Spearman's R) a representational dissimilarity matrix (RDM) derived from captions of short videos (purple) and captions of the middle frame of each video (beige) with an RDM at each voxel in the brain. The correlation is normalized by the voxel's upper noise ceiling. Noise-normalized correlations are averaged within each ROI and plotted for each individual subject. Five frame captions were computed from the image captioning GIT model version git-large-coco. The five video captions were human annotated and described in the "text descriptions" metadata section. Source data are provided as Source Data files. **B) Difference in correlations:** The difference between the video-fMRI normalized correlation and frame-fMRI normalized correlation for each subject was computed and plotted. Statistically significant ROIs are colored in green and marked with a black asterisk above ($p < 0.05$, one sample two-sided t-test against a null correlation of 0, Bonferroni corrected with $n = 22$ ROIs). Source data are provided as Source Data files. The box plots in both panels encompass the first and third data quartiles and the median (horizontal line). The whiskers extend to the minimum and maximum values within 1.5 times the interquartile range, and values falling outside that range are considered

outliers (denoted by a diamond). The overlaid points show the value at each observation (n=10 for all ROIs except TOS (n=8) RSC (n=9)).

We add the following paragraph to the main text section Semantic metadata reveal strong similarity between sentence-level descriptions and visual brain activity:

Lastly, we assess if a representation of single frame text descriptions (generated by GIT (Wang et al., 2022)) would correlate just as strongly as a representation of our full video text descriptions (Supplementary Figure 11). Although both sets of captions use sentences to describe the core elements of the video, the representation of the full video text descriptions correlates with the neural representations significantly better primarily throughout ventral visual cortex (V3v, hV4, EBA, FFA, OFA, STS, LOC, PPA, and V3ab). These results strongly suggest that BMD’s brain responses are not only capturing dynamic information content, but also this information content is reflected in the full video text descriptions.

We add a reference to this analysis in the Discussion section:

For example, the higher correlations between brain activity and complex sentence descriptions over word-level labels (see Fig. 7 and Supplementary Fig. 11) suggest that the function of these brain regions extends beyond object recognition (Doerig et al., 2022).

We add the Methods section “Metadata RSA analysis procedure”:

The RSA analysis comparing the sentence text descriptions of the full video with the sentence text descriptions of a single frame presented in Supplementary 1 Figure 11 follow a similar pipeline as above. First, we generate five different captions of the middle frame of each video using the captioning model GIT, version git-large-coco (Wang et al., 2022) (generation parameters max_length=100, num_beams=5, temperature=1, top_k=250, top_p=1). These captions are available alongside the human-annotated metadata but in a separate file.

Following the same pipeline as used for the full video sentence text descriptions above, we compute vector embeddings using Sentence-BERT (Reimers & Gurevych, 2019) for each of the five frame captions. We average the top 3 most similar captions and compute the pairwise cosine distance (1-cosine similarity) between each test set video’s averaged embedding to obtain a 102 x 102 RDM.

The frame text description RDM has the following Spearman correlations with the other metadata RDMs:

[frame text description: object]: 0.2451

[frame text description: action]: 0.1335

[frame text description: scene]: 0.2025
[frame text description: scene+object+action]: 0.2878
[frame text description: video text description]: 0.6653

We correlate (Spearman's R) the frame text description RDM with the searchlight-based neural RDMs at each voxel for each subject separately. We then normalize the correlation by each voxel's upper noise-ceiling and average the correlations within the ROI (Supplementary 1 Figure 11a). We compute the difference in correlation (Supplementary 1 Figure 11b) between the full video text description and the frame text description at the subject-level and compute statistical significance for each ROI against 0 correlation ($p < 0.05$, one sample two-sided t-test, Bonferroni corrected with $n = 22$ ROIs).

We add to the supplementary 1 discussion "The added value of a short video versus a static image neuroimaging dataset":

Concerning text descriptions, short videos can capture temporal sequences of events that an image cannot. **We contrast these video captions with captions of only each video's middle frame (frame captions generated by GIT (Wang et al., 2022)) below (emphasis our own):**

Video 0001:

- **Video caption:** "A mallard is in the water alone swimming around and putting its beak in."
- **Frame caption:** "A duck floating on top of a blue body of water."

Video 0002:

- **Video caption:** "A man is showing another man how to move feet back and forth."
- **Frame caption:** "a couple of men standing in a garage."

Video 0005:

- **Video caption:** "A woman guides a little boy's arms up and down as other kids stretch around him."
- **Frame caption:** "a group of children standing around a room."

Video 0006:

- **Video caption:** "a chess tournament is going on this is focused on two players one is moving their queen and taking something to put the king in checkmate"
- **Frame caption:** "a group of people sitting at tables playing chess."

Static frames of these videos cannot capture the temporal facts that the mallard is "putting its beak in", the man "is showing another man how", "a woman guides...as other kids stretch", and a chess player "is moving their queen and taking something to put the king in checkmate." This temporal information adds valuable context that often makes one's understanding of the 3s video vastly different compared to any one of its single static frames. [...] . **We further show**

that the full video captions lead to higher representational similarity with BMD's brain responses than the frame captions through much of the ventral visual cortex (Supplementary Fig. 11).

Supplementary F9: The effect of Temporal Shift Module (TSM) on brain prediction performance

Supplementary F7: Encoding model performance on BMD

Supplementary F10: The effect of frame shuffling on brain prediction performance across different architectures

2. On page 47 in the response letter, the authors state that “We conclude that the TSM vs TSN analysis and the frame shuffling analysis are perturbing different temporal dynamics”. What does the temporal dynamics mean indeed?

In this context we refer to temporal dynamics broadly as a brain process that utilizes some time-based property of the stimulus. For example, properties like a video’s temporal continuity, camera (or head) motion, time-based information content, and motion may be used in subsequent brain processes to predict information at a future time point, interpret intentions, establish a point of view, navigate the environment, etc.

The TSM and TSN architectures affect the video property of time-based information content, where the TSM model allows information to be shared across time (or frames) while TSN does not. We then say this analysis targets brain processes, or temporal dynamics in the brain, that rely upon such time-based information content.

The frame shuffling analysis affects numerous video properties, such as temporal continuity, time-based information content, camera/head motion, direction of object motion etc. It follows that brain processes that leverage those properties will be affected.

The results from the TSM vs TSN analysis and frame shuffling analysis show different patterns in how the ROI prediction accuracies are affected in different model blocks. Thus, we say that these analyses are perturbing different temporal dynamics in the brain, or brain processes that leverage various time-based stimulus properties.

We intentionally refrain from defining the temporal dynamics themselves (e.g., the temporal dynamic of “motion prediction”), as BMD is not an ideal dataset to disentangle the exact relationship between a stimulus property and brain processes. For this endeavor, we would recommend a controlled dataset using more simplistic stimuli that systematically varies these time-based properties. However, these analyses make us confident that the brain responses captured in BMD are not equivalent to brain responses evoked by static images - by disrupting time-based properties of the stimuli (frame shuffling) and constraining how a model can utilize these time-based features (TSM vs TSN), we observe better/worse brain prediction performance.

3. On page 49 in the response letter, the authors respond to the reliability of the TR peak estimation. The p-value here is not feasible. The p-value only gives that it may exist but it does not give how reliable it is. The authors should calculate the confidence interval instead. If it is overly wide, the discussion on the position of the TR peak may not be meaningful enough then.

Thank you for the comment! The peak TRs were estimated at both the subject level (Figure 6d and 6e, main panel, reproduced below) and subject-averaged group level (Figure 6e ROI panels

and supplementary Figure 6b, reproduced below). Below, we address the reliability of the TR peak estimation for both methods and soften some language used in the manuscript describing the results.

(1) Subject-level analysis (Figure 6d and 6e, main panel):

For each subject individually, the unique variance at each of their 9 TRs and 2 video epochs (first and third epoch) was computed, the TR with the maximum unique variance at each of the 2 video epochs was identified, and the difference between those two peak TRs was calculated. Thus, each voxel in the brain (Figure 6d) or ROI (Figure 6e, main panel) corresponded to a total of ten 'TR peak differences', one for each of the ten subjects. Per your suggestion, we compute a 95% confidence interval of these ROI peak differences using the Standard Error of the Mean (SEM) and a t-statistic with n-1 degrees of freedom. We display the confidence interval results in the table below. In summary, we observe a pattern of higher means and tighter confidence intervals in early visual ROIs. The confidence interval's lower bound is above a peak difference of 1 TR in V1v, V1d, V3v, V3d, and LOC, suggesting that in these ROIs we are confident the peak TR difference is at least 1 full TR. We add these confidence intervals in the corresponding Source Data files.

ROI	Mean Difference in Peak TR	95% Confidence Interval (lower, upper)	Confidence Interval Width (upper - lower)
V1v	2.3	(1.338, 3.262)	1.924
V1d	2.8	(1.966, 3.634)	1.668
V2v	1.8	(0.528, 3.072)	2.543
V2d	2.0	(0.162, 3.838)	3.675
V3v	2.2	(1.665, 2.735)	1.070
V3d	2.7	(2.141, 3.259)	1.117
hV4	0.9	(-0.978, 2.778)	3.755
EBA	2.2	(0.313, 4.087)	3.774
FFA	1.6	(0.578, 2.622)	2.043
OFA	2.1	(0.653, 3.547)	2.893
STS	1.4	(0.283, 2.517)	2.234
LOC	2.2	(1.012, 3.388)	2.376
PPA	1.1	(-0.482, 2.682)	3.163

RSC	0.55	(-1.109, 2.220)	3.329
TOS	1.375	(-1.200, 3.950)	5.149
V3ab	2.0	(0.720, 3.280)	2.559
IPS0	2.7	(0.675, 4.725)	4.049
IPS1-2-3	1.1	(-0.935, 3.135)	3.644
7AL	1.1	(-0.722, 2.922)	3.644
BA2	0.1	(-2.566, 2.766)	5.332
PFt	-1.3	(-3.273, 0.673)	3.946
PFop	-1.4	(-3.099, 0.299)	3.397

(2) *Subject-averaged group analysis (Figure 6e ROI panels and supplementary Figure 6b):*
An ROI's peak TR was identified as the TR was the highest subject-averaged unique variance for the first (blue star) and third (orange star) video epoch, computed separately. To quantify whether the identified peak TRs in an ROI are "reliable", we compute a 95% confidence interval around its average unique variance using the Standard Error of the Mean (SEM) and a t-statistic with n-1 degrees of freedom. If another TR's subject-averaged unique variance falls within this 95% confidence interval, we can say the originally identified peak TR was not reliable. This computation was done for each ROI and first/third video epoch independently. We find that in all ROIs at least one other TR's subject-averaged unique variance overlapped with the original peak TR's 95% confidence interval. These results suggest it is inappropriate to confidently identify a specific peak TR (e.g., to say, "In ROI V1v, the first video epoch peaks at TR 4"). Graphically, we observe a similar trend as the above subject-level analysis where the peak TR difference is most pronounced in early visual ROIs.

In conclusion, the results from Figures 6d and 6e converge to show the early visual cortex has statistically significant and highly confident peak TR differences. In light of these reliability analyses, we review our text to make sure we do not make any claims about the exact TR the first/third video epochs peak. We additionally edit the following text:

In section "BMD tracks the temporal dynamics unfolding within events":

Results highlight significant temporal delays ~~throughout the ventral and dorsal cortex, but~~ most pronounced in the early visual cortex. Equivalent ROI-based analysis (Figure 6e, ~~upper ROI panels and main panel, see Supplementary Fig. 6 for all ROIs~~) yielded a similar result pattern. ROIs in the ~~early and ventral~~ visual brain (14 of the 22 total), ~~mostly in the early visual and ventral stream,~~ showed a significant timing difference (black asterisks) between the time points at which

fmRI responses are most related to the contents of the first and the third epoch of video **with tighter confidence intervals in early visual regions (See Source Data).**

Supplementary Figure 6: Encoding the temporal dynamics of the BOLD signal.

Figure 6: Encoding the temporal dynamics of the BOLD signal.

Reviewer #2 (Remarks on code availability):

The codes are accessible under the directory and the codes are organized. Maybe it would be better if more instructions are provided within the scripts. However, I didn't install and run the application

We provide additional README files in the code directories with instructions for code execution. We also provide a GitHub repository of starter code in heavily commented Jupyter notebooks to familiarize the user with downloading the dataset, running basic analyses, and visualizing the results [<https://github.com/blahner/BOLDMomentsDataset>]. Note that we have moved all data and manuscript code to a repository in OpenNeuro.org for even greater accessibility [<https://openneuro.org/datasets/ds005165>].

References:

Wang, J., Yang, Z., Hu, X., Li, L., Lin, K., Gan, Z., ... & Wang, L. (2022). Git: A generative image-to-text transformer for vision and language. *arXiv preprint arXiv:2205.14100*.